# Improved Convergence Rate of Stochastic Gradient Langevin Dynamics with Variance Reduction and its Application to Optimization

**Yuri Kinoshita**
Department of Mathematical Informatics,
Graduate School of Information Science and Technology,
The University of Tokyo, Tokyo, Japan
`yuri-kinoshita111@g.ecc.u-tokyo.ac.jp`

**Taiji Suzuki**
Department of Mathematical Informatics,
Graduate School of Information Science and Technology,
The University of Tokyo, Tokyo, Japan
Center for Advanced Intelligence Project, RIKEN, Tokyo, Japan
`taiji@mist.i.u-tokyo.ac.jp`

## Abstract

The stochastic gradient Langevin Dynamics is one of the most fundamental algorithms to solve sampling problems and non-convex optimization appearing in several machine learning applications. Especially, its variance reduced versions have nowadays gained particular attention. In this paper, we study two variants of this kind, namely, the Stochastic Variance Reduced Gradient Langevin Dynamics and the Stochastic Recursive Gradient Langevin Dynamics. We prove their convergence to the objective distribution in terms of KL-divergence under the sole assumptions of smoothness and Log-Sobolev inequality which are weaker conditions than those used in prior works for these algorithms. With the batch size and the inner loop length set to $\sqrt{n}$, the gradient complexity to achieve an $\epsilon$-precision is $\tilde{O}((n + dn^{1/2}\epsilon^{-1})\gamma^2 L^2 \alpha^{-2})$, which is an improvement from any previous analyses. We also show some essential applications of our result to non-convex optimization.

## 1 Introduction

### 1.1 Background and Organization

Over the past decade, the gradient Langevin Dynamics (GLD) has gained particular attention for providing an effective tool for sampling from a Gibbs distribution, a fundamental task omnipresent in the field of machine learning and statistics, and for non-convex optimization, which is nowadays witnessing an unignorable empirical success. Notably, GLD is a stochastic differential equation (SDE) that can be viewed as the steepest descent flow of the Kullback-Leibler (KL) divergence towards the stationary Gibbs distribution in the space of measures endowed with the 2-Wasserstein metric (Jordan et al., 1998). As a consequence of the unique properties of GLD, its implementable discrete schemes and their ability to suitably track it have been the subject of a large number of studies.

The Euler-Maruyama scheme of GLD gives rise to an algorithm known as the Langevin Monte Carlo method (LMC). This algorithm is biased (Wibisono, 2018): that is, the distribution of the

36th Conference on Neural Information Processing Systems (NeurIPS 2022).

discrete scheme does not converge to the same as GLD. Nonetheless, it has been shown that this bias could be made arbitrarily small under certain assumptions by taking a sufficiently small step size (Dalalyan, 2017b; Vempala and Wibisono, 2019). Dalalyan (2017a,b) provided one of the first non-asymptotic rates of convergence of LMC for smooth log-concave distributions. Assumptions to obtain a non-asymptotic analysis and this controllable bias have been relaxed by further research to dissipativity and smoothness (Raginsky et al., 2017; Xu et al., 2018), and recently to Log-Sobolev inequality (LSI) and smoothness (Vempala and Wibisono, 2019). This relaxation of conditions is especially meaningful as the objective distribution nowadays tends to become more and more complicated beyond the classical assumption of log-concavity.

However, in the field of machine learning, the main function can often be formulated as the average of the loss function of an enormous number of training data points (Welling and Teh, 2011), which subsequently makes it difficult to calculate its full gradient. As a result, research on stochastic algorithms has been also conducted to avoid this computational burden (Chen et al., 2021; Dubey et al., 2016; Raginsky et al., 2017; Welling and Teh, 2011; Xu et al., 2018; Zou et al., 2018, 2019a,b, 2021). Welling and Teh (2011) introduced the concept of Stochastic Gradient Langevin Dynamics (SGLD) which combines the Stochastic Gradient Descent with LMC. This has been the subject of successful studies (Raginsky et al., 2017; Welling and Teh, 2011; Xu et al., 2018). Nevertheless, the variance of its stochastic gradient is too large, which leads to a slow convergence compared to LMC. Therefore, stochastic gradient Langevin Dynamics algorithms with variance reduction, such as the Stochastic Variance Reduced Gradient Langevin Dynamics (SVRG-LD), have been considered and their convergence has been thoroughly analyzed for both sampling (Dubey et al., 2016; Zou et al., 2018, 2019a, 2021) and optimization (Huang and Becker, 2021; Xu et al., 2018).

Dubey et al. (2016) first united SGLD with variance reduction techniques and proposed two new algorithms, namely, SVRG-LD and SAGA-LD. Chatterji et al. (2018) and Zou et al. (2018) proved the convergence rate of SVRG-LD to the target distribution in 2-Wasserstein distance for smooth log-concave distributions. Xu et al. (2018) showed the weak convergence of SVRG-LD under the smoothness and dissipativity conditions. They expanded the non-asymptotic analysis of Raginsky et al. (2017) to LMC and SVRG-LD, and improved the result for SGLD. Few years ago, Zou et al. (2019a) provided the gradient complexity of SVRG-LD to converge to the stationary distribution in 2-Wasserstein distance under the smoothness and dissipativity assumptions. This convergence can be even improved if we make a warm-start (Zou et al., 2021). While these works investigated algorithms with fixed hyperparameters, Huang and Becker (2021) additionally assumed a strict saddle and some other minor conditions to study SVRG-LD with a decreasing step size and improved its convergence in high probability to the second order stationary point. Zou et al. (2019b) also applied variance reduction techniques to the Hamiltonian Langevin Dynamics, or underdamped Langevin Dynamics in opposition to GLD also known as overdamped Langevin Dynamics. As we can observe, the current convergence analyses of the stochastic schemes with variance reduction are mostly restricted to log-concavity and dissipativity, and do not enjoy the same broad convergence guarantee with a concrete gradient complexity as LMC does under LSI and smoothness in terms of KL-divergence.

Therefore, in order to bridge this theoretical gap between LMC and stochastic gradient Langevin Dynamics with variance reduction, we study in this paper the convergence of the latter under the relaxed assumptions of smoothness and LSI. In Section 3, we study the convergence to the Gibbs distribution of SVRG-LD and the Stochastic Recursive Gradient Langevin Dynamics (SARAH-LD), another variant of stochastic gradient Langevin Dynamics with variance reduction inspired by the Stochastic Recursive Gradient algorithm (SARAH) of Nguyen et al. (2017a,b). On the other hand, optimization and sampling are only two sides of the same coin for GLD. That is why, in Section 4, we also investigate implications of Section 3 for non-convex optimization. We prove the convergence of SVRG-LD and SARAH-LD to the global minimum of dissipative functions and we provide their non-asymptotic rate of convergence. We also consider the additional Morse assumption and study its effect. Finally, we illustrate our main result with a simple experiment.

## 1.2 Contributions

The major contributions of this paper can be summarized as follows. We provide a non-asymptotic analysis of the convergence of SVRG-LD and SARAH-LD to the Gibbs distribution in terms of KL-divergence under smoothness and LSI which are weaker conditions than those used in prior works for these algorithms. KL-divergence is generally a stronger convergence criterion than both total

Table 1: Comparison of our main result with prior works (sampling). The first three works are about LMC. Compared to Vempala et al. (2019), with the same assumptions and criterion, the order of gradient complexity is improved from $n$ to $\sqrt{n}$. The others are about SVRG-LD except the last one which is about the Stochastic Gradient Hamiltonian Monte Carlo Methods with Recursive Variance Reduction. $\epsilon$ is the accuracy required on the criterion, $d$ is the dimension of the input of the main function, $n$ is the number of data points, and $L$ is the smoothness constant. * 2-Wass. stands for "2-Wasserstein", and conv. stands for "convergence". ** $\text{poly}(M, L)$ stands for a polynomial of $M$ and $L$.

| Method | Major Assumptions | Criterion* | Gradient Complexity** |
|--------|-------------------|-----------|----------------------|
| Dalalyan (2017a) | Smooth, Log-concave ($M$) | 2-Wass. | $\tilde{O}\left(\frac{nd}{\epsilon^2}\cdot\text{poly}(M,L)\right)$ |
| Xu et al. (2018) | Smooth, Dissipative | Weak conv. | $\tilde{O}\left(\frac{nd}{\epsilon}\right)\cdot e^{\tilde{O}(d)}$ |
| Vempala et al. (2019) | Smooth, Log-Sobolev ($\alpha$) | KL | $\tilde{O}\left(\frac{n}{\epsilon}\cdot d\gamma^2 L^2\alpha^{-2}\right)$ |
| Zou et al. (2018) | Smooth, Log-concave ($M$) | 2-Wass. | $\tilde{O}\left(n+\frac{L^{3/2}n^{1/2}d^{1/2}}{M^{3/2}\epsilon}\right)$ |
| Zou et al. (2019a) | Smooth, Dissipative | 2-Wass. | $\tilde{O}\left(n+\frac{n^{3/4}}{\epsilon^2}+\frac{n^{1/2}}{\epsilon^4}\right)\cdot e^{\tilde{O}(\gamma+d)}$ |
| Zou et al. (2021) | Smooth, Dissipative, Warm-start | TV | $\tilde{O}\left(\frac{\gamma^2}{\epsilon^2}\right)\cdot e^{\tilde{O}(d)}$ |
| Zou et al. (2019b) | Smooth, Dissipative | 2-Wass. | $\tilde{O}\left((n+\frac{n^{1/2}}{\epsilon^2\mu_*^{3/2}})\wedge\frac{\mu_*^{-2}}{\epsilon^4}\right)$ |
| **This paper** | Smooth, Log-Sobolev ($\alpha$) | KL | $\tilde{O}\left(\left(n+\frac{dn^{1/2}}{\epsilon}\right)\cdot\gamma^2 L^2\alpha^{-2}\right)$ |

variation (TV) and 2-Wasserstein distance as they can be controlled by KL-divergence under the LSI condition. Notably, we prove that, with the batch size and inner loop length set to $\sqrt{n}$, the gradient complexity to achieve an $\epsilon$-precision in terms of KL-divergence is $\tilde{O}((n + dn^{1/2}\epsilon^{-1})\gamma^2 L^2\alpha^{-2})$, which is better than any previous analyses. See Table 1 for a comparison with previous research in terms of assumptions, criterion and gradient complexity. We also prove the convergence of SVRG-LD and SARAH-LD to the global minimum under an additional assumption of dissipativity with a gradient complexity of $\tilde{O}((n + n^{1/2}\epsilon^{-1}dL\alpha^{-1})\gamma^2 L^2\alpha^{-2})$ which is better than previous work since it has almost all the time a dependence on $n$ of $O(\sqrt{n})$ and does not require the batch size and the inner loop length to depend on the accuracy $\epsilon$. On the other hand, we import the idea of Li and Erdogdu (2020) from product manifolds of spheres to the Euclidean space in order to show that under the additional assumption of Morse, the convergence in the Euclidean space can be accelerated by eliminating the exponential dependence on $1/\epsilon$.

## 1.3 Other Related Works

The theoretical study of GLD goes back to Chiang et al. (1987) who showed that global convergence could be achieved with a proper annealing schedule. This work did not specify how to implement this SDE, but Gelfand and Mitter (1991) filled this gap. Later, Borkar and Mitter (1999) proved an asymptotic convergence in terms of relative entropy for the discrete scheme of gradient Langevin Dynamics when the inverse temperature and the step size are kept constant.

The variance reduction technique, introduced to Langevin Dynamics by Dubey et al. (2016), was originally presented by Johnson and Zhang (2013) as Stochastic Variance Reduced Gradient (SVRG) to improve the convergence speed of Stochastic Gradient Descent. Other variance reduction techniques were also considered such as the Stochastic Recursive Gradient Langevin Dynamics (SARAH) from Nguyen et al. (2017a,b) which outperforms SVRG in non-convex optimization (Pham et al., 2020) and is used in many algorithms such as SSRGD (Li, 2019) and SpiderBoost (Wang et al., 2019).

Li and Erdogdu (2020) extended Vempala and Wibisono's result to Riemannian manifolds. One of the highlights of their work is that they showed the Log-Sobolev constant of the Gibbs distribution for a product manifold of spheres only depends on a polynomial of the inverse temperature under some particular conditions including Morse. We will adapt this result to our situation.

In the concurrent work of Balasubramanian et al. (2022) (especially Section 6), they also studied the convergence of stochastic schemes of GLD with more relaxed conditions than prior analyses. However, our contributions are not overshadowed by theirs, and we clarify the reasons. In Subsection

6.1 of their paper, Balasubramanian et al. (2022) focused on stochastic discrete schemes with finite variance and bias (which is not the case for SVRG-LD) and provided a first-order convergence guarantee in the space of measures equipped with the 2-Wasserstein distance. Subsection 6.2 proved a global convergence under some other conditions but most of these two analyses did not consider in particular the usual case in machine learning when $F$ is the average of some other functions, which leads to a generally worse gradient complexity than ours. Concerning this finite sum setting, Balasubramanian et al. (2022) investigated the Variance Reduced LMC algorithm (slightly different from SVRG-LD in this paper) in Subsection 6.3 and gave a first-order convergence under the sole assumption of smoothness. When restrained in our problem setting, the gradient complexity of SVRG-LD and SARAH-LD we provide is still considerably better (see Section 3 for more details).

## 1.4 Notation

We denote deterministic vectors by a lower case symbol (e.g., $x$) and random variables by an upper case symbol (e.g., $X$). The Euclidean norm is denoted by $\|\cdot\|$ for vectors and the inner product by $\langle \cdot, \cdot \rangle$. For matrices, $\|\cdot\|$ is the norm induced by the Euclidean norm for vectors. We only treat distributions absolutely continuous with respect to the Lebesgue measure in $\mathbb{R}^d$ for simplicity. Especially, throughout the paper, $\nu$ refers to the probability measure with the density function $\mathrm{d}\nu \propto e^{-\gamma F} \mathrm{d}x$, where $F$ is a function introduced below. $a \vee b$ is equivalent to $\max\{a, b\}$ and $a \wedge b$ to $\min\{a, b\}$. We also use the shorthand $\tilde{O}$ to hide logarithmic polynomials.

## 2 Preliminaries

In this section, we briefly explain the problem setting, necessary mathematical background and assumptions used in this paper.

### 2.1 Problem Setting and GLD

In Section 3, we consider sampling from a distribution written in the form $\mathrm{d}\nu \propto e^{-\gamma F} \mathrm{d}x$ where $\gamma$ is a positive constant (which corresponds to the inverse temperature) and $F : \mathbb{R}^d \to \mathbb{R}$ is formulated as $F(x) := \frac{1}{n} \sum_{i=1}^{n} f_i(x)$, the average of the loss function of $n$ training data points $\{x^{(i)}\}_{i=1}^{n}$. Here, $f_i(x) := f(x, x^{(i)})$ can be regarded as the loss of data $x^{(i)}$. For instance, $F$ can be the average of the negative log likelihood of $n$ training data points. In Section 4, we consider the non-convex optimization (minimization) of the same $F$ as above.

GLD can be described as the following stochastic differential equation (SDE):

$$\mathrm{d}X_t^{\mathrm{GLD}} = -\nabla F(X_t^{\mathrm{GLD}})\mathrm{d}t + \sqrt{2/\gamma}\mathrm{d}B(t), \tag{1}$$

where $\gamma > 0$ is called the inverse temperature parameter and $\{B(t)\}_{t \geq 0}$ is the standard Brownian motion in $\mathbb{R}^d$. It can be used for sampling since under some reasonable assumptions of $F$, the distribution $\rho_t^{\mathrm{GLD}}$ of $X_t^{\mathrm{GLD}}$ governed by SDE (1) converges to the invariant stationary distribution $\mathrm{d}\nu \propto e^{-\gamma F} \mathrm{d}x$, also known as the Gibbs distribution (Chiang et al., 1987). Moreover, as previously mentioned, this convergence is efficient in the sense that SDE (1) corresponds to the steepest descent flow of the Kullback-Leibler (KL) divergence towards the stationary distribution in the space of measures endowed with the 2-Wasserstein metric (Jordan et al., 1998). Alternatively, GLD can be interpreted as the composite optimization problem of a negative entropy and an expected function value as follows (Wibisono, 2018):

$$\min_{q:\mathrm{density}} \mathbb{E}_q[\gamma F] + \mathbb{E}_q[\log q].$$

The gradient flow is the well-known Fokker-Planck equation associated to SDE (1):

$$\frac{\partial \rho_t^{\mathrm{GLD}}}{\partial t} = \nabla \cdot (\rho_t^{\mathrm{GLD}} \nabla F) + \frac{1}{\gamma} \Delta \rho_t^{\mathrm{GLD}} = \frac{1}{\gamma} \nabla \cdot \left( \rho_t^{\mathrm{GLD}} \nabla \log \frac{\rho_t^{\mathrm{GLD}}}{\nu} \right). \tag{2}$$

This will be useful in our analysis. In addition to its potential for sampling, GLD can also be employed for non-convex optimization as the Gibbs distribution concentrates on the global minimum of $F$ for sufficiently large values of $\gamma$ (Hwang, 1980).

---

**Algorithm 1:** SVRG-LD / SARAH-LD

---

**1** input: step size $\eta > 0$, batch size $B$, epoch length $m$, inverse temperature $\gamma \geq 1$

**2** initialization: $X_0 = 0$, $X^{(0)} = X_0$

**3** **foreach** $s = 0, 1, \ldots, (K/m)$ **do**

**4** $\quad$ $v_{sm} = \nabla F(X^{(s)})$

**5** $\quad$ randomly draw $\epsilon_{sm} \sim N(0, I_{d \times d})$

**6** $\quad$ $X_{sm+1} = X_{sm} - \eta v_{sm} + \sqrt{2\eta/\gamma} \epsilon_{sm}$

**7** $\quad$ **foreach** $l = 1, \ldots, m-1$ **do**

**8** $\quad\quad$ $k = sm + l$

**9** $\quad\quad$ randomly pick a subset $I_k$ from $\{1, \ldots, n\}$ of size $|I_k| = B$

**10** $\quad\quad$ randomly draw $\epsilon_k \sim N(0, I_{d \times d})$

**11** $\quad\quad$ **if** *SVRG-LD* **then**

**12** $\quad\quad\quad$ $v_k = \frac{1}{B} \sum_{i_k \in I_k} (\nabla f_{i_k}(X_k) - \nabla f_{i_k}(X^{(s)})) + v_{sm}$

**13** $\quad\quad$ **else if** *SARAH-LD* **then**

**14** $\quad\quad\quad$ $v_k = \frac{1}{B} \sum_{i_k \in I_k} (\nabla f_{i_k}(X_k) - \nabla f_{i_k}(X_{k-1})) + v_{k-1}$

**15** $\quad\quad$ **end**

**16** $\quad\quad$ $X_{k+1} = X_k - \eta v_k + \sqrt{2\eta/\gamma} \epsilon_k$

**17** $\quad$ **end**

**18** $\quad$ $X^{(s+1)} = X_{(s+1)m}$

**19** **end**

---

## 2.2 Algorithms of GLD

Applying the Euler-Maruyama scheme to (1), we obtain the Langevin Monte Carlo (LMC)

$$X_{k+1} = X_k - \eta \nabla F(X_k) + \sqrt{2\eta/\gamma} \epsilon_k,$$

where $\eta$ is called the step size. This is similar to the gradient descent except the additional Gaussian noise $\sqrt{2\eta/\gamma} \epsilon_k$, where $\epsilon_k \sim N(0, I_{d \times d})$ and $I_{d \times d}$ is the $d \times d$ unit matrix. In the case $n$ is huge and the computation of $\nabla F$ is too difficult, we are incited to use stochastic gradient methods in analogy to stochastic gradient optimization. This gives

$$X_{k+1} = X_k - \eta v(X_k) + \sqrt{2\eta/\gamma} \epsilon_k,$$

where $v(X_k)$ is the stochastic gradient. When $v(X_k)$ is defined as $\frac{1}{B} \sum_{i_k \in I_k} \nabla f_{i_k}(X_k)$, where $B$ is called the batch size and $I_k$ is a random subset uniformly chosen from $\{1, \ldots, n\}$ such that $|I_k| = B$, we obtain the Stochastic Gradient Langevin Dynamics (SGLD). As this method exhibits a slow convergence, it has been popular to use variance reduction methods such as the Stochastic Variance Reduced Gradient Langevin Dynamics (SVRG-LD) where $v(X_k) = \frac{1}{B} \sum_{i_k \in I_k} (\nabla f_{i_k}(X_k) - \nabla f_{i_k}(X^{(s)})) + \nabla F(X^{(s)})$. Details of this algorithm is stated in Algorithm 1. $X^{(s)}$ is a reference point updated every $m$ steps so that $X_{sm} = X^{(s)}$. As we can observe in Lemma A.4, around the optimal point, the variance of the stochastic gradient is indeed decreased as $X^{(s)}$ and $X_k$ are both close to each other. We can also easily extend some successful stochastic gradient algorithms to Langevin Dynamics. Hence, we are motivated to extend the Stochastic Recursive Gradient Algorithm (SARAH) to Langevin Dynamics since we can expect that some bottlenecks of the analysis of SVRG-LD can be removed in that of SARAH-LD as subtracting the previous stochastic gradient enables a stabler performance than SVRG-LD. This algorithm can be described as Algorithm 1 with $v(X_k) = \frac{1}{B} \sum_{i_k \in I_k} (\nabla f_{i_k}(X_k) - \nabla f_{i_k}(X_{k-1})) + v(X_{k-1})$.

**Definition 1.** *We define $\rho_k$ as the distribution of $X_k$ generated at the $k$th step of SVRG-LD, and similarly $\phi_k$ for SARAH-LD.*

## 2.3 Assumptions

The assumptions used throughout this paper can be summarized as follows.

**Assumption 1.** *For all $i = 1, \ldots, n$, $\nabla f_i$ is twice differentiable, and $\forall x, y \in \mathbb{R}^d$, $\|\nabla^2 f_i(x)\| \leq L$. In other words, $f_i$ $(i = 1, \ldots, n)$ and $F$ are L-smooth.*

**Assumption 2.** *Distribution $\nu$ satisfies the Log-Sobolev inequality (LSI) with a constant $\alpha$. That is, for all probability density functions $\rho$ absolutely continuous with respect to $\nu$, the following holds:*

$$H_\nu(\rho) \leq \frac{1}{2\alpha} J_\nu(\rho),$$

*where $H_\nu(\rho) := \mathbb{E}_\rho \left[ \log \frac{\rho}{\nu} \right]$ is the KL-divergence of $\rho$ with respect to $\nu$, and $J_\nu(\rho) := \mathbb{E}_\rho \left[ \left\| \nabla \log \frac{\rho}{\nu} \right\|^2 \right]$ is the relative Fisher information of $\rho$ with respect to $\nu$.*

The recent work of Vempala and Wibisono (2019) motivates us to use the combination of smoothness and LSI for the analysis of SVRG-LD and SARAH-LD. Indeed, they showed that these conditions were enough to assure for the Euler-Maruyama scheme an exponentially fast convergence and a bias controllable by the step size. Under smoothness, LSI is not only the necessary condition of log-concavity and dissipativity, but is also robust to bounded perturbation and Lipschitz mapping, contrary to log-concavity (Vempala and Wibisono, 2019). For example, for any distribution $\mathrm{d}\nu$ that satisfies LSI and bounded function $B : \mathbb{R}^d \to \mathbb{R}$, $\mathrm{d}\tilde{\nu} \propto \mathrm{e}^B \mathrm{d}\nu$ satisfies LSI as well (Holley and Stroock, 1986). Moreover, while KL-divergence is not in general convex with regard to the Wasserstein geodesic, thanks to LSI, the Polyak-Łojaciewicz condition is satisfied. It is well-known that LSI suffices to realize an exponential convergence for the case of continuous time Langevin Dynamics (Vempala and Wibisono, 2019). That is why, it is actually both useful and natural to suppose LSI in this context. Note that under $L$-smoothness of $F$ and LSI with constant $\alpha$ for $\mathrm{d}\nu \propto \mathrm{e}^{-\gamma F} \mathrm{d}x$, it holds that $\alpha \leq \gamma L$ (Vempala and Wibisono, 2019).

As for optimization, we additionally use the following conditions.

**Assumption 3.** *$F$ is $(M, b)$-dissipative. That is, there exist constants $M > 0$ and $b > 0$ such that for all $x \in \mathbb{R}^d$ the following holds: $\langle \nabla F(x), x \rangle \geq M \|x\|^2 - b$.*

**Assumption 4** (Li and Erdogdu (2020), Assumption 3.3 adapted). *$F$ satisfies the Morse condition. That is, for all eigenvalues of the Hessian of stationary points, there exists a constant $\lambda^\dagger \in (0, 1]$ such that*

$$\lambda^\dagger \leq \inf \left\{ \left| \lambda_i \left( \nabla^2 F(x) \right) \right| \mid \nabla F(x) = 0, \ i \in 1, \ldots, d \right\}.$$

*Furthermore, for the set $\mathcal{S}$ of stationary points that are not a global minimum, $\sup_{x \in \mathcal{S}} \lambda_{\min} \left( \nabla^2 F(x) \right) \leq -\lambda^\dagger$.*

**Assumption 5.** *$\nabla^2 f_i$ is $L'$-Lipschitz and without loss of generality, we let $\min_{x \in \mathbb{R}^d} F(x) = 0$.*

**Assumption 6.** *$F$ has a unique global minimum.*

Smoothness and dissipativity are a classical combination of assumptions for this kind of problem setting (Raginsky et al., 2017; Xu et al., 2018; Zou et al., 2019a). We assume dissipativity instead of LSI for non-convex optimization in order to obtain an explicit value of the Log-Sobolev constant of $\mathrm{d}\nu \propto \mathrm{e}^{-\gamma F} \mathrm{d}x$ in function of the inverse temperature parameter $\gamma$ (see Property C.3), making a non-asymptotic analysis possible. Furthermore, Assumptions 4 to 6 can ameliorate the exponential dependence of the inverse of the Log-Sobolev constant on the inverse temperature parameter to a polynomial one (see Property C.4).

# 3 Main Results

In this section, we state our main results which prove that SVRG-LD and SARAH-LD (Algorithm 1) achieve an exponentially fast convergence to the Gibbs distribution and a controllable bias in terms of KL-divergence under the sole assumptions of LSI and smoothness. We provide their gradient complexity as well. The proofs can be found in Appendix A and B respectively.

## 3.1 Improved Convergence of SVRG-LD

Our analysis shows that the convergence of SVRG-LD to the stationary distribution $\mathrm{d}\nu \propto \mathrm{e}^{-\gamma F} \mathrm{d}x$ can be formulated as the theorem below.

**Theorem 1.** *Under Assumptions 1 and 2, $0 < \eta < \frac{\alpha}{16\sqrt{6}L^2 m\gamma}$, $\gamma \geq 1$ and $B \geq m$, for all $k = 1, 2, \ldots$, the following holds in the update of SVRG-LD where $\Xi = \frac{(n-B)}{B(n-1)}$ :*

$$H_\nu(\rho_k) \leq \mathrm{e}^{-\frac{\alpha\eta}{\gamma}k} H_\nu(\rho_0) + \frac{224\eta\gamma dL^2}{3\alpha} \left( 2 + 3\Xi + 2m\Xi \right).$$

We observe that the bias term of the upper bound, which is the second term linearly dependent on $\eta$, can be easily controlled while the first term exponentially converges to 0 with $k \to \infty$. This is more precisely formulated in the following corollary.

**Corollary 1.1.** *Under the same assumptions as Theorem 1, for all $\epsilon \geq 0$, if we choose step size $\eta$ such that $\eta \leq \frac{3\alpha\epsilon}{448\gamma dL^2}$, then a precision $H_\nu(\rho_k) \leq \epsilon$ is reached after $k \geq \frac{\gamma}{\alpha\eta} \log \frac{2H_\nu(\rho_0)}{\epsilon}$ steps. Especially, if we take $B = m = \sqrt{n}$ and the largest permissible step size $\eta = \frac{\alpha}{16\sqrt{6}L^2\sqrt{n}\gamma} \wedge \frac{3\alpha\epsilon}{448dL^2\gamma}$, then the gradient complexity becomes*

$$\tilde{O}\left(\left(n + \frac{dn^{\frac{1}{2}}}{\epsilon}\right) \cdot \frac{\gamma^2 L^2}{\alpha^2}\right).$$

This gradient complexity is an improvement compared with prior works for three reasons. First of all, we provide a non-asymptotic analysis of the convergence of SVRG-LD under smoothness and Log-Sobolev inequality which are conditions weaker than those (e.g., log-concavity or dissipativity) used in prior works for these algorithms. Moreover, we prove it in terms of KL-divergence which is generally a stronger convergence criterion than both total variation (TV) and 2-Wasserstein distance as they can both be controlled by KL-divergence under the LSI condition. For instance, TV was used by Zou et al. (2021) and 2-Wasserstein distance by Dalalyan (2017a) and Zou et al. (2019a). KL-divergence makes it possible to unify these two different criteria. Finally, while prior research generally used Girsanov's theorem which generates a bias term that accumulates through the iteration (see for example Raginsky et al. (2017) and Xu et al. (2018)), we solve this issue by taking benefit of the exponential convergence of GLD to the Gibbs distribution under LSI and smoothness that enables us to forget about past bias. That way, with the batch size and inner loop set to $\sqrt{n}$, the gradient complexity to achieve an $\epsilon$-precision in terms of KL-divergence becomes $\tilde{O}((n + dn^{1/2}\epsilon^{-1})\gamma^2 L^2\alpha^{-2})$, which is better than previous analyses. For example, Vempala and Wibisono (2019) provided a gradient complexity of $\tilde{O}\left(n\epsilon^{-1}d\gamma^2 L^2\alpha^{-2}\right)$ for LMC under Assumptions 1 and 2, and Zou et al. (2019a) a gradient complexity of $\tilde{O}(n + n^{3/4}\epsilon^{-2} + n^{1/2}\epsilon^{-4}) \cdot e^{\tilde{O}(\gamma+d)}$ for SVRG-LD under Assumptions 1 and 3. Note that the dependence on the dimension $d$ is not improved since $\alpha^{-1}$ may exponentially depend on $d$. Recently, Zou et al. (2019b) proposed the Stochastic Gradient Hamiltonian Monte Carlo Methods with Recursive Variance Reduction with a gradient complexity of $\tilde{O}((n + n^{1/2}\epsilon^{-2}\mu_*^{-3/2}) \wedge \mu_*^{-2}\epsilon^{-4})$ in terms of 2-Wasserstein distance. Even though their algorithm is based on the underdamped Langevin Dynamics whose discrete schemes use to perform better than those of the overdamped Langevin Dynamics such as SVRG-LD, our gradient complexity, which applies to a broader family of distributions, is almost the same except for a small interval of $\epsilon$, but we do not require the batch size $B$ and the inner loop length $m$ to depend on $\epsilon$ while Zou et al. (2019b) do, i.e., $B \lesssim B_0^{1/2}$, $m = O(B_0/B)$, where $B_0 = \tilde{O}\left(\epsilon^{-4}\mu_*^{-1} \wedge n\right)$. This strengthens the importance of our result since it shows that adapting this analysis to other stochastic schemes of GLD is promising and could lead to tighter bounds and relaxation of conditions. See Table 1 for a summary. Concerning the concurrent work of Balasubramanian et al. (2022), under the sole assumption of smoothness, they provided a gradient complexity of $O(L^2 d^2 n/\epsilon^2)$ for the Variance Reduced LMC algorithm that updates the stochastic gradient differently as SVRG-LD and SARAH-LD. This is almost the square of our result, and in some extent, our work can be interpreted as an acceleration of their result with a slightly stronger additional condition than Poincaré inequality.

**Proof Sketch** Proceeding in a similar way as Vempala and Wibisono (2019), we evaluate how $H_\nu(\rho_k)$ decreases at each step as shown in Theorem A.1 of Appendix A. This is realized by comparing the evolution of the continuous-time GLD for time $\eta$ and one step of SVRG-LD. Since we use a stochastic gradient, we need at the same time to evaluate the variance of the stochastic gradient. Theorem 1 can be obtained by recursively solving the inequality derived in Theorem A.1.

### 3.2 Convergence Analysis of SARAH-LD

As for SARAH-LD, its convergence to the stationary distribution $d\nu \propto e^{-\gamma F} dx$ can be formulated as the theorem below. Interestingly, we obtain the same result as SVRG-LD (Theorem 1) but we do not require $B \geq m$ anymore.

**Theorem 2.** *Under Assumptions 1 and 2, $0 < \eta < \frac{\alpha}{16\sqrt{2}L^2 m\gamma}$ and $\gamma \geq 1$, for all $k = 1, 2, \ldots$, the following holds in the update of SARAH-LD where $\Xi = \frac{(n-B)}{B(n-1)}$ :*

$$H_\nu(\phi_k) \leq \mathrm{e}^{-\frac{\alpha\eta}{\gamma}k} H_\nu(\phi_0) + \frac{32\eta\gamma dL^2}{3\alpha}\left(2 + \Xi + 2m\Xi\right).$$

This is the first convergence guarantee of SARAH-LD in this problem setting so far, and it leads to the following gradient complexity.

**Corollary 2.1.** *Under the same assumptions as Theorem 2, for all $\epsilon \geq 0$, if we choose step size $\eta$ such that $\eta \leq \frac{3\alpha\epsilon}{64\gamma dL^2}\left(2 + \Xi + 2m\Xi\right)^{-1}$, then a precision $H_\nu(\phi_k) \leq \epsilon$ is reached after $k \geq \frac{\gamma}{\alpha\eta}\log\frac{2H_\nu(\phi_0)}{\epsilon}$ steps. Especially, if we take $B = m = \sqrt{n}$ and the largest permissible step size $\eta = \frac{\alpha}{16\sqrt{2}L^2\sqrt{n}\gamma} \wedge \frac{3\alpha\epsilon}{320dL^2\gamma}$, then the gradient complexity becomes*

$$\tilde{O}\left(\left(n + \frac{dn^{\frac{1}{2}}}{\epsilon}\right)\cdot\frac{\gamma^2 L^2}{\alpha^2}\right).$$

The reason why we obtain the same gradient complexity for both SARAH-LD and SVRG-LD (except better coefficients for SARAH-LD) is that in our analysis, the Brownian noise added at each step of the Langevin Dynamics plays the role of a fundamental bottleneck that even SARAH-LD could not eliminate, and we still need to set $B = m = \sqrt{n}$. We can hypothesize that this order of gradient complexity might be tight for variance-reduced stochastic gradient Langevin Dynamics algorithms.

## 4 Some Applications to Non-Convex Optimization

Here, we apply our main results to non-convex optimization. Thanks to our analysis applicable to a broader family of probability distributions satisfying LSI, the additional conditions we pose in this section are mainly reflected in the concrete formulation of the Log-Sobolev constant, which keeps our study simple and clear. The proofs can be found in Appendix C. Since SVRG-LD and SARAH-LD exhibited almost the same performance in sampling, we can simultaneously analyse them. We first prove the convergence to the global minimum of SVRG-LD and SARAH-LD without clarifying the explicit formulation of the Log-Sobolev constant in function of $\gamma$.

**Theorem 3.** *Using SVRG-LD or SARAH-LD, under Assumptions 1 to 3, $0 < \eta < \frac{\alpha}{16\sqrt{6}L^2 m\gamma}$, $\gamma \geq \frac{4d}{\epsilon}\log\left(\frac{\mathrm{e}L}{M}\right) \vee \frac{8db}{\epsilon^2} \vee 1 \vee \frac{2}{M}$ and $B \geq m$, if we take $B = m = \sqrt{n}$ and the largest permissible step size $\eta = \frac{\alpha}{16\sqrt{6}L^2\sqrt{n}\gamma} \wedge \frac{3}{1792}\frac{\alpha^2\epsilon}{L^2 d\gamma}$, the gradient complexity to reach a precision of*

$$\mathbb{E}_{X_k}[F(X_k)] - F(X^*) \leq \epsilon$$

*is*

$$\tilde{O}\left(\left(n + \frac{n^{\frac{1}{2}}}{\epsilon}\cdot\frac{dL}{\alpha}\right)\frac{\gamma^2 L^2}{\alpha^2}\right),$$

*where $\alpha$ is a function of $\gamma$, and $X^*$ is the global minimum of $F$.*

**Remark 1.** *Under Assumptions 1 and 3, Assumption 2 is negligible as shown in Property C.2.*

Under Assumptions 1 to 3 only, this leads to a gradient complexity which exponentially depends on the inverse of the precision level $\epsilon$ as shown in the next corollary since the inverse of the Log-Sobolev constant exponentially depends on $\gamma$.

**Corollary 3.1.** *Under the same assumptions as Theorem 3, taking $\gamma = i(\epsilon) := \frac{4d}{\epsilon}\log\left(\frac{\mathrm{e}L}{M}\right) \vee \frac{8db}{\epsilon^2} \vee 1 \vee \frac{2}{M}$, we obtain a gradient complexity of*

$$\tilde{O}\left(\left(n + \frac{n^{\frac{1}{2}}}{\epsilon}\cdot\frac{dL}{C_1 i(\epsilon)}\mathrm{e}^{C_2 i(\epsilon)}\right)L^2\mathrm{e}^{2C_2 i(\epsilon)}\right)$$

*since $\alpha = \gamma C_1 \mathrm{e}^{-C_2\gamma}$ (Property C.3).*

The second term with $n^{1/2}$ is almost all the time dominant since it has a factor that exponentially depends on $1/\epsilon$ and the first term not. This dependence on $n$ of $O(n^{1/2})$ is the best so far for these algorithms. Moreover, comparing with the gradient complexity $\tilde{O}\left(n^{1/2}\lambda^{-4}\epsilon^{-5/2}\right) \cdot e^{\tilde{O}(d)}$, also of order $n^{1/2}$, provided by Xu et al. (2018) who used SVRG-LD and the same assumptions, our gradient complexity is an improvement since their analysis required a batch size $B$ and an inner loop length $m$ that strongly depend on $\epsilon$ (i.e., $B = \sqrt{n}\epsilon^{-3/2}$, $m = \sqrt{n}\epsilon^{3/2}$) and ours does not. Note that the dependence of the gradient complexity of Xu et al. (2018) on $1/\epsilon$ is not necessarily better than ours as $\lambda$ is actually the spectral gap of the discrete-time Markov chain generated by (1) and its inverse exponentially depends on $1/\epsilon$ as well. Although Xu et al. (2018) did not investigate the explicit nature of $\lambda$, this is supported by Raginsky et al. (2017) who proved this exponential dependence for the spectral gap of the continuous-time SDE and by Mattingly et al. (2002) who showed the spectral gap of continuous-time SDE and that of discrete-time version are almost the same in this context.

**Analysis under the Morse condition**  Now, under the additional Assumptions 4 to 6, it is interesting to note that a *polynomial dependence* on $1/\epsilon$ is achieved as the following corollary shows.

**Corollary 3.2.** *Under the same assumptions as Theorem 3 and Assumptions 4 to 6, taking $\gamma = j(\epsilon) := \frac{4d}{\epsilon} \log\left(\frac{eL}{M}\right) \vee \frac{8db}{\epsilon^2} \vee 1 \vee \frac{2}{M} \vee C_\gamma$, where $C_\gamma$ is a constant independent of $\epsilon$ defined in Property C.4, we obtain a gradient complexity of*

$$\tilde{O}\left(\left(n + \frac{n^{\frac{1}{2}}}{\epsilon} \cdot \frac{dL}{C_3} j(\epsilon)\right) C_3^2 j(\epsilon)^4 L^2\right),$$

*since $\alpha = C_3/\gamma$ (Property C.4).*

The crux of this corollary is Property C.4. To prove this, we show like Li and Erdogdu (2020) that $\nu$ satisfies the Poincaré inequality with a constant independent of $\gamma$. Since it is not hard to show this around the global minimum, we can step by step extend the set where this inequality holds by a Lyapunov argument (Theorems D.1 and D.2). The essential difference between this analysis and that of Li and Erdogdu (2020) is that we do not work on compact manifolds anymore. Some rather minor difficulties emerge as we cannot employ the compactness but they can be addressed by supposing dissipativity which assures a quadratic growth for large $x$.

**Remark 2.** *These results do not definitively assert that SARAH-LD and SVRG-LD show the exact same performance in terms of optimization. Indeed, suppose we are close enough to the global optimum. Then, a big noise is not necessary anymore since it is more important to stably converge to the global minimum. Here, we should be able to significantly decrease the noise $\epsilon_k$, and the bottleneck from the noise should disappear. In this case, SARAH-LD would perform better than SVRG-LD as we approach the original non-convex optimization setting where SARAH outperforms SVRG.*

**Remark 3.** *We also investigated an annealed version of SVRG-LD and SARAH-LD but could not ameliorate the gradient complexity. The detailed analysis can be found in Appendix E.*

## 5  Experiment

In this section, we illustrate our main result with a simple experiment.[1] We follow the same problem setting as that of Welling and Teh (2011) in Section 5.1. That is, we aim to sample from a non-log-concave posterior distribution $p(\theta|x) \propto p(\theta) \prod_{i=1}^{n} p(x_i|\theta)$ where $\{x_i\}_{i=1}^{n}$ is sampled from $p(x|\theta)$, a distribution parameterized by $\theta = (\theta_1, \theta_2)$. The prior $p(\theta)$ and the distribution of $x$ parametrized by $\theta$ are respectively defined as $\theta_1 \sim N(0, 10)$, $\theta_2 \sim N(0, 1)$ and $x \sim 1/2N(\theta_1, 2) + 1/2N(\theta_1 + \theta_2, 2)$. Here, we set $n = 10000$, $\theta_1 = 0$ and $\theta_2 = 1$. Using the obtained 10000 samples, we simulated 1000 points of SVRG-LD with the inner loop length $m = n/B$ and different batch sizes $B$, namely, 100, 1000 and 10000 so that $B \geq m$ as required in Theorem 1. Evolution of KL-divergence between the true posterior, estimated by the Metropolis-adjusted Langevin algorithm, and that simulated by SVRG-LD is plotted in Figure 1. KL-divergence was approximated following Pérez-Cruz (2008).

As we can observe, Figure 1 correctly reproduces the theoretical bound of Theorem 1, with an exponential convergence in the beginning and a persistent bias due to the use of a discrete scheme and mini-batches. The fastest convergence in terms of gradient complexity under the condition $B \geq m$

---

[1]Source code can be found in `https://github.com/yuri-k111/NeurIPS2022_code`

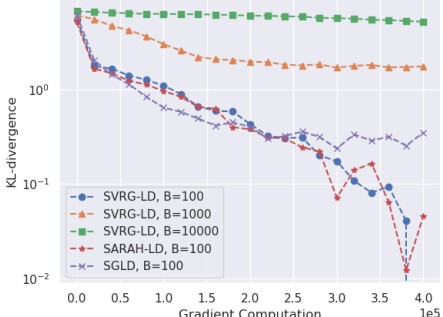

Figure 1: KL-divergence between the true and the simulated posterior. 1000 points were simulated for each algorithm with step size $\eta = 0.00001$. The inner loop length $m$ for SVRG-LD was defined as $n/B$, and initial points were randomly drawn from the standard normal distribution. 1 gradient computation refers to one computation of $\nabla f_i$.

is achieved by SVRG-LD with $B = \sqrt{n}$, which confirms our main theorem. Furthermore, with this best batch size, we also simulated 1000 points of SGLD and SARAH-LD as shown in Figure 1 as well. While SGLD and SVRG-LD have similar convergence speed in the beginning, the latter eventually achieves a higher precision thanks to the variance reduction method adopted in this scheme. SARAH-LD exhibits a similar performance as SVRG-LD, which agrees with Theorem 2.

## 6    Discussion and Conclusion

The main limitations of our work reside in the gap between practice and theory. Indeed, while our paper supposes assumptions quite standard in the literature of GLD, it cannot explain the whole empirical success that machine learning is currently experiencing. Some choices of parameters may also seem different than the practical use. However, compared to previous work, we succeeded in proving convergence of GLD with the popular stochastic gradient with relaxed conditions, and deleting the dependence of batch size and inner loop length on epsilon, which are all more realistic situations than prior work. The theoretical study in machine learning and deep learning precisely plays the role of filling as much as possible this large gap, and our work could be regarded as a further step forward to achieve this goal. Furthermore, in this paper, we focused on the pure sampling and optimization performance of the algorithms, and some of the drawbacks are simply due to this fact. For example, another limitation is that we did not investigate the generalization error in Section 4, but this was only outside the scope of this work.

In conclusion, we analysed the convergence rate of stochastic gradient Langevin Dynamics with variance reduction under smoothness and LSI and its application to optimization. In Section 3, we proved the convergence of SVRG-LD in terms of KL-divergence with more relaxed conditions (LSI and smoothness) and with a better gradient complexity than previous works. We also expanded SARAH to SARAH-LD and showed that this algorithm enjoyed the same advantages as SVRG-LD with only an improvement in the coefficients of the gradient complexity. These results led us to apply SVRG-LD and SARAH-LD to non-convex optimization in Section 4. We provided the global convergence and a non-asymptotic analysis of SVRG-LD and SARAH-LD. We obtained better conditions than prior works. Furthermore, we showed that under the additional assumption including Morse and Hessian Lipschitzness, the gradient complexity could be ameliorated, eliminating the exponential dependence on the inverse of the required error.

## Acknowledgments and Disclosure of Funding

This study was partially supported by Japan Digital Design and JST CREST. We would like to thank Mufan Li and Murat A. Erdogdu for their valuable comments to fix some issues in the proof in Appendix D and their suggestion to improve it. We also thank anonymous reviewers for their feedback.

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
