# A  Proof of Theorem 1 and Corollary 1.1

In this Section, to clearly differentiate from SARAH-LD, we redefine the random variable generated at the $k$-th step of SVRG-LD (Algorithm 1) as $Y_k$ and the stochastic gradient as $v_k^{(Y)}$. The distribution of $Y_k$ is $\rho_k$.

## A.1  Preparation for the Proof

We first prepare some lemmas.

**Lemma A.1.** *Under Assumption 1,*

$$\mathbb{E}_\nu[\|\nabla F\|^2] \leq dL/\gamma.$$

*Proof.* As $d\nu = e^{-\gamma F}dx$, using integration by parts, we obtain

$$\mathbb{E}_\nu[\|\nabla(\gamma F)\|^2] = \mathbb{E}_\nu[\|\Delta(\gamma F)\|].$$

Now, since $F$ is $L$-smooth by Assumption 1, $\nabla^2 F \preceq LI$ holds, which implies $\Delta F \leq dL$. As a result,

$$\mathbb{E}_\nu[\|\nabla F\|^2] = \frac{1}{\gamma}\mathbb{E}_\nu[\|\Delta F\|] \leq \frac{dL}{\gamma}.$$

$$Q.E.D$$

The relation between 2-Wasserstein distance and KL-divergence is given by the following inequality.

**Lemma A.2.** *Under Assumption 2, $\nu$ satisfies the following Talagrand's inequality with the same Log-Sobolev constant $\alpha$:*

$$\frac{\alpha}{2}W_2(\rho, \nu)^2 \leq H_\nu(\rho). \tag{3}$$

**Remark A.1.** *See Theorem 1 of Otto and Villani (2000) for a proof of Lemma A.2.*

The following two lemmas that bound $\mathbb{E}[\|v_k^{(\mathrm{Y})}\|^2]$ and the variance of the stochastic gradient $v_k^{(\mathrm{Y})}$ with the KL-divergences $H_\nu(\rho_k), H_\nu(\rho_{k-1}), \dots$ are crucial in our proof.

**Lemma A.3.** *Under Assumption 1, suppose Talagrand's inequality (3) holds for $\nu$ with a constant $\alpha$, then for all $k = sm + r$, where $s \in \mathbb{N} \cup \{0\}$ and $r = 0, \dots, m-1$, the following holds in the update of SVRG-LD:*

$$\mathbb{E}_{Y_k, I_k, Y^{(s)}}[\|v_k^{(\mathrm{Y})}\|^2] \leq \Lambda' H_\nu(\rho_{sm+r}) + T + \sum_{i=0}^{r-1} S(S+1)^{r-i-1}\left(\Lambda' H_\nu(\rho_{sm+i}) + T\right),$$

*where $\Lambda = \left(1 + \frac{2(n-B)}{B(n-1)}\right)$, $\Xi = \frac{(n-B)}{B(n-1)}$,*

$$\Lambda' = \frac{4L^2}{\alpha}\Lambda,$$

$$S = 4L^2 m\eta^2 \Xi,$$

*and*

$$T = \frac{2dL}{\gamma}\Lambda + \frac{8\eta mdL^2}{\gamma}\Xi.$$

*Proof.* Let $v_i^{(1)}(Y_k) := \nabla f_i(Y_k) - \nabla f_i(Y^{(s)}) + \nabla F(Y^{(s)})$, then

$$
\begin{aligned}
\mathbb{E}_{Y_k,I_k,Y^{(s)}}[\|v_k^{(Y)}\|^2] =& \mathbb{E}_{Y_k,I_k,Y^{(s)}}\left[\left\|\frac{1}{B}\sum_{i\in I_k}v_i^{(1)}(Y_k)\right\|^2\right]\\
=& \frac{1}{B^2}\mathbb{E}_{Y_k,I_k,Y^{(s)}}\left[\sum_{i\neq i',\{i,i'\}\in I_k}\left\langle v_i^{(1)}(Y_k),v_{i'}^{(1)}(Y_k)\right\rangle\right]\\
&+ \frac{1}{B^2}\mathbb{E}_{Y_k,I_k,Y^{(s)}}\left[\sum_{i\in I_k}\|v_i^{(1)}(Y_k)\|^2\right]\\
=& \frac{B-1}{Bn(n-1)}\mathbb{E}_{Y_k,Y^{(s)}}\left[\sum_{i\neq i'}\left\langle v_i^{(1)}(Y_k),v_{i'}^{(1)}(Y_k)\right\rangle\right]\\
&+ \frac{1}{B}\mathbb{E}_{Y_k,i,Y^{(s)}}\left[\|v_i^{(1)}(Y_k)\|^2\right]\\
&(i \text{ follows the uniform distribution under } \{1,\dots,n\})\\
=& \frac{B-1}{Bn(n-1)}\mathbb{E}_{Y_k,Y^{(s)}}\left[\sum_{i,i'}\left\langle v_i^{(1)}(Y_k),v_{i'}^{(1)}(Y_k)\right\rangle\right]\\
&- \frac{B-1}{B(n-1)}\mathbb{E}_{Y_k,i,Y^{(s)}}\left[\|v_i^{(1)}(Y_k)\|^2\right]\\
&+ \frac{1}{B}\mathbb{E}_{Y_k,i,Y^{(s)}}\left[\|v_i^{(1)}(Y_k)\|^2\right]\\
=& \frac{(B-1)n}{B(n-1)}\mathbb{E}_{Y_k}[\|\nabla F(Y_k)\|^2] + \frac{n-B}{B(n-1)}\mathbb{E}_{Y_k,i,Y^{(s)}}[\|v_i^{(1)}(Y_k)\|^2],
\end{aligned}
$$

where we used $\frac{1}{n}\sum_{i=1}^n v_i^{(1)}(Y_k) = \nabla F(Y_k)$ for the last equality.

As a result, taking into account $\frac{(B-1)n}{B(n-1)} - 1 = \frac{B-n}{B(n-1)} \leq 0$,

$$
\mathbb{E}_{Y_k,I_k,Y^{(s)}}[\|v_k^{(Y)}\|^2] = \mathbb{E}_{Y_k}[\|\nabla F(Y_k)\|^2] + \frac{n-B}{B(n-1)}\mathbb{E}_{Y_k,i,Y^{(s)}}[\|v_i^{(1)}(Y_k)\|^2]. \tag{4}
$$

Choosing an optimal coupling $Y_k \sim \rho_k$ and $Y^* \sim \nu$ so that $\mathbb{E}[\|Y_k - Y^*\|^2] = W_2(\rho_k,\nu)^2$, we obtain

$$
\begin{aligned}
\mathbb{E}_{Y_k}[\|\nabla F(Y_k)\|^2] &\leq 2\mathbb{E}_{Y_k,Y^*}[\|\nabla F(Y_k) - \nabla F(Y^*)\|^2] + 2\mathbb{E}_{Y^*}[\|\nabla F(Y^*)\|^2]\\
&\leq 2L^2\mathbb{E}[\|Y_k - Y^*\|^2] + 2dL/\gamma\\
&= 2L^2 W_2(\rho_k,\nu)^2 + 2dL/\gamma\\
&\leq \frac{4L^2}{\alpha}H_\nu(\rho_k) + 2dL/\gamma, \tag{5}
\end{aligned}
$$

where we used Lemma A.1 and the smoothness of $F$ for the second inequality, the definition of $W_2$ for the equality and Talagrand's inequality (Lemma A.2) for the last inequality.

Moreover,

$$
\begin{aligned}
\mathbb{E}_{Y_k,i,Y^{(s)}}[\|v_i^{(1)}(Y_k)\|^2] =& \mathbb{E}_{Y_k,i,Y^{(s)}}\left[\left\|\nabla f_i(Y_k) - \nabla f_i(Y^{(s)}) + \nabla F(Y^{(s)})\right\|^2\right] \\
\leq & 2\mathbb{E}\left[\left\|(\nabla f_i(Y_k) - \nabla f_i(Y^{(s)})) - \left(\nabla F(Y_k) - \nabla F(Y^{(s)})\right)\right\|^2\right] \\
& + 2\mathbb{E}[\|\nabla F(Y_k)\|^2] \\
\leq & 2\mathbb{E}\left[\left\|\nabla f_i(Y_k) - \nabla f_i(Y^{(s)})\right\|^2\right] + 2\mathbb{E}[\|\nabla F(Y_k)\|^2] \\
\leq & 2L^2\mathbb{E}\left[\left\|Y_k - Y^{(s)}\right\|^2\right] + \frac{8L^2}{\alpha}H_\nu(\rho_k) + 4dL/\gamma \\
= & 2L^2\mathbb{E}\left[\left\|\sum_{i=1}^{r}(Y_{sm+i} - Y_{sm+i-1})\right\|^2\right] \\
& + \frac{8L^2}{\alpha}H_\nu(\rho_k) + 4dL/\gamma \\
= & 2L^2\mathbb{E}\left[\left\|\sum_{i=1}^{r}\left(-\eta v_{sm+i-1}^{(Y)} + \sqrt{\frac{2\eta}{\gamma}}\epsilon_{sm+i-1}\right)\right\|^2\right] \\
& + \frac{8L^2}{\alpha}H_\nu(\rho_k) + 4dL/\gamma \\
\leq & 4L^2\mathbb{E}\left[\left\|\sum_{i=1}^{r}\eta v_{sm+i-1}^{(Y)}\right\|^2\right] + 4L^2\mathbb{E}\left[\left\|\sum_{i=1}^{r}\left(\sqrt{\frac{2\eta}{\gamma}}\epsilon_{sm+i-1}\right)\right\|^2\right] \\
& + \frac{8L^2}{\alpha}H_\nu(\rho_k) + 4dL/\gamma \\
\leq & 4r\eta^2 L^2\sum_{i=1}^{r}\mathbb{E}[\|v_{sm+i-1}^{(Y)}\|^2] + \frac{8\eta L^2}{\gamma}\sum_{i=1}^{r}\mathbb{E}[\|\epsilon_{sm+i-1}\|^2] \\
& + \frac{8L^2}{\alpha}H_\nu(\rho_k) + 4dL/\gamma \\
\leq & 4m\eta^2 L^2\sum_{i=1}^{r}\mathbb{E}[\|v_{sm+i-1}^{(Y)}\|^2] + \frac{8\eta m L^2 d}{\gamma} + \frac{8L^2}{\alpha}H_\nu(\rho_k) + 4dL/\gamma.
\end{aligned}
$$

We used $\mathbb{E}[\|y - \mathbb{E}[y]\|^2] \leq \mathbb{E}[\|y\|^2]$ for the second inequality, smoothness of $F$ and equation (5) for the third inequality and $r < m$ for the last inequality.

Plugging these to equation (4), we conclude

$$
\begin{aligned}
\mathbb{E}_{Y_k,I_k,Y^{(s)}}[\|v_k^{(Y)}\|^2] \leq & \left(1 + \frac{2(n-B)}{B(n-1)}\right)\left(\frac{4L^2}{\alpha}H_\nu(\rho_k) + \frac{2dL}{\gamma}\right) \\
& + \frac{(n-B)}{B(n-1)}\left(4m\eta^2 L^2\sum_{i=1}^{r}\mathbb{E}[\|v_{sm+i-1}^{(Y)}\|^2] + \frac{8\eta m L^2 d}{\gamma}\right).
\end{aligned}
$$

Therefore, setting

$$
\Lambda' = \frac{4L^2}{\alpha}\left(1 + \frac{2(n-B)}{B(n-1)}\right),
$$

$$
S = 4L^2 m\eta^2\frac{(n-B)}{B(n-1)},
$$

and

$$
T = \frac{2dL}{\gamma}\left(1 + \frac{2(n-B)}{B(n-1)}\right) + \frac{8\eta m d L^2}{\gamma}\frac{(n-B)}{B(n-1)},
$$

we can rearrange this so that

$$\mathbb{E}_{Y_k,I_k,Y^{(s)}}[\|v_k^{(Y)}\|^2] \leq \sum_{i=1}^{r} S\mathbb{E}[\|v_{sm+i-1}^{(Y)}\|^2] + \Lambda' H_\nu(\rho_{sm+r}) + T. \tag{6}$$

Now, we are ready to prove by mathematical induction that the inequality of the statement holds for all $r = 0, \ldots, m-1$. When $r = 0$, the inequality holds from equation (5) as follows:

$$\begin{aligned}
\mathbb{E}_{Y_k,I_k,Y^{(s)}}[\|v_k^{(Y)}\|^2] &= \mathbb{E}_{Y_{sm}}[\|v_{sm}^{(Y)}\|^2] \\
&\leq \mathbb{E}_{Y_{sm}}[\|\nabla F(Y_{sm})\|^2] + \frac{n-B}{B(n-1)}\mathbb{E}_{Y_{sm}}[\|v_i^{(1)}(Y_{sm})\|^2] \\
&= \left(1 + \frac{n-B}{B(n-1)}\right)\mathbb{E}_{Y_{sm}}[\|\nabla F(Y_{sm})\|^2] \\
&\leq \left(1 + \frac{n-B}{B(n-1)}\right)\left(\frac{4L^2}{\alpha}H_\nu(\rho_{sm}) + \frac{2dL}{\gamma}\right) \\
&\leq \Lambda' H_\nu(\rho_{sm}) + T,
\end{aligned}$$

where for the second equality we used $v_i^{(1)}(Y_{sm}) = \nabla F(Y_{sm})$.

Next, let us assume that the inequality of the lemma holds for $r \leq l$. Then, from equation (6), we obtain

$$\begin{aligned}
\mathbb{E}[\|v_{sm+l+1}^{(Y)}\|^2] &\leq \sum_{i=0}^{l} S\mathbb{E}[\|v_{sm+i}^{(Y)}\|^2] + \Lambda' H_\nu(\rho_{sm+l+1}) + T \\
&\leq \sum_{i=0}^{l} S\left(\Lambda' H_\nu(\rho_{sm+i}) + T + \sum_{j=0}^{i-1} S(S+1)^{i-j-1}\left(\Lambda' H_\nu(\rho_{sm+j}) + T\right)\right) \\
&\quad + \Lambda' H_\nu(\rho_{sm+l+1}) + T \\
&= \sum_{i=0}^{l} S\left(\Lambda' H_\nu(\rho_{sm+i}) + T\right)\left(1 + \sum_{j=0}^{l-i-1} S(S+1)^j\right) \\
&\quad + \Lambda' H_\nu(\rho_{sm+l+1}) + T \\
&= \sum_{i=0}^{l} S\left(\Lambda' H_\nu(\rho_{sm+i}) + T\right)\left(1 + S\frac{(S+1)^{l-i} - 1}{(S+1) - 1}\right) \\
&\quad + \Lambda' H_\nu(\rho_{sm+l+1}) + T \\
&= \Lambda' H_\nu(\rho_{sm+l+1}) + T + \sum_{i=0}^{l} S(S+1)^{l+1-i-1}\left(\Lambda' H_\nu(\rho_{sm+i}) + T\right).
\end{aligned}$$

In the second inequality, we used the hypothesis of mathematical induction. This is equivalent to using Gronwall's lemma. This concludes the proof.

$$Q.E.D$$

**Lemma A.4.** *Under Assumption 1, for all $k = sm + r$, where $s \in \mathbb{N} \cup \{0\}$ and $r = 0, \ldots, m-1$, the following holds in the update of SVRG-LD:*

$$\mathbb{E}_{Y_k,I_k,Y^{(s)}}[\|v_k^{(Y)} - \nabla F(Y_k)\|^2] \leq \frac{L^2(n-B)}{B(n-1)}\mathbb{E}_{Y_k,I_k,Y^{(s)}}[\|Y_k - Y^{(s)}\|^2].$$

*Proof.* Let $v_i^{(2)}(Y_k) = \nabla f_i(Y_k) - \nabla f_i(Y^{(s)}) + \nabla F(Y^{(s)}) - \nabla F(Y_k)$. Then,

$$\mathbb{E}_{Y_k,I_k,Y^{(s)}}[\|v_k^{(Y)} - \nabla F(Y_k)\|^2] = \mathbb{E}_{Y_k,I_k,Y^{(s)}}\left[\left\|\frac{1}{B}\sum_{i\in I_k} v_i^{(2)}(Y_k)\right\|^2\right]$$

$$= \frac{1}{B^2}\mathbb{E}_{Y_k,I_k,Y^{(s)}}\left[\sum_{i\neq i',\{i,i'\}\in I_k}\left\langle v_i^{(2)}(Y_k), v_{i'}^{(2)}(Y_k)\right\rangle\right]$$

$$+ \frac{1}{B^2}\mathbb{E}_{Y_k,I_k,Y^{(s)}}\left[\sum_{i\in I_k}\|v_i^{(2)}(Y_k)\|^2\right]$$

$$= \frac{B-1}{Bn(n-1)}\mathbb{E}_{Y_k,Y^{(s)}}\left[\sum_{i\neq i'}\left\langle v_i^{(2)}(Y_k), v_{i'}^{(2)}(Y_k)\right\rangle\right]$$

$$+ \frac{1}{B}\mathbb{E}_{Y_k,i,Y^{(s)}}\left[\|v_i^{(2)}(Y_k)\|^2\right]$$

($i$ follows the uniform distribution under $\{1,\dots,n\}$)

$$= \frac{B-1}{Bn(n-1)}\mathbb{E}_{Y_k,Y^{(s)}}\left[\sum_{i,i'}\left\langle v_i^{(2)}(Y_k), v_{i'}^{(2)}(Y_k)\right\rangle\right]$$

$$- \frac{B-1}{B(n-1)}\mathbb{E}_{Y_k,i,Y^{(s)}}\left[\|v_i^{(2)}(Y_k)\|^2\right]$$

$$+ \frac{1}{B}\mathbb{E}_{Y_k,i,Y^{(s)}}\left[\|v_i^{(2)}(Y_k)\|^2\right]$$

$$= \frac{n-B}{B(n-1)}\mathbb{E}_{Y_k,i,Y^{(s)}}[\|v_i^{(2)}(Y_k)\|^2].$$

In the last equality, we used $\frac{1}{n}\sum_{i=1}^n v_i^{(2)}(Y_k) = 0$.

Now, since

$$\mathbb{E}_{Y_k,i,Y^{(s)}}[\|v_i^{(2)}(Y_k)\|^2] = \mathbb{E}_{Y_k,i,Y^{(s)}}[\|\nabla f_i(Y_k) - \nabla f_i(Y^{(s)}) + \nabla F(Y^{(s)}) - \nabla F(Y_k)\|^2]$$

$$= \mathbb{E}[\|\nabla f_i(Y_k) - \nabla f_i(Y^{(s)}) - \mathbb{E}[\nabla f_i(Y_k) - \nabla f_i(Y^{(s)})]\|^2]$$

$$\leq \mathbb{E}[\|\nabla f_i(Y_k) - \nabla f_i(Y^{(s)})\|^2]$$

$$\leq L^2\mathbb{E}[\|Y_k - Y^{(s)}\|^2],$$

we obtain the desired result.

*Q.E.D*

## A.2 Main Proof

We are now ready to prove the main results. The main idea of the following proofs is due to Vempala and Wibisono (2019). We first evaluate how $H_\nu(\rho_k)$ decreases compared with the previous steps.

**Theorem A.1.** *Under Assumptions 1 and 2, $0 < \eta < \frac{\alpha}{16\sqrt{6}L^2m\gamma}$, $\gamma \geq 1$ and $B \geq m$, for all $k = sm + r$, where $s \in \mathbb{N}\cup\{0\}$ and $r = 0,\dots,m-1$, the following holds in the update of SVRG-LD:*

$$H_\nu(\rho_{k+1}) \leq e^{-\frac{3\alpha}{2\gamma}\eta}\left(1 + \frac{\alpha}{4\gamma}\eta\right)H_\nu(\rho_{sm+r}) + e^{-\frac{3\alpha}{2\gamma}\eta}\sum_{i=0}^{r-1}\frac{\alpha}{4m\gamma}\eta e^{-\frac{\alpha m}{\gamma}\eta}H_\nu(\rho_{sm+i})$$

$$+ 8\eta^2 dL^2\Upsilon,$$

*where $\Lambda = \left(1 + \frac{2(n-B)}{B(n-1)}\right)$, $\Xi = \frac{(n-B)}{B(n-1)}$ and $\Upsilon = (\Lambda + \Xi + 1 + 2m\Xi)$.*

*Proof.* Note that from Lemma A.2, Talagrand's inequality is satisfied with constant $\alpha$.

One step of SVRG-LD can be formulated as follows:

$$Y_{sm+r+1} \leftarrow Y_{sm+r} - \eta v_{sm+r}^{(Y)} + \sqrt{2\eta/\gamma}\epsilon_{sm+r}.$$

This can be further interpreted as the output at time $t = \eta$ of the following SDE:

$$d\tilde{Y}_t = -v_{sm+r}^{(Y)}dt + \sqrt{2/\gamma}dB_t, \ \tilde{Y}_0 = Y_{sm+r}. \tag{7}$$

In this context, the distribution $\tilde{\rho}_t$ of $\tilde{Y}_t$ depends on both $Y_{sm+r}$ and

$$\beta_{sm+r}^{(Y)} := (I_{sm+r}, Y^{(s)}).$$

Let us define their joint distribution $\tilde{\rho}_{rt\beta_{sm+r}^{(Y)}}$ as follows:

$$\begin{aligned}
d\tilde{\rho}_{rt\beta_{sm+r}^{(Y)}}(Y_{sm+r}, \tilde{Y}_t, \beta_{sm+r}^{(Y)}) &= d\tilde{\rho}_{r\beta_{sm+r}^{(Y)}}(Y_{sm+r}, \beta_{sm+r}^{(Y)})d\tilde{\rho}_{t|r\beta_{sm+r}^{(Y)}}(\tilde{Y}_t|Y_{sm+r}, \beta_{sm+r}^{(Y)}) \\
&= d\tilde{\rho}_{t\beta_{sm+r}^{(Y)}}(\tilde{Y}_t, \beta_{sm+r}^{(Y)})d\tilde{\rho}_{r|t\beta_{sm+r}^{(Y)}}(Y_{sm+r}|\tilde{Y}_t, \beta_{sm+r}^{(Y)}).
\end{aligned}$$

Then, the Fokker-Planck equation (2) when $Y_{sm+r}$ and $\beta_{sm+r}^{(Y)}$ are fixed becomes

$$\begin{aligned}
\frac{\partial\tilde{\rho}_{t|r\beta_{sm+r}^{(Y)}}(\tilde{Y}_t|Y_{sm+r}, \beta_{sm+r}^{(Y)})}{\partial t} &= \nabla \cdot (\tilde{\rho}_{t|r\beta_{sm+r}^{(Y)}}(\tilde{Y}_t|Y_{sm+r}, \beta_{sm+r}^{(Y)})v_{sm+r}^{(Y)}) \\
&\quad + \frac{1}{\gamma}\Delta\tilde{\rho}_{t|r\beta_{sm+r}^{(Y)}}(\tilde{Y}_t|Y_{sm+r}, \beta_{sm+r}^{(Y)}).
\end{aligned} \tag{8}$$

Therefore, the following holds about the distribution $\tilde{\rho}_t$ of $\tilde{Y}_t$ governed by equation (7):

$$\begin{aligned}
\frac{\partial\tilde{\rho}_t(y)}{\partial t} &= \int \frac{\partial\tilde{\rho}_{t|r\beta_{sm+r}^{(Y)}}(y|Y_{sm+r}, \beta_{sm+r}^{(Y)})}{\partial t}\tilde{\rho}_{r\beta_{sm+r}^{(Y)}}(Y_{sm+r}, \beta_{sm+r}^{(Y)})dY_{sm+r}d\beta_{sm+r}^{(Y)} \\
&= \int \left(\nabla \cdot (\tilde{\rho}_{t|r\beta_{sm+r}^{(Y)}}(y|Y_{sm+r}, \beta_{sm+r}^{(Y)})v_{sm+r}^{(Y)}) + \frac{1}{\gamma}\Delta\tilde{\rho}_{t|r\beta_{sm+r}^{(Y)}}(y|Y_{sm+r}, \beta_{sm+r}^{(Y)})\right) \\
&\qquad\qquad\qquad\qquad\qquad \cdot \tilde{\rho}_{r\beta_{sm+r}^{(Y)}}(Y_{sm+r}, \beta_{sm+r}^{(Y)})dY_{sm+r}d\beta_{sm+r}^{(Y)} \\
&= \int \nabla \cdot (\tilde{\rho}_{rt\beta_{sm+r}^{(Y)}}(Y_{sm+r}, y, \beta_{sm+r}^{(Y)})v_{sm+r}^{(Y)})dY_{sm+r}d\beta_{sm+r}^{(Y)} \\
&\quad + \int \frac{1}{\gamma}\Delta\tilde{\rho}_{rt\beta_{sm+r}^{(Y)}}(Y_{sm+r}, y, \beta_{sm+r}^{(Y)})dY_{sm+r}d\beta_{sm+r}^{(Y)} \\
&= \nabla \cdot \left(\tilde{\rho}_t(y)\int \tilde{\rho}_{r\beta_{sm+r}^{(Y)}|t}v_{sm+r}^{(Y)}dY_{sm+r}d\beta_{sm+r}^{(Y)}\right) + \frac{1}{\gamma}\Delta\tilde{\rho}_t(y) \\
&= \nabla \cdot \left(\tilde{\rho}_t(y)\mathbb{E}_{\tilde{\rho}_{r\beta_{sm+r}^{(Y)}|t}}[v_{sm+r}^{(Y)}|\tilde{Y}_t = y]\right) + \frac{1}{\gamma}\Delta\tilde{\rho}_t(y),
\end{aligned}$$

where for the second equation we used equation (8).

Plugging this to

$$\frac{d}{dt}H_\nu(\tilde{\rho}_t) = \frac{d}{dt}\int_{\mathbb{R}^n}\tilde{\rho}_t\log\frac{\tilde{\rho}_t}{\nu}dy = \int_{\mathbb{R}^n}\frac{\partial\tilde{\rho}_t}{\partial t}\log\frac{\tilde{\rho}_t}{\nu}dy,$$

we obtain

$$
\begin{aligned}
\frac{\mathrm{d}}{\mathrm{d}t} H_\nu(\tilde{\rho}_t) &= \int_{\mathbb{R}^d} \left( \nabla \cdot \left( \tilde{\rho}_t(y) \mathbb{E}_{\tilde{\rho}_{r\beta^{(\mathrm{Y})}_{sm+r}|t}} [v^{(\mathrm{Y})}_{sm+r}|\tilde{Y}_t = y] \right) + \frac{1}{\gamma} \Delta \tilde{\rho}_t(y) \right) \log \frac{\tilde{\rho}_t}{\nu} \mathrm{d}y \\
&= \int \left( \nabla \cdot \left( \tilde{\rho}_t(y) \left( \frac{1}{\gamma} \nabla \log \frac{\tilde{\rho}_t(y)}{\nu(y)} + \mathbb{E}_{\tilde{\rho}_{r\beta^{(\mathrm{Y})}_{sm+r}|t}} [v^{(\mathrm{Y})}_{sm+r}|\tilde{Y}_t = y] - \nabla F(y) \right) \right) \right) \\
&\qquad\qquad\qquad\qquad\qquad\qquad \cdot \log \frac{\tilde{\rho}_t(y)}{\nu(y)} \mathrm{d}y \\
&= -\int \tilde{\rho}_t(y) \left\langle \frac{1}{\gamma} \nabla \log \frac{\tilde{\rho}_t(y)}{\nu(y)} + \mathbb{E}_{\tilde{\rho}_{r\beta^{(\mathrm{Y})}_{sm+r}|t}} [v^{(\mathrm{Y})}_{sm+r}|\tilde{Y}_t = y] - \nabla F, \nabla \log \frac{\tilde{\rho}_t}{\nu} \right\rangle \mathrm{d}y \\
&= -\int \tilde{\rho}_t(y) \frac{1}{\gamma} \left\| \log \frac{\tilde{\rho}_t(y)}{\nu(y)} \right\|^2 \mathrm{d}y \\
&\quad + \int \tilde{\rho}_t(y) \left\langle \nabla F(y) - \mathbb{E}_{\tilde{\rho}_{r\beta^{(\mathrm{Y})}_{sm+r}|t}} [v^{(\mathrm{Y})}_{sm+r}|\tilde{Y}_t = y], \nabla \log \frac{\tilde{\rho}_t(y)}{\nu(y)} \right\rangle \mathrm{d}y \\
&= -\frac{1}{\gamma} J_\nu(\tilde{\rho}_t) \\
&\quad + \int \tilde{\rho}_{rt\beta^{(\mathrm{Y})}_{sm+r}} \left\langle \nabla F - v^{(\mathrm{Y})}_{sm+r}, \nabla \log \frac{\tilde{\rho}_t}{\nu} \right\rangle \mathrm{d}Y_{sm+r} \mathrm{d}y \mathrm{d}\beta^{(\mathrm{Y})}_{sm+r} \\
&= -\frac{1}{\gamma} J_\nu(\tilde{\rho}_t) + \mathbb{E}_{\tilde{\rho}_{rt\beta^{(\mathrm{Y})}_{sm+r}}} \left[ \left\langle \nabla F(\tilde{Y}_t) - v^{(\mathrm{Y})}_{sm+r}, \nabla \log \frac{\tilde{\rho}_t(\tilde{Y}_t)}{\nu(\tilde{Y}_t)} \right\rangle \right].
\end{aligned}
$$

Now, let us define the second term of the right-hand side of the very last equality as ⒜. Applying $\langle a, b \rangle \leq \gamma \|a\|^2 + \frac{1}{4\gamma} \|b\|^2$ to this, we obtain

$$
\begin{aligned}
⒜ &\leq \gamma \mathbb{E}_{\tilde{\rho}_{rt\beta^{(\mathrm{Y})}_{sm+r}}} \left[ \|\nabla F(\tilde{Y}_t) - v^{(\mathrm{Y})}_{sm+r}\|^2 \right] + \frac{1}{4\gamma} \mathbb{E}_{\tilde{\rho}_{rt\beta^{(\mathrm{Y})}_{sm+r}}} \left[ \left\| \nabla \log \frac{\tilde{\rho}_t(\tilde{Y}_t)}{\nu(\tilde{Y}_t)} \right\|^2 \right] \\
&\leq 2\gamma \mathbb{E}_{\tilde{\rho}_{rt\beta^{(\mathrm{Y})}_{sm+r}}} \left[ \|\nabla F(\tilde{Y}_t) - \nabla F(Y_{sm+r})\|^2 \right] \\
&\quad + 2\gamma \mathbb{E}_{\tilde{\rho}_{rt\beta^{(\mathrm{Y})}_{sm+r}}} \left[ \|\nabla F(Y_{sm+r}) - v^{(\mathrm{Y})}_{sm+r}\|^2 \right] \\
&\quad + \frac{1}{4\gamma} J_\nu(\tilde{\rho}_t) \\
&\leq 2\gamma L^2 \mathbb{E}_{\tilde{\rho}_{rt\beta^{(\mathrm{Y})}_{sm+r}}} \left[ \|\tilde{Y}_t - Y_{sm+r}\|^2 \right] + \frac{2\gamma L^2 (n - B)}{B(n-1)} \mathbb{E}_{\tilde{\rho}_{rt\beta^{(\mathrm{Y})}_{sm+r}}} \left[ \|Y_{sm+r} - Y_{sm}\|^2 \right] \\
&\quad + \frac{1}{4\gamma} J_\nu(\tilde{\rho}_t),
\end{aligned}
$$

where for the last inequality we used the smoothness of $F$ and Lemma A.4.

As $\tilde{Y}_t = Y_{sm+r} - t v^{(\mathrm{Y})}_{sm+r} + \sqrt{2t/\gamma} \epsilon_{sm+r}$ ($\epsilon_{sm+r} \sim N(0, I)$), from Lemma A.3, we have

$$
\begin{aligned}
\mathbb{E}[\|\tilde{Y}_t - Y_{sm+r}\|^2] &= \mathbb{E}[\| - t v^{(\mathrm{Y})}_{sm+r} + \sqrt{2t/\gamma} \epsilon_{sm+r}\|^2] \\
&= t^2 \mathbb{E}[\|v^{(\mathrm{Y})}_{sm+r}\|^2] + 2td/\gamma \\
&\leq t^2 \left( \Lambda' H_\nu(\rho_{sm+r}) + T + \sum_{i=0}^{r-1} S(S+1)^{r-i-1} \left( \Lambda' H_\nu(\rho_{sm+i}) + T \right) \right) \\
&\quad + 2td/\gamma.
\end{aligned}
$$

Furthermore, by the proof of Lemma A.3 we know that the following holds:

$$\mathbb{E}\left[\|Y_{sm+r} - Y_{sm}\|^2\right] \leq 2m\eta^2 \sum_{i=0}^{r-1} \mathbb{E}[\|v_{sm+i}^{(Y)}\|^2] + 4\eta md/\gamma$$

$$\leq 2m\eta^2 \sum_{i=0}^{r-1} (S+1)^{r-i-1}(\Lambda' H_\nu(\rho_{sm+i}) + T) + 4\eta md/\gamma.$$

As a result, taking into account that we are only concerned about the time interval $0 \leq t \leq \eta$, applying $t \leq \eta$, we conclude

$$\text{Ⓐ} \leq 2\gamma L^2 \eta^2 \left(\Lambda' H_\nu(\rho_{sm+r}) + T + \sum_{i=0}^{r-1} S(S+1)^{r-i-1}\left(\Lambda' H_\nu(\rho_{sm+i}) + T\right)\right)$$

$$+ 4\eta dL^2 + 4\gamma L^2 \eta^2 m\Xi \sum_{i=0}^{r-1} (S+1)^{r-i-1}(\Lambda' H_\nu(\rho_{sm+i}) + T) + 8\eta mdL^2\Xi$$

$$+ \frac{1}{4\gamma} J_\nu(\rho_t)$$

$$\leq 2\gamma L^2 \eta^2 \Lambda' H_\nu(\rho_{sm+r}) + \sum_{i=0}^{r-1} 4\gamma L^2 \eta^2 (S+1)^{r-i}\Lambda' H_\nu(\rho_{sm+i})$$

$$+ 4\gamma L^2 \eta^2 \sum_{i=0}^{r} (S+1)^{r-i}T + 4\eta dL^2(1 + 2m\Xi) + \frac{1}{4\gamma} J_\nu(\rho_t)$$

$$\leq 2\gamma L^2 \eta^2 \Lambda' H_\nu(\rho_{sm+r}) + \sum_{i=0}^{r-1} 4\gamma L^2 \eta^2 (S+1)^{r}\Lambda' H_\nu(\rho_{sm+i})$$

$$+ 4\gamma L^2 \eta^2 \sum_{i=0}^{r} (S+1)^{r}T + 4\eta dL^2(1 + 2m\Xi) + \frac{1}{4\gamma} J_\nu(\rho_t)$$

$$\leq 2\gamma L^2 \eta^2 \Lambda' H_\nu(\rho_{sm+r}) + \sum_{i=0}^{r-1} 4\gamma L^2 \eta^2 (S+1)^{m}\Lambda' H_\nu(\rho_{sm+i})$$

$$+ 4\gamma L^2 \eta^2 m(S+1)^{m}T + 4\eta dL^2(1 + 2m\Xi) + \frac{1}{4\gamma} J_\nu(\rho_t),$$

where for the second inequality we used $m\Xi \leq 1$ and for the last inequality $r < m$. Here, as $\Xi \leq 1$ and $\eta \leq \frac{1}{4mL}$ by $\alpha \leq \gamma L$,

$$(S+1)^m \leq e^{Sm} = e^{4L^2 m^2 \eta^2 \Xi} \leq e^{1/4} \leq 2.$$

Therefore,

$$
\text{Ⓐ} \leq 2\gamma L^2\eta^2\Lambda' H_\nu(\rho_{sm+r}) + \sum_{i=0}^{r-1} 8\gamma L^2\eta^2\Lambda' H_\nu(\rho_{sm+i})
$$
$$
+ 8\gamma L^2\eta^2 mT + 4\eta dL^2(1+2m\Xi) + \frac{1}{4\gamma}J_\nu(\rho_t)
$$
$$
= \frac{8\gamma L^4\eta^2}{\alpha}\Lambda H_\nu(\rho_{sm+r}) + \sum_{i=0}^{r-1} \frac{32\gamma L^4\eta^2}{\alpha}\Lambda H_\nu(\rho_{sm+i})
$$
$$
+ 8\gamma L^2\eta^2 m\left(\frac{2dL}{\gamma}\Lambda + \frac{8\eta mdL^2}{\gamma}\Xi\right) + 4\eta dL^2(1+2m\Xi) + \frac{1}{4\gamma}J_\nu(\rho_t)
$$
$$
= \frac{8\gamma L^4\eta^2}{\alpha}\Lambda H_\nu(\rho_{sm+r}) + \sum_{i=0}^{r-1} \frac{32\gamma L^4\eta^2}{\alpha}\Lambda H_\nu(\rho_{sm+i})
$$
$$
+ 4\eta dL^2\left(4\eta mL\Lambda + 16\eta^2 m^2 L^2\Xi + 1 + 2m\Xi\right) + \frac{1}{4\gamma}J_\nu(\rho_t)
$$
$$
\leq \frac{8\gamma L^4\eta^2}{\alpha}\Lambda H_\nu(\rho_{sm+r}) + \sum_{i=0}^{r-1} \frac{32\gamma L^4\eta^2}{\alpha}\Lambda H_\nu(\rho_{sm+i})
$$
$$
+ 4\eta dL^2\left(\Lambda + \Xi + 1 + 2m\Xi\right) + \frac{1}{4\gamma}J_\nu(\rho_t),
$$

where for the first equality, we used $\Lambda' = \frac{4L^2}{\alpha}\Lambda$ and $T = \left(\frac{2dL}{\gamma}\Lambda + \frac{8\eta mdL^2}{\gamma}\Xi\right)$, and for the last inequality $\eta \leq \frac{1}{4mL}$. Thus, setting $\Upsilon = \Lambda + \Xi + 1 + 2m\Xi$, we obtain

$$
\frac{\mathrm{d}}{\mathrm{d}t}H_\nu(\tilde{\rho}_t) \leq -\frac{3}{4\gamma}J_\nu(\tilde{\rho}_t) + \frac{8\gamma L^4\eta^2}{\alpha}\Lambda H_\nu(\rho_{sm+r}) + \sum_{i=0}^{r-1}\frac{32\gamma L^4\eta^2}{\alpha}\Lambda H_\nu(\rho_{sm+i})
$$
$$
+ 4\eta dL^2\Upsilon.
$$

According to Assumption 2,

$$
\frac{\mathrm{d}}{\mathrm{d}t}H_\nu(\tilde{\rho}_t) \leq -\frac{3\alpha}{2\gamma}H_\nu(\tilde{\rho}_t) + \frac{8\gamma L^4\eta^2}{\alpha}\Lambda H_\nu(\rho_{sm+r}) + \sum_{i=0}^{r-1}\frac{32\gamma L^4\eta^2}{\alpha}\Lambda H_\nu(\rho_{sm+i})
$$
$$
+ 4\eta dL^2\Upsilon.
$$

Grouping the second to fourth terms as $U_{sm+r}^{(\Upsilon)}$ and multiplying both sides by $\mathrm{e}^{\frac{3\alpha}{2\gamma}t}$, we can write the above equation as

$$
\frac{\mathrm{d}}{\mathrm{d}t}\left(\mathrm{e}^{\frac{3\alpha}{2\gamma}t}H_\nu(\tilde{\rho}_t)\right) \leq \mathrm{e}^{\frac{3\alpha}{2\gamma}t}U_{sm+r}^{(\Upsilon)}.
$$

Integrating both sides from $t = 0$ to $t = \eta$ and using $\tilde{\rho}_\eta = \rho_{sm+r+1}$, we obtain

$$
\mathrm{e}^{\frac{3\alpha}{2\gamma}\eta}H_\nu(\rho_{sm+r+1}) - H_\nu(\rho_{sm+r}) \leq \frac{2\gamma(\mathrm{e}^{\frac{3\alpha}{2\gamma}\eta}-1)}{3\alpha}U_{sm+r}^{(\Upsilon)}
$$
$$
\leq 2\eta U_{sm+r}^{(\Upsilon)}.
$$

Here, for the last inequality, we used that $\mathrm{e}^c \leq 1+2c$ $(0 < c = \frac{3\alpha}{2\gamma}\eta \leq 1)$ holds since $0 < \eta \leq \frac{\alpha}{16\sqrt{6}L^2 m\gamma} \leq \frac{2\gamma}{3\alpha}$, where we used $1/L \leq \gamma/\alpha$ and $m \geq 1$. Rearranging this, we obtain

$$
H_\nu(\rho_{sm+r+1}) \leq \mathrm{e}^{-\frac{3\alpha}{2\gamma}\eta}\left(1 + \frac{16\gamma L^4\eta^3}{\alpha}\Lambda\right)H_\nu(\rho_{sm+r}) + \mathrm{e}^{-\frac{3\alpha}{2\gamma}\eta}\sum_{i=0}^{r-1}\frac{64\gamma L^4\eta^3}{\alpha}\Lambda H_\nu(\rho_{sm+i})
$$
$$
+ \mathrm{e}^{-\frac{3\alpha}{2\gamma}\eta}8\eta^2 dL^2\Upsilon. \tag{9}
$$

Furthermore, since $\eta \le \frac{\alpha}{16\sqrt{6}mL^2\gamma} \le \frac{\alpha}{8\sqrt{3}L^2\gamma}$, $e^{-\frac{3\alpha}{2\gamma}\eta} \le 1$ and $\Lambda \le 3$

$$H_\nu(\rho_{sm+r+1}) \le e^{-\frac{3\alpha}{2\gamma}\eta}\left(1 + \frac{\alpha}{4\gamma}\eta\right)H_\nu(\rho_{sm+r}) + e^{-\frac{3\alpha}{2\gamma}\eta}\sum_{i=0}^{r-1}\frac{\alpha}{8\gamma m}\eta H_\nu(\rho_{sm+i}) + 8\eta^2 dL^2\Upsilon.$$

On the other hand, since $\eta \le \frac{\alpha}{8mL^2\gamma}$ and $\alpha \le \gamma L$ holds,

$$e^{-\frac{\alpha m}{\gamma}\eta} \ge e^{-\frac{\alpha m}{\gamma}\cdot\frac{\alpha}{8mL^2\gamma}} = e^{-\frac{\alpha^2}{8L^2\gamma^2}} \ge e^{-1/8} \ge 0.88 \ge \frac{1}{2},$$

which further implies

$$H_\nu(\rho_{sm+r+1}) \le e^{-\frac{3\alpha}{2\gamma}\eta}\left(1 + \frac{\alpha}{4\gamma}\eta\right)H_\nu(\rho_{sm+r}) + e^{-\frac{3\alpha}{2\gamma}\eta}\sum_{i=0}^{r-1}\frac{\alpha}{4m\gamma}\eta e^{-\frac{\alpha m}{\gamma}\eta}H_\nu(\rho_{sm+i})$$
$$+ 8\eta^2 dL^2\Upsilon.$$

$$Q.E.D$$

Finally, let us prove Theorem 1 and Corollary 1.1.

**Theorem A.2** (Theorem 1 restated). *Under Assumptions 1 and 2, $0 < \eta < \frac{\alpha}{16\sqrt{6}L^2 m\gamma}$, $\gamma \ge 1$ and $B \ge m$, for all $k \ge 1$, the following holds in the update of SVRG-LD:*

$$H_\nu(\rho_k) \le e^{-\frac{\alpha\eta}{\gamma}k}H_\nu(\rho_0) + \frac{224\eta\gamma dL^2}{3\alpha}\Upsilon,$$

*where $\Xi = \frac{(n-B)}{B(n-1)}$ and $\Upsilon = (\Lambda + \Xi + 1 + 2m\Xi)$.*

*Proof.* Let us first prove by mathematical induction that the following inequality holds for all $k = 1, 2\ldots$:

$$H_\nu(\rho_k) \le e^{-\frac{\alpha\eta}{\gamma}k}H_\nu(\rho_0) + 8\eta^2 dL^2\Upsilon \cdot \left(1 - e^{-\frac{\alpha\eta}{\gamma}}\right)^{-1}. \qquad \ldots \ (*)$$

(I) When $k = 1$, from Theorem A.1, since $Y^{(s)} = Y_0$,

$$\begin{aligned}
H_\nu(\rho_1) &\le e^{-\frac{3\alpha}{2\gamma}\eta}\left(1 + \frac{\alpha}{4\gamma}\eta\right)H_\nu(\rho_0) + e^{-\frac{3\alpha}{2\gamma}\eta}\frac{\alpha}{4m\gamma}\eta e^{-\frac{\alpha m}{\gamma}\eta}H_\nu(\rho_0) + 8\eta^2 dL^2\Upsilon \\
&\le e^{-\frac{3\alpha}{2\gamma}\eta}\left(1 + \frac{\alpha}{4\gamma}\eta + \frac{\alpha}{4m\gamma}\eta\right)H_\nu(\rho_0) + 8\eta^2 dL^2\Upsilon \\
&\le e^{-\frac{3\alpha}{2\gamma}\eta}\left(1 + \frac{\alpha}{2\gamma}\eta\right)H_\nu(\rho_0) + 8\eta^2 dL^2\Upsilon \\
&\le e^{-\frac{3\alpha}{2\gamma}\eta}e^{\frac{\alpha}{2\gamma}\eta}H_\nu(\rho_0) + 8\eta^2 dL^2\Upsilon \\
&= e^{-\frac{\alpha}{\gamma}\eta}H_\nu(\rho_0) + 8\eta^2 dL^2\Upsilon \\
&\le e^{-\frac{\alpha}{\gamma}\eta}H_\nu(\rho_0) + 8\eta^2 dL^2\Upsilon \cdot \left(1 - e^{-\frac{\alpha\eta}{\gamma}}\right)^{-1}.
\end{aligned}$$

Here, for the second and last inequality, we used $e^{-\frac{\alpha m\eta}{\gamma}} \le e^{-\frac{\alpha\eta}{\gamma}} \le 1$. Thus, $(*)$ holds for $k = 1$.
(II) Now, let us assume that $(*)$ holds for all $k \le l$. Letting $r$ and $s$ the remainder and quotient of the Euclidian division of $l$ by $m$ respectively, when $k = l + 1$ we obtain from Theorem A.1,

$$H_\nu(\rho_{sm+r+1}) \le e^{-\frac{3\alpha}{2\gamma}\eta}\left(1 + \frac{\alpha}{4\gamma}\eta\right)H_\nu(\rho_{sm+r}) + e^{-\frac{3\alpha}{2\gamma}\eta}\sum_{i=0}^{r-1}\frac{\alpha}{4m\gamma}\eta e^{-\frac{\alpha m}{\gamma}\eta}H_\nu(\rho_{sm+i})$$
$$+ 8\eta^2 dL^2\Upsilon.$$

From the hypothesis of mathematical induction,

$$H_\nu(\rho_{sm+r+1})$$

$$\leq e^{-\frac{3\alpha}{2\gamma}\eta}\left(1+\frac{\alpha}{4\gamma}\eta\right)\left(e^{-\frac{\alpha\eta}{\gamma}(sm+r)}H_\nu(\rho_0)+8\eta^2 dL^2\Upsilon\cdot\left(1-e^{-\frac{\alpha\eta}{\gamma}}\right)^{-1}\right)$$

$$+e^{-\frac{3\alpha}{2\gamma}\eta}\sum_{i=0}^{r-1}\frac{\alpha}{4m\gamma}\eta e^{-\frac{\alpha m}{\gamma}\eta}\left(e^{-\frac{\alpha\eta}{\gamma}(sm+i)}H_\nu(\rho_0)+8\eta^2 dL^2\Upsilon\cdot\left(1-e^{-\frac{\alpha\eta}{\gamma}}\right)^{-1}\right)$$

$$+8\eta^2 dL^2\Upsilon.$$

Since $e^{-\frac{\alpha}{\gamma}\eta m}\leq e^{-\frac{\alpha}{\gamma}\eta r}\leq e^{-\frac{\alpha}{\gamma}\eta(r-i)}$ when $0\leq i < r < m$,

$$H_\nu(\rho_{sm+r+1})\leq e^{-\frac{3\alpha}{2\gamma}\eta}\left(1+\frac{\alpha}{4\gamma}\eta\right)\left(e^{-\frac{\alpha\eta}{\gamma}(sm+r)}H_\nu(\rho_0)+8\eta^2 dL^2\Upsilon\cdot\left(1-e^{-\frac{\alpha\eta}{\gamma}}\right)^{-1}\right)$$

$$+e^{-\frac{3\alpha}{2\gamma}\eta}\sum_{i=0}^{r-1}\frac{\alpha}{4m\gamma}\eta e^{-\frac{\alpha\eta}{\gamma}(r-i)}\left(e^{-\frac{\alpha\eta}{\gamma}(sm+i)}H_\nu(\rho_0)\right)$$

$$+e^{-\frac{3\alpha}{2\gamma}\eta}\sum_{i=0}^{r-1}\frac{\alpha}{4m\gamma}\eta e^{-\frac{\alpha\eta}{\gamma}(r-i)}\left(8\eta^2 dL^2\Upsilon\cdot\left(1-e^{-\frac{\alpha\eta}{\gamma}}\right)^{-1}\right)$$

$$+8\eta^2 dL^2\Upsilon$$

$$\leq e^{-\frac{3\alpha}{2\gamma}\eta}\left(1+\frac{\alpha}{4\gamma}\eta\right)\left(e^{-\frac{\alpha\eta}{\gamma}(sm+r)}H_\nu(\rho_0)+8\eta^2 dL^2\Upsilon\cdot\left(1-e^{-\frac{\alpha\eta}{\gamma}}\right)^{-1}\right)$$

$$+e^{-\frac{3\alpha}{2\gamma}\eta}\sum_{i=0}^{r-1}\frac{\alpha}{4m\gamma}\eta\left(e^{-\frac{\alpha\eta}{\gamma}(sm+r)}H_\nu(\rho_0)+8\eta^2 dL^2\Upsilon\cdot\left(1-e^{-\frac{\alpha\eta}{\gamma}}\right)^{-1}\right)$$

$$+8\eta^2 dL^2\Upsilon$$

$$\leq e^{-\frac{3\alpha}{2\gamma}\eta}\left(1+\frac{\alpha}{2\gamma}\eta\right)\left(e^{-\frac{\alpha\eta}{\gamma}(sm+r)}H_\nu(\rho_0)+8\eta^2 dL^2\Upsilon\cdot\left(1-e^{-\frac{\alpha\eta}{\gamma}}\right)^{-1}\right)$$

$$+8\eta^2 dL^2\Upsilon$$

$$\leq e^{-\frac{3\alpha}{2\gamma}\eta}e^{\frac{\alpha}{2\gamma}\eta}\left(e^{-\frac{\alpha\eta}{\gamma}(sm+r)}H_\nu(\rho_0)+8\eta^2 dL^2\Upsilon\cdot\left(1-e^{-\frac{\alpha\eta}{\gamma}}\right)^{-1}\right)$$

$$+8\eta^2 dL^2\Upsilon$$

$$= e^{-\frac{\alpha\eta}{\gamma}(sm+r+1)}H_\nu(\rho_0)+\left(1+e^{-\frac{\alpha\eta}{\gamma}}\cdot\left(1-e^{-\frac{\alpha\eta}{\gamma}}\right)^{-1}\right)8\eta^2 dL^2\Upsilon$$

$$= e^{-\frac{\alpha\eta}{\gamma}(sm+r+1)}H_\nu(\rho_0)+8\eta^2 dL^2\Upsilon\cdot\left(1-e^{-\frac{\alpha\eta}{\gamma}}\right)^{-1}.$$

Therefore, $(*)$ holds for all $k\geq 1$.

Now, using the inequality $1-e^{-c}\geq\frac{3}{4}c$ for $0<c=\frac{\alpha\eta}{\gamma}\leq\frac{1}{4}$ (since $y=1-e^{-x}$ and $y=\frac{3}{4}x$ are both concave increasing functions intersecting at $x=0$ and $1-e^{-1/4}\geq\frac{3}{4}\times\frac{1}{4}$), which holds here because $\eta\leq\frac{\alpha}{16\sqrt{6}L^2\gamma}\leq\frac{\gamma}{4\alpha}$ since $1/L\leq\gamma/\alpha$ and $m\geq 1$, we conclude

$$H_\nu(\rho_k)\leq e^{-\frac{\alpha\eta}{\gamma}k}H_\nu(\rho_0)+\frac{32\eta\gamma dL^2}{3\alpha}\Upsilon$$

$$\leq e^{-\frac{\alpha\eta}{\gamma}k}H_\nu(\rho_0)+\frac{224\eta\gamma dL^2}{3\alpha},$$

which is the desired result. Here, for the last inequality, we used $\Upsilon=\Lambda+\Xi+1+2m\Xi\leq 3+1+1+2=7$.

$$Q.E.D$$

**Corollary A.2.1** (Corollary 1.1 restated). *Under the same assumptions as Theorem A.2, for all $\epsilon\geq 0$, if we choose step size $\eta$ such that*

$$\eta\leq\frac{3\alpha\epsilon}{448\gamma dL^2},$$

*then a precision $H_\nu(\rho_k) \leq \epsilon$ is reached after*

$$k \geq \frac{\gamma}{\alpha\eta} \log \frac{2H_\nu(\rho_0)}{\epsilon}$$

*steps. Especially, if we take $B = m = \sqrt{n}$ and the largest permissible step size $\eta = \frac{\alpha}{16\sqrt{6}L^2\sqrt{n}\gamma} \wedge \frac{3\alpha\epsilon}{448dL^2\gamma}$, then the gradient complexity becomes*

$$\tilde{O}\left(\left(n + \frac{dn^{\frac{1}{2}}}{\epsilon}\right) \frac{\gamma^2 L^2}{\alpha^2}\right).$$

*Proof.* Let $\epsilon > 0$. Then, by additionally requiring

$$\eta \leq \frac{3\alpha\epsilon}{448\gamma dL^2},$$

we obtain

$$H_\nu(\rho_k) \leq e^{-\frac{\alpha\eta}{\gamma}k} H_\nu(\rho_0) + \frac{\epsilon}{2}.$$

Thus, $H_\nu(\rho_k) \leq \epsilon$ can be reached for

$$k \geq \frac{\gamma}{\alpha\eta} \log \frac{2H_\nu(\rho_0)}{\epsilon}.$$

As a result, if $0 < \epsilon \leq \frac{28d}{3\sqrt{6}m}$ and we select the largest permissible step size, the gradient complexity becomes

$$O\left(k \cdot B + \frac{k}{m} \cdot n\right) = \tilde{O}\left(\left(\frac{B + n/m}{\epsilon}\right) \frac{d\gamma^2 L^2}{\alpha^2}\right),$$

and the optimal complexity is

$$\tilde{O}\left(\frac{dn^{1/2}\gamma^2 L^2}{\epsilon\alpha^2}\right)$$

with $B = \sqrt{n}$ and $m = \sqrt{n}$.

On the other hand, if $\epsilon \geq \frac{28d}{3\sqrt{6}m}$ and we select the largest permissible step size, the gradient complexity becomes

$$O\left(k \cdot B + \frac{k}{m} \cdot n\right) = \tilde{O}\left((mB + n) \frac{\gamma^2 L^2}{\alpha^2}\right),$$

and the optimal complexity is

$$\tilde{O}\left(\frac{n\gamma^2 L^2}{\alpha^2}\right)$$

with $B = \sqrt{n}$ and $m = \sqrt{n}$

Therefore, for all $\epsilon \geq 0$, the gradient complexity is

$$\tilde{O}\left(\left(n + \frac{dn^{\frac{1}{2}}}{\epsilon}\right) \frac{\gamma^2 L^2}{\alpha^2}\right).$$

$$Q.E.D$$

# B    Proof of Theorem 2 and Corollary 2.1

In this Section, to clearly differentiate from SVRG-LD, we redefine the random variable generated at the $k$-th step of SARAH-LD (Algorithm 1) as $Z_k$ and the stochastic gradient as $v_k^{(Z)}$. The distribution of $Z_k$ is $\phi_k$.

## B.1 Preparation for the Proof

Let us first provide an upper bound of $\mathbb{E}[\|v_k^{(\mathrm{Z})}\|^2]$ and the variance of the stochastic gradient $v_k^{(\mathrm{Z})}$ using the KL-divergences $H_\nu(\phi_k), H_\nu(\phi_{k-1}), \dots$.

**Lemma B.1.** *Under Assumption 1, for all $k = sm + r$, where $s \in \mathbb{N} \cup \{0\}$ and $r = 0, \dots, m-1$, the following holds in the update of SARAH-LD:*

$$\mathbb{E}[\|\nabla F(Z_k) - v_k^{(\mathrm{Z})}\|^2] = \sum_{i=1}^r \mathbb{E}[\|v_{sm+i}^{(\mathrm{Z})} - v_{sm+i-1}^{(\mathrm{Z})}\|^2] - \sum_{i=1}^r \mathbb{E}[\|\nabla F(Z_{sm+i}) - \nabla F(Z_{sm+i-1})\|^2].$$

*Proof.* Let us define

$$\mathcal{F}_r = \sigma\left(Z^{(s)}, \epsilon_{sm}, I_{sm+1}, \epsilon_{sm+1}, I_{sm+2}, \epsilon_{sm+2}, \dots, I_{sm+r-1}, \epsilon_{sm+r-1}\right),$$

which is the $\sigma$-*algebra* generated by

$$Z^{(s)}, \epsilon_{sm}, I_{sm+1}, \epsilon_{sm+1}, I_{sm+2}, \epsilon_{sm+2}, \dots, I_{sm+r-1}, \text{ and } \epsilon_{sm+r-1}.$$

When $r = 0$, the statement clearly holds. In the remainder of the proof, we assume $r \geq 1$. Then,

$$
\begin{aligned}
\mathbb{E}[\|\nabla F(Z_k) - v_k^{(\mathrm{Z})}\|^2 \mid \mathcal{F}_r] = {} & \mathbb{E}[\|\nabla F(Z_{k-1}) - v_{k-1}^{(\mathrm{Z})} + \nabla F(Z_k) - \nabla F(Z_{k-1}) \\
& \qquad - (v_k^{(\mathrm{Z})} - v_{k-1}^{(\mathrm{Z})})\|^2 \mid \mathcal{F}_r] \\
= {} & \|\nabla F(Z_{k-1}) - v_{k-1}^{(\mathrm{Z})}\|^2 + \|\nabla F(Z_k) - \nabla F(Z_{k-1})\|^2 \\
& + \mathbb{E}[\|v_k^{(\mathrm{Z})} - v_{k-1}^{(\mathrm{Z})}\|^2 \mid \mathcal{F}_r] \\
& + 2\left\langle \nabla F(Z_{k-1}) - v_{k-1}^{(\mathrm{Z})}, \nabla F(Z_k) - \nabla F(Z_{k-1})\right\rangle \\
& - 2\left\langle \nabla F(Z_{k-1}) - v_{k-1}^{(\mathrm{Z})}, \mathbb{E}[v_k^{(\mathrm{Z})} - v_{k-1}^{(\mathrm{Z})} \mid \mathcal{F}_r]\right\rangle \\
& - 2\left\langle \nabla F(Z_k) - \nabla F(Z_{k-1}), \mathbb{E}[v_k^{(\mathrm{Z})} - v_{k-1}^{(\mathrm{Z})} \mid \mathcal{F}_r]\right\rangle \\
= {} & \|\nabla F(Z_{k-1}) - v_{k-1}^{(\mathrm{Z})}\|^2 - \|\nabla F(Z_k) - \nabla F(Z_{k-1})\|^2 \\
& + \mathbb{E}[\|v_k^{(\mathrm{Z})} - v_{k-1}^{(\mathrm{Z})}\|^2 \mid \mathcal{F}_r].
\end{aligned}
$$

Here in the last equality, we used that the following holds:

$$
\begin{aligned}
\mathbb{E}[v_k^{(\mathrm{Z})} - v_{k-1}^{(\mathrm{Z})} \mid \mathcal{F}_r] = {} & \mathbb{E}\left[\frac{1}{B}\sum_{i \in I_k} \nabla f_i(Z_k) - \nabla f_i(Z_{k-1}) \mid \mathcal{F}_r\right] \\
= {} & \nabla F(Z_k) - \nabla F(Z_{k-1}).
\end{aligned}
$$

Taking expectation, we obtain

$$
\begin{aligned}
\mathbb{E}[\|\nabla F(Z_k) - v_k^{(\mathrm{Z})}\|^2] = {} & \mathbb{E}[\|\nabla F(Z_{k-1}) - v_{k-1}^{(\mathrm{Z})}\|^2] - \mathbb{E}[\|\nabla F(Z_k) - \nabla F(Z_{k-1})\|^2] \\
& + \mathbb{E}[\|v_k^{(\mathrm{Z})} - v_{k-1}^{(\mathrm{Z})}\|^2].
\end{aligned}
$$

Since this equation holds for all $k = sm + r$ ($r = 1, \dots m-1$), recalling that

$$\mathbb{E}[\|\nabla F(Z_{sm}) - v_{sm}^{(\mathrm{Z})}\|^2] = 0,$$

and recursively applying this, we conclude that

$$\mathbb{E}[\|\nabla F(Z_k) - v_k^{(\mathrm{Z})}\|^2] = \sum_{i=1}^r \mathbb{E}[\|v_{sm+i}^{(\mathrm{Z})} - v_{sm+i-1}^{(\mathrm{Z})}\|^2] - \sum_{i=1}^r \mathbb{E}[\|\nabla F(Z_{sm+i}) - \nabla F(Z_{sm+i-1})\|^2].$$

$$Q.E.D$$

**Lemma B.2.** *Under Assumption 1, for all $k = sm + r$, where $s \in \mathbb{N} \cup \{0\}$ and $r = 0, \dots, m-1$, the following holds in the update of SARAH-LD:*

$$\mathbb{E}[\|\nabla F(Z_k) - v_k^{(\mathrm{Z})}\|^2] \leq \sum_{i=1}^r \Xi L^2 \eta^2 \mathbb{E}[\|v_{sm+i-1}^{(\mathrm{Z})}\|^2] + \frac{2\eta m d L^2}{\gamma}\Xi,$$

*where $\Xi = \frac{n-B}{B(n-1)}$.*

*Proof.* When $r = 0$, the statement clearly holds. In the remainder of the proof, we assume $r \geq 1$. Since $v_k^{(Z)} - v_{k-1}^{(Z)} = \frac{1}{B} \sum_{j \in I_k} (\nabla f_j(Z_k) - \nabla f_j(Z_{k-1}))$, defining

$$w_j := \nabla f_j(Z_k) - \nabla f_j(Z_{k-1}),$$

we obtain

$$
\begin{aligned}
\mathbb{E}[\|v_k^{(Z)} - v_{k-1}^{(Z)}\|^2 \mid \mathcal{F}_k] &= \mathbb{E}\left[\left\|\frac{1}{B}\sum_{j \in I_k} w_j\right\|^2 \mid \mathcal{F}_k\right] \\
&= \frac{1}{B^2}\mathbb{E}\left[\sum_{j \neq j', \{j,j'\} \in I_k} \langle w_j, w_{j'} \rangle \mid \mathcal{F}_k\right] + \frac{1}{B^2}\mathbb{E}\left[\sum_{j \in I_k} \|w_j\|^2 \mid \mathcal{F}_k\right] \\
&= \frac{B-1}{Bn(n-1)}\mathbb{E}\left[\sum_{j \neq j'} \langle w_j, w_{j'} \rangle \mid \mathcal{F}_k\right] + \frac{1}{B}\mathbb{E}\left[\|w_j\|^2 \mid \mathcal{F}_k\right] \\
&\quad (j \text{ follows a uniform distribution under } \{1, \ldots, n\}) \\
&= \frac{B-1}{Bn(n-1)}\mathbb{E}\left[\sum_{j,j'} \langle w_j, w_{j'} \rangle \mid \mathcal{F}_k\right] - \frac{B-1}{B(n-1)}\mathbb{E}\left[\|w_j\|^2 \mid \mathcal{F}_k\right] \\
&\quad + \frac{1}{B}\mathbb{E}\left[\|w_j\|^2 \mid \mathcal{F}_k\right] \\
&= \frac{(B-1)n}{B(n-1)}\mathbb{E}[\|\nabla F(Z_k) - \nabla F(Z_{k-1})\|^2 \mid \mathcal{F}_k] \\
&\quad + \frac{n-B}{B(n-1)}\mathbb{E}[\|\nabla f_j(Z_k) - \nabla f_j(Z_{k-1})\|^2 \mid \mathcal{F}_k] \\
&\leq \mathbb{E}[\|\nabla F(Z_k) - \nabla F(Z_{k-1})\|^2 \mid \mathcal{F}_k] \\
&\quad + \frac{n-B}{B(n-1)}\mathbb{E}[\|\nabla f_j(Z_k) - \nabla f_j(Z_{k-1})\|^2 \mid \mathcal{F}_k],
\end{aligned}
$$

where for the fifth equation we used $\frac{1}{n}\sum_{j=1}^n w_j = \nabla F(Z_k) - \nabla F(Z_{k-1})$ and for the inequality, $\frac{(B-1)n}{B(n-1)} \leq 1$.

As a result,

$$
\begin{aligned}
\mathbb{E}[\|v_k^{(Z)} - v_{k-1}^{(Z)}\|^2 \mid \mathcal{F}_k] &\leq \|\nabla F(Z_k) - \nabla F(Z_{k-1})\|^2 \\
&\quad + \frac{n-B}{B(n-1)}\mathbb{E}[\|\nabla f_j(Z_k) - \nabla f_j(Z_{k-1})\|^2 \mid \mathcal{F}_k] \\
&\leq \|\nabla F(Z_k) - \nabla F(Z_{k-1})\|^2 + L^2\Xi\mathbb{E}[\|Z_k - Z_{k-1}\|^2 \mid \mathcal{F}_k] \\
&= \|\nabla F(Z_k) - \nabla F(Z_{k-1})\|^2 + L^2\Xi\left\|-\eta v_{k-1}^{(Z)} + \sqrt{\frac{2\eta}{\gamma}}\epsilon_{k-1}\right\|^2.
\end{aligned}
$$

Taking expectation, we obtain

$$
\begin{aligned}
\mathbb{E}[\|v_k^{(Z)} - v_{k-1}^{(Z)}\|^2] - \mathbb{E}[\|\nabla F(Z_k) - \nabla F(Z_{k-1})\|^2] &\leq L^2\Xi\mathbb{E}\left[\left\|-\eta v_{k-1}^{(Z)} + \sqrt{\frac{2\eta}{\gamma}}\epsilon_{k-1}\right\|^2\right] \\
&= L^2\Xi\left(\eta^2\mathbb{E}[\|v_{k-1}^{(Z)}\|^2] + \frac{2\eta d}{\gamma}\right).
\end{aligned}
$$

Since this equation holds for all $k = sm + r$ $(r = 1, \ldots m - 1)$, from Lemma B.1,

$$
\begin{aligned}
\mathbb{E}[\|\nabla F(Z_k) - v_k^{(\mathrm{Z})}\|^2] &\leq \sum_{i=1}^{r} \mathbb{E}[\|v_{sm+i}^{(\mathrm{Z})} - v_{sm+i-1}^{(\mathrm{Z})}\|^2] \\
&\quad - \sum_{i=1}^{r} \mathbb{E}[\|\nabla F(Z_{sm+i}) - \nabla F(Z_{sm+i-1})\|^2] \\
&\leq \Xi L^2 \eta^2 \sum_{i=1}^{r} \mathbb{E}[\|v_{sm+i-1}^{(\mathrm{Z})}\|^2] + \frac{2\eta r d L^2}{\gamma} \Xi \\
&\leq \Xi L^2 \eta^2 \sum_{i=1}^{r} \mathbb{E}[\|v_{sm+i-1}^{(\mathrm{Z})}\|^2] + \frac{2\eta m d L^2}{\gamma} \Xi.
\end{aligned}
$$

$$Q.E.D$$

**Lemma B.3.** *Under Assumption 1, suppose Talagrand's inequality holds for $\nu$ with a constant $\alpha$, then for all $k = sm + r$, where $s \in \mathbb{N} \cup \{0\}$ and $r = 0, \ldots, m - 1$, the following holds in the update of SARAH-LD:*

$$
\mathbb{E}[\|v_k^{(\mathrm{Z})}\|^2] \leq \frac{8L^2}{\alpha} H_\nu(\phi_{sm+r}) + P + \sum_{i=0}^{r-1} Q(Q+1)^{r-i-1}\left(\frac{8L^2}{\alpha} H_\nu(\phi_{sm+i}) + P\right),
$$

*where*

$$
\Xi = \frac{(n - B)}{B(n - 1)},
$$

$$
P = \frac{4dL}{\gamma} + \frac{4\eta m d L^2}{\gamma}\Xi,
$$

*and*

$$
Q = 2\Xi L^2 \eta^2.
$$

*Proof.* First, from Lemma B.2, we have

$$
\begin{aligned}
\mathbb{E}[\|v_k^{(\mathrm{Z})}\|^2] &\leq 2\mathbb{E}[\|v_k^{(\mathrm{Z})} - \nabla F(Z_k)\|^2] + 2\mathbb{E}[\|\nabla F(Z_k)\|^2] \\
&\leq 2\left(\sum_{i=1}^{r} \Xi L^2 \eta^2 \mathbb{E}[\|v_{sm+i-1}^{(\mathrm{Z})}\|^2] + \frac{2\eta m d L^2}{\gamma}\Xi\right) + 2\mathbb{E}[\|\nabla F(Z_k)\|^2].
\end{aligned}
$$

Choosing an optimal coupling $Z_k \sim \phi_k$ and $Z^* \sim \nu$ so that $\mathbb{E}[\|Z_k - Z^*\|^2] = W_2(\phi_k, \nu)^2$, we obtain

$$
\begin{aligned}
\mathbb{E}_{Z_k}[\|\nabla F(Z_k)\|^2] &\leq 2\mathbb{E}_{Z_k, Z^*}[\|\nabla F(Z_k) - \nabla F(Z^*)\|^2] + 2\mathbb{E}_{Z^*}[\|\nabla F(Z^*)\|^2] \\
&\leq 2L^2 \mathbb{E}[\|Z_k - Z^*\|^2] + 2dL/\gamma \\
&= 2L^2 W_2(\phi_k, \nu)^2 + 2dL/\gamma \\
&\leq \frac{4L^2}{\alpha} H_\nu(\phi_k) + 2dL/\gamma, \tag{10}
\end{aligned}
$$

where, we used the smoothness of $F$ and Lemma A.1 for the second inequality, the definition of $W_2$ for the equality and Talagrand's inequality for the last inequality.

As a result,

$$
\begin{aligned}
\mathbb{E}[\|v_k^{(\mathrm{Z})}\|^2] &\leq 2\left(\sum_{i=1}^{r} \Xi L^2 \eta^2 \mathbb{E}[\|v_{sm+i-1}^{(\mathrm{Z})}\|^2] + \frac{2\eta m d L^2}{\gamma}\Xi\right) + \frac{8L^2}{\alpha} H_\nu(\phi_k) + 4dL/\gamma \\
&= \sum_{i=1}^{r} Q\mathbb{E}[\|v_{sm+i-1}^{(\mathrm{Z})}\|^2] + \frac{8L^2}{\alpha} H_\nu(\phi_k) + P. \tag{11}
\end{aligned}
$$

Here, we set

$$P = \frac{4dL}{\gamma} + \frac{4\eta m dL^2}{\gamma}\Xi,$$

and

$$Q = 2\Xi L^2 \eta^2.$$

Now, let us prove by mathematical induction that the inequality of the statement holds for all $r = 0, \ldots, m - 1$. When $r = 0$, the inequality holds from equation (10) as follows:

$$\mathbb{E}[\|v_{sm}^{(\mathrm{Z})}\|^2] = \mathbb{E}[\|\nabla F(Z_{sm})\|^2]$$

$$\leq \frac{4L^2}{\alpha}H_\nu(\phi_{sm}) + 2dL/\gamma$$

$$\leq \frac{8L^2}{\alpha}H_\nu(\phi_{sm}) + P.$$

Next, let us assume that the inequality of the lemma holds for $r \leq l$. Then, from equation (11), we obtain

$$\mathbb{E}[\|v_{sm+l+1}^{(\mathrm{Z})}\|^2]$$

$$\leq \sum_{i=0}^{l} Q\mathbb{E}[\|v_{sm+i}^{(\mathrm{Z})}\|^2] + \frac{8L^2}{\alpha}H_\nu(\phi_{sm+l+1}) + P$$

$$\leq \sum_{i=0}^{l} Q\left(\frac{8L^2}{\alpha}H_\nu(\phi_{sm+i}) + P + \sum_{j=0}^{i-1} Q(Q+1)^{i-j-1}\left(\frac{8L^2}{\alpha}H_\nu(\phi_{sm+j}) + P\right)\right)$$

$$+ \frac{8L^2}{\alpha}H_\nu(\phi_{sm+l+1}) + P$$

$$= \frac{8L^2}{\alpha}H_\nu(\phi_{sm+l+1}) + P + \sum_{i=0}^{l} Q\left(\frac{8L^2}{\alpha}H_\nu(\phi_{sm+i}) + P\right)\left(1 + \sum_{j=0}^{l-i-1} Q(Q+1)^j\right)$$

$$= \frac{8L^2}{\alpha}H_\nu(\phi_{sm+l+1}) + P + \sum_{i=0}^{l} Q\left(\frac{8L^2}{\alpha}H_\nu(\phi_{sm+i}) + P\right)\left(1 + Q\frac{(Q+1)^{l-i} - 1}{(Q+1) - 1}\right)$$

$$= \frac{8L^2}{\alpha}H_\nu(\phi_{sm+l+1}) + P + \sum_{i=0}^{l} Q(Q+1)^{l+1-i-1}\left(\frac{8L^2}{\alpha}H_\nu(\phi_{sm+i}) + P\right).$$

In the second inequality, we used the hypothesis of mathematical induction. This is equivalent to using Gronwall's lemma. This concludes the proof.

$$Q.E.D$$

## B.2 Main Proof

We are now ready to prove the main results. The main idea of the following proofs is due to Vempala and Wibisono (2019). We first evaluate how $H_\nu(\phi_k)$ decreases compared with the previous steps.

**Theorem B.1.** *Under Assumptions 1 and 2, $0 < \eta < \frac{\alpha}{16\sqrt{2}L^2 m\gamma}$ and $\gamma \geq 1$, for all $k = sm + r$, where $s \in \mathbb{N} \cup \{0\}$ and $r = 0, \ldots, m - 1$, the following holds in the update of SARAH-LD:*

$$H_\nu(\phi_{sm+r+1}) \leq \mathrm{e}^{-\frac{3\alpha}{2\gamma}\eta}\left(1 + \frac{\alpha}{4\gamma}\eta\right)H_\nu(\phi_{sm+r}) + \mathrm{e}^{-\frac{3\alpha}{2\gamma}\eta}\sum_{i=0}^{r-1}\frac{\alpha}{4m\gamma}\eta\mathrm{e}^{-\frac{\alpha m}{\gamma}\eta}H_\nu(\phi_{sm+i})$$

$$+ 8\eta^2 dL^2\left(2 + \Xi + 2m\Xi\right),$$

*where $\Xi = \frac{(n-B)}{B(n-1)}$.*

*Proof.* Note that from Lemma A.2, Talagrand's inequality is satisfied with constant $\alpha$.

One step of SVRG-LD can be formulated as follows:

$$Z_{sm+r+1} \leftarrow Z_{sm+r} - \eta v^{(Z)}_{sm+r} + \sqrt{2\eta/\gamma}\epsilon_{sm+r}.$$

This can be further interpreted as the output at time $t = \eta$ of the following SDE:

$$\mathrm{d}\tilde{Z}_t = -v^{(Z)}_{sm+r}\mathrm{d}t + \sqrt{2/\gamma}\mathrm{d}B_t, \ \tilde{Z}_0 = Z_{sm+r}. \tag{12}$$

In this context, the distribution $\tilde{\phi}_t$ of $\tilde{Z}_t$ depends on both $Z_{sm+r}$ and

$$\beta^{(Z)}_{sm+r} := (v^{(Z)}_{sm+r-1}, I_{sm+r}).$$

Let us define their joint distribution as follows:

$$\begin{aligned}
\mathrm{d}\tilde{\phi}_{rt\beta^{(Z)}_{sm+r}}(Z_{sm+r}, \tilde{Z}_t, \beta^{(Z)}_{sm+r}) &= \mathrm{d}\tilde{\phi}_{r\beta^{(Z)}_{sm+r}}(Z_{sm+r}, \beta^{(Z)}_{sm+r})\mathrm{d}\tilde{\phi}_{t|r\beta^{(Z)}_{sm+r}}(\tilde{Z}_t|Z_{sm+r}, \beta^{(Z)}_{sm+r}) \\
&= \mathrm{d}\tilde{\phi}_{t\beta^{(Z)}_{sm+r}}(\tilde{Z}_t, \beta^{(Z)}_{sm+r})\mathrm{d}\tilde{\phi}_{r|t\beta^{(Z)}_{sm+r}}(Z_{sm+r}|\tilde{Z}_t, \beta^{(Z)}_{sm+r}).
\end{aligned}$$

Then, the Fokker-Planck equation (2) when $Z_{sm+r}$ and $\beta^{(Z)}_{sm+r}$ are fixed becomes

$$\begin{aligned}
\frac{\partial\tilde{\phi}_{t|r\beta^{(Z)}_{sm+r}}(\tilde{Z}_t|Z_{sm+r}, \beta^{(Z)}_{sm+r})}{\partial t} &= \nabla \cdot (\tilde{\phi}_{t|r\beta^{(Z)}_{sm+r}}(\tilde{Z}_t|Z_{sm+r}, \beta^{(Z)}_{sm+r})v^{(Z)}_{sm+r}) \\
&\quad + \frac{1}{\gamma}\Delta\tilde{\phi}_{t|r\beta^{(Z)}_{sm+r}}(\tilde{Z}_t|Z_{sm+r}, \beta^{(Z)}_{sm+r}). \tag{13}
\end{aligned}$$

Therefore, the following holds about the distribution $\tilde{\phi}_t$ of $\tilde{Z}_t$ governed by equation (12),

$$\begin{aligned}
\frac{\partial\tilde{\phi}_t(z)}{\partial t} &= \int \frac{\partial\tilde{\phi}_{t|r\beta^{(Z)}_{sm+r}}(z|Z_{sm+r}, \beta^{(Z)}_{sm+r})}{\partial t}\tilde{\phi}_{r\beta^{(Z)}_{sm+r}}(Z_{sm+r}, \beta^{(Z)}_{sm+r})\mathrm{d}Z_{sm+r}\mathrm{d}\beta^{(Z)}_{sm+r} \\
&= \int \left(\nabla \cdot (\tilde{\phi}_{t|r\beta^{(Z)}_{sm+r}}(z|Z_{sm+r}, \beta^{(Z)}_{sm+r})v^{(Z)}_{sm+r}) + \frac{1}{\gamma}\Delta\tilde{\phi}_{t|r\beta^{(Z)}_{sm+r}}(z|Z_{sm+r}, \beta^{(Z)}_{sm+r})\right) \\
&\qquad\qquad\qquad\qquad \cdot \tilde{\phi}_{r\beta^{(Z)}_{sm+r}}(Z_{sm+r}, \beta^{(Z)}_{sm+r})\mathrm{d}Z_{sm+r}\mathrm{d}\beta^{(Z)}_{sm+r} \\
&= \int \nabla \cdot (\tilde{\phi}_{rt\beta^{(Z)}_{sm+r}}(Z_{sm+r}, z, \beta^{(Z)}_{sm+r})v^{(Z)}_{sm+r})\mathrm{d}Z_{sm+r}\mathrm{d}\beta^{(Z)}_{sm+r} \\
&\quad + \int \frac{1}{\gamma}\Delta\tilde{\phi}_{rt\beta^{(Z)}_{sm+r}}(Z_{sm+r}, z, \beta^{(Z)}_{sm+r})\mathrm{d}Z_{sm+r}\mathrm{d}\beta^{(Z)}_{sm+r} \\
&= \nabla \cdot \left(\tilde{\phi}_t(z)\int \tilde{\phi}_{r\beta^{(Z)}_{sm+r}|t}v^{(Z)}_{sm+r}\mathrm{d}Z_{sm+r}\mathrm{d}\beta^{(Z)}_{sm+r}\right) + \frac{1}{\gamma}\Delta\tilde{\phi}_t(z) \\
&= \nabla \cdot \left(\tilde{\phi}_t(z)\mathbb{E}_{\tilde{\phi}_{r\beta^{(Z)}_{sm+r}|t}}[v^{(Z)}_{sm+r}|\tilde{Z}_t = z]\right) + \frac{1}{\gamma}\Delta\tilde{\phi}_t(z),
\end{aligned}$$

where for the second equation we used equation (13).

Plugging this to

$$\frac{\mathrm{d}}{\mathrm{d}t}H_\nu(\tilde{\phi}_t) = \frac{\mathrm{d}}{\mathrm{d}t}\int_{\mathbb{R}^n}\tilde{\phi}_t\log\frac{\tilde{\phi}_t}{\nu}\mathrm{d}z = \int_{\mathbb{R}^n}\frac{\partial\tilde{\phi}_t}{\partial t}\log\frac{\tilde{\phi}_t}{\nu}\mathrm{d}z,$$

we obtain

$$
\begin{aligned}
\frac{\mathrm{d}}{\mathrm{d}t} H_\nu(\tilde{\phi}_t) &= \int_{\mathbb{R}^n} \left( \nabla \cdot \left( \tilde{\phi}_t(z) \mathbb{E}_{\tilde{\phi}_{rZ|t}}[v^{(\mathrm{Z})}_{sm+r} | \tilde{Z}_t = z] \right) + \frac{1}{\gamma} \Delta \tilde{\phi}_t(z) \right) \log \frac{\tilde{\phi}_t}{\nu} \mathrm{d}z \\
&= \int \left( \nabla \cdot \left( \tilde{\phi}_t \left( \frac{1}{\gamma} \nabla \log \frac{\tilde{\phi}_t}{\nu} + \mathbb{E}_{\tilde{\phi}_{r\beta^{(\mathrm{Z})}_{sm+r}|t}}[v^{(\mathrm{Z})}_{sm+r} | \tilde{Z}_t = z] - \nabla F \right) \right) \right) \log \frac{\tilde{\phi}_t}{\nu} \mathrm{d}z \\
&= -\int \tilde{\phi}_t \left\langle \frac{1}{\gamma} \nabla \log \frac{\tilde{\phi}_t}{\nu} + \mathbb{E}_{\tilde{\phi}_{r\beta^{(\mathrm{Z})}_{sm+r}|t}}[v^{(\mathrm{Z})}_{sm+r} | \tilde{Z}_t = z] - \nabla F, \nabla \log \frac{\tilde{\phi}_t}{\nu} \right\rangle \mathrm{d}z \\
&= -\int \tilde{\phi}_t \frac{1}{\gamma} \left\| \log \frac{\tilde{\phi}_t}{\nu} \right\|^2 \mathrm{d}z \\
&\quad + \int_{\mathbb{R}^n} \tilde{\phi}_t \left\langle \nabla F - \mathbb{E}_{\tilde{\phi}_{r\beta^{(\mathrm{Z})}_{sm+r}|t}}[v^{(\mathrm{Z})}_{sm+r} | \tilde{Z}_t = z], \nabla \log \frac{\tilde{\phi}_t}{\nu} \right\rangle \mathrm{d}z \\
&= -\frac{1}{\gamma} J_\nu(\tilde{\phi}_t) \\
&\quad + \int \tilde{\phi}_{rt\beta^{(\mathrm{Z})}_{sm+r}} \left\langle \nabla F - v^{(\mathrm{Z})}_{sm+r}, \nabla \log \frac{\tilde{\phi}_t}{\nu} \right\rangle \mathrm{d}Z_{sm+r} \mathrm{d}z \mathrm{d}\beta^{(\mathrm{Z})}_{sm+r} \\
&= -\frac{1}{\gamma} J_\nu(\tilde{\phi}_t) + \mathbb{E}_{\tilde{\phi}_{rt\beta^{(\mathrm{Z})}_{sm+r}}} \left[ \left\langle \nabla F(\tilde{Z}_t) - v^{(\mathrm{Z})}_{sm+r}, \nabla \log \frac{\tilde{\phi}_t(\tilde{Z}_t)}{\nu(\tilde{Z}_t)} \right\rangle \right].
\end{aligned}
$$

Now, let us define the second term of the right-hand side of the very last equality as Ⓑ. Applying $\langle a, b \rangle \leq \gamma \|a\|^2 + \frac{1}{4\gamma} \|b\|^2$ to this, we obtain

$$
\begin{aligned}
\text{Ⓑ} &\leq \gamma \mathbb{E}_{\tilde{\phi}_{rt\beta^{(\mathrm{Z})}_{sm+r}}} \left[ \| \nabla F(\tilde{Z}_t) - v^{(\mathrm{Z})}_{sm+r} \|^2 \right] + \frac{1}{4\gamma} \mathbb{E}_{\tilde{\phi}_{rt\beta^{(\mathrm{Z})}_{sm+r}}} \left[ \left\| \nabla \log \frac{\tilde{\phi}_t(\tilde{Z}_t)}{\nu(\tilde{Z}_t)} \right\|^2 \right] \\
&\leq 2\gamma \mathbb{E}_{\tilde{\phi}_{rt\beta^{(\mathrm{Z})}_{sm+r}}} \left[ \| \nabla F(\tilde{Z}_t) - \nabla F(Z_{sm+r}) \|^2 \right] + 2\gamma \mathbb{E}_{\tilde{\phi}_{rt\beta^{(\mathrm{Z})}_{sm+r}}} \left[ \| \nabla F(Z_{sm+r}) - v^{(\mathrm{Z})}_{sm+r} \|^2 \right] \\
&\quad + \frac{1}{4\gamma} J_\nu(\tilde{\phi}_t) \\
&\leq 2\gamma L^2 \mathbb{E}_{\tilde{\phi}_{rt\beta^{(\mathrm{Z})}_{sm+r}}} \left[ \| \tilde{Z}_t - Z_{sm+r} \|^2 \right] + \sum_{i=1}^{r} 2\gamma \Xi L^2 \eta^2 \mathbb{E}[\| v^{(\mathrm{Z})}_{sm+i-1} \|^2] + 4\eta m d L^2 \Xi \\
&\quad + \frac{1}{4\gamma} J_\nu(\tilde{\phi}_t),
\end{aligned}
$$

where for the last inequality, we used the smoothness of $F$ and Lemma B.2.

As $\tilde{Z}_t = Z_{sm+r} - t v^{(\mathrm{Z})}_{sm+r} + \sqrt{2t/\gamma} \epsilon_{sm+r}$ ($\epsilon_{sm+r} \sim N(0, I)$), from Lemma B.3, we have

$$
\begin{aligned}
\mathbb{E}[\| \tilde{Z}_t - Z_{sm+r} \|^2] &= \mathbb{E}[\| - t v^{(\mathrm{Z})}_{sm+r} + \sqrt{2t/\gamma} \epsilon_{sm+r} \|^2] \\
&= t^2 \mathbb{E}[\| v^{(\mathrm{Z})}_{sm+r} \|^2] + 2td/\gamma \\
&\leq t^2 \left( \frac{8L^2}{\alpha} H_\nu(\phi_{sm+r}) + P \right) \\
&\quad + t^2 \sum_{i=0}^{r-1} Q(Q+1)^{r-i-1} \left( \frac{8L^2}{\alpha} H_\nu(\phi_{sm+i}) + P \right) \\
&\quad + 2td/\gamma.
\end{aligned}
$$

Furthermore, by the proof of Lemma B.3, we know that the following holds:

$$
\sum_{i=1}^{r} \mathbb{E}[\| v^{(\mathrm{Z})}_{sm+i-1} \|^2] \leq \sum_{i=0}^{r-1} (Q+1)^{r-i-1} \left( \frac{8L^2}{\alpha} H_\nu(\phi_{sm+i}) + P \right).
$$

As a result, taking into account that we are only concerned about the time interval $0 \le t \le \eta$, applying $t \le \eta$, we conclude

$$\text{\textcircled{B}} \le 2\gamma L^2 \eta^2 \left( \frac{8L^2}{\alpha} H_\nu(\phi_{sm+r}) + P + \sum_{i=0}^{r-1} Q(Q+1)^{r-i-1} \left( \frac{8L^2}{\alpha} H_\nu(\phi_{sm+i}) + P \right) \right)$$

$$+ 4\eta d L^2 + 2\gamma L^2 \eta^2 \Xi \sum_{i=0}^{r-1} (Q+1)^{r-i-1} \left( \frac{8L^2}{\alpha} H_\nu(\phi_{sm+i}) + P \right) + 4\eta m d L^2 \Xi$$

$$+ \frac{1}{4\gamma} J_\nu(\tilde{\phi}_t)$$

$$\le \frac{16L^4 \gamma \eta^2}{\alpha} H_\nu(\phi_{sm+r}) + \sum_{i=0}^{r-1} (Q+1)^{r-i} \frac{16\gamma L^4 \eta^2}{\alpha} H_\nu(\phi_{sm+i}) + 2\gamma L^2 \eta^2 \sum_{i=0}^{r} (Q+1)^{r-i} P$$

$$+ 4\eta d L^2 (1 + 2m\Xi) + \frac{1}{4\gamma} J_\nu(\tilde{\phi}_t)$$

$$\le \frac{16L^4 \gamma \eta^2}{\alpha} H_\nu(\phi_{sm+r}) + \sum_{i=0}^{r-1} \frac{16L^4 \gamma \eta^2}{\alpha} (Q+1)^r H_\nu(\phi_{sm+i}) + 2\gamma L^2 \eta^2 \sum_{i=0}^{r} (Q+1)^r P$$

$$+ 4\eta d L^2 (1 + 2m\Xi) + \frac{1}{4\gamma} J_\nu(\tilde{\phi}_t)$$

$$\le \frac{16L^4 \gamma \eta^2}{\alpha} H_\nu(\phi_{sm+r}) + \sum_{i=0}^{r-1} \frac{16L^4 \gamma \eta^2}{\alpha} (Q+1)^m H_\nu(\phi_{sm+i}) + 2\gamma L^2 \eta^2 m (Q+1)^m P$$

$$+ 4\eta d L^2 (1 + 2m\Xi) + \frac{1}{4\gamma} J_\nu(\tilde{\phi}_t).$$

where for the second inequality we used $\Xi \le 1$ and for the last inequality $r < m$.

Here, as $\Xi \le 1$ and $\eta \le \frac{1}{4mL}$ by $\alpha \le \gamma L$,

$$(Q+1)^m \le e^{Qm} = e^{2L^2 m \eta^2 \Xi} \le e^{1/4} \le 2.$$

Therefore,

$$\text{\textcircled{B}} \le \frac{16L^4 \gamma \eta^2}{\alpha} H_\nu(\phi_{sm+r}) + \sum_{i=0}^{r-1} \frac{32L^4 \gamma \eta^2}{\alpha} H_\nu(\phi_{sm+i})$$

$$+ 4\gamma L^2 \eta^2 m P + 4\eta d L^2 (1 + 2m\Xi) + \frac{1}{4\gamma} J_\nu(\tilde{\phi}_t)$$

$$\le \frac{16L^4 \gamma \eta^2}{\alpha} H_\nu(\phi_{sm+r}) + \sum_{i=0}^{r-1} \frac{32L^4 \gamma \eta^2}{\alpha} H_\nu(\phi_{sm+i})$$

$$+ 4\gamma L^2 \eta^2 m \left( \frac{4dL}{\gamma} + \frac{4\eta m d L^2}{\gamma} \Xi \right) + 4\eta d L^2 (1 + 2m\Xi) + \frac{1}{4\gamma} J_\nu(\tilde{\phi}_t)$$

$$\le \frac{16L^4 \gamma \eta^2}{\alpha} H_\nu(\phi_{sm+r}) + \sum_{i=0}^{r-1} \frac{32L^4 \gamma \eta^2}{\alpha} H_\nu(\phi_{sm+i})$$

$$+ 4\eta d L^2 (2 + \Xi + 2m\Xi) + \frac{1}{4\gamma} J_\nu(\tilde{\phi}_t).$$

where for the last inequality, we used $\eta \le \frac{1}{4mL}$.

Thus,

$$\frac{\mathrm{d}}{\mathrm{d}t} H_\nu(\tilde{\phi}_t) \le -\frac{3}{4\gamma} J_\nu(\tilde{\phi}_t) + \frac{16L^4 \gamma \eta^2}{\alpha} H_\nu(\phi_{sm+r}) + \sum_{i=0}^{r-1} \frac{32L^4 \gamma \eta^2}{\alpha} H_\nu(\phi_{sm+i})$$

$$+ 4\eta d L^2 (2 + \Xi + 2m\Xi).$$

According to Assumption 2,

$$\frac{\mathrm{d}}{\mathrm{d}t} H_\nu(\tilde{\phi}_t) \le -\frac{3\alpha}{2\gamma} H_\nu(\tilde{\phi}_t) + \frac{16L^4\gamma\eta^2}{\alpha} H_\nu(\phi_{sm+r}) + \sum_{i=0}^{r-1} \frac{32L^4\gamma\eta^2}{\alpha} H_\nu(\phi_{sm+i})$$
$$+ 4\eta dL^2 \left(2 + \Xi + 2m\Xi\right).$$

Grouping the second to fourth terms as $U_{sm+r}^{(Z)}$ and multiplying both sides by $\mathrm{e}^{\frac{3\alpha}{2\gamma}t}$, we can write the above equation as

$$\frac{\mathrm{d}}{\mathrm{d}t}\left(\mathrm{e}^{\frac{3\alpha}{2\gamma}t} H_\nu(\tilde{\phi}_t)\right) \le \mathrm{e}^{\frac{3\alpha}{2\gamma}t} U_{sm+r}^{(Z)}.$$

Integrating both sides from $t = 0$ to $t = \eta$ and using $\tilde{\phi}_\eta = \phi_{sm+r+1}$, we obtain

$$\mathrm{e}^{\frac{3\alpha}{2\gamma}\eta} H_\nu(\phi_{sm+r+1}) - H_\nu(\phi_{sm+r}) \le \frac{2\gamma(\mathrm{e}^{\frac{3\alpha}{2\gamma}\eta} - 1)}{3\alpha} U_{sm+r}^{(Z)}$$
$$\le 2\eta U_{sm+r}^{(Z)}.$$

Here, for the last inequality, we used $\mathrm{e}^c \le 1 + 2c$ $(0 < c = \frac{3\alpha}{2\gamma}\eta \le 1)$ holds since $0 < \eta \le \frac{\alpha}{16\sqrt{2}L^2m\gamma} \le \frac{2\gamma}{3\alpha}$, where we used $1/L \le \gamma/\alpha$ and $m \ge 1$. Rearranging this, we obtain

$$H_\nu(\phi_{sm+r+1}) \le \mathrm{e}^{-\frac{3\alpha}{2\gamma}\eta}\left(1 + \frac{32\gamma L^4\eta^3}{\alpha}\right) H_\nu(\phi_{sm+r}) + \mathrm{e}^{-\frac{3\alpha}{2\gamma}\eta}\sum_{i=0}^{r-1} \frac{64\gamma L^4\eta^3}{\alpha} H_\nu(\phi_{sm+i})$$
$$+ \mathrm{e}^{-\frac{3\alpha}{2\gamma}\eta} 8\eta^2 dL^2 \left(2 + \Xi + 2m\Xi\right). \tag{14}$$

Furthermore, since $\eta \le \frac{\alpha}{16\sqrt{2}mL^2\gamma} \le \frac{\alpha}{8\sqrt{3}L^2\gamma}$ and $\mathrm{e}^{-\frac{3\alpha}{2\gamma}\eta} \le 1$,

$$H_\nu(\phi_{sm+r+1}) \le \mathrm{e}^{-\frac{3\alpha}{2\gamma}\eta}\left(1 + \frac{\alpha}{4\gamma}\eta\right) H_\nu(\phi_{sm+r}) + \mathrm{e}^{-\frac{3\alpha}{2\gamma}\eta}\sum_{i=0}^{r-1} \frac{\alpha}{8\gamma m}\eta H_\nu(\phi_{sm+i})$$
$$+ 8\eta^2 dL^2 \left(2 + \Xi + 2m\Xi\right).$$

On the other hand, since $\eta \le \frac{\alpha}{8mL^2\gamma}$ and $\alpha \le \gamma L$ holds,

$$\mathrm{e}^{-\frac{\alpha m}{\gamma}\eta} \ge \mathrm{e}^{-\frac{\alpha m}{\gamma}\cdot\frac{\alpha}{8mL^2\gamma}} = \mathrm{e}^{-\frac{\alpha^2}{8L^2\gamma^2}} \ge \mathrm{e}^{-1/8} \ge 0.88 \ge \frac{1}{2},$$

which further implies

$$H_\nu(\phi_{sm+r+1}) \le \mathrm{e}^{-\frac{3\alpha}{2\gamma}\eta}\left(1 + \frac{\alpha}{4\gamma}\eta\right) H_\nu(\phi_{sm+r}) + \mathrm{e}^{-\frac{3\alpha}{2\gamma}\eta}\sum_{i=0}^{r-1} \frac{\alpha}{4m\gamma}\eta\mathrm{e}^{-\frac{\alpha m}{\gamma}\eta} H_\nu(\phi_{sm+i})$$
$$+ 8\eta^2 dL^2 \left(2 + \Xi + 2m\Xi\right).$$

Q.E.D

Finally, let us prove Theorem 2 and Corollary 2.1.

**Theorem B.2** (Theorem 2 restated). *Under Assumptions 1 and 2, $0 < \eta < \frac{\alpha}{16\sqrt{2}L^2m\gamma}$ and $\gamma \ge 1$, for all $k$, the following holds in the update of SARAH-LD:*

$$H_\nu(\phi_k) \le \mathrm{e}^{-\frac{\alpha\eta}{\gamma}k} H_\nu(\phi_0) + \frac{32\eta\gamma dL^2}{3\alpha}\left(2 + \Xi + 2m\Xi\right),$$

*where $\Xi = \frac{(n-B)}{B(n-1)}$.*

*Proof.* Same as Theorem A.2.

Q.E.D

**Corollary B.2.1** (Corollary 2.1 restated). *Under the same assumptions as Theorem B.2, for all $\epsilon \geq 0$, if we choose step size $\eta$ such that*

$$\eta \leq \frac{3\alpha\epsilon}{64\gamma dL^2} \left(2 + \Xi + 2m\Xi\right)^{-1},$$

*then a precision $H_\nu(\phi_k) \leq \epsilon$ is reached after*

$$k \geq \frac{\gamma}{\alpha\eta} \log \frac{2H_\nu(\phi_0)}{\epsilon}$$

*steps. Especially, if we take $B = m = \sqrt{n}$ and the largest permissible step size $\eta = \frac{\alpha}{16\sqrt{2}L^2\sqrt{n}\gamma} \wedge \frac{3\alpha\epsilon}{320dL^2\gamma}$, then the gradient complexity becomes*

$$\tilde{O}\left(\left(n + \frac{dn^{\frac{1}{2}}}{\epsilon}\right) \cdot \frac{\gamma^2 L^2}{\alpha^2}\right).$$

*Proof.* The first half of the statement is the same as Corollary A.2.1.

When $B \geq m$, from Theorem B.2, we obtain

$$H_\nu(\phi_k) \leq \mathrm{e}^{-\frac{\alpha\eta}{\gamma}k} H_\nu(\phi_0) + \frac{32\eta\gamma dL^2}{3\alpha}\left(2 + \Xi + 2m\Xi\right)$$

$$\leq \mathrm{e}^{-\frac{\alpha\eta}{\gamma}k} H_\nu(\phi_0) + \frac{160\eta\gamma dL^2}{3\alpha}.$$

Proceeding in the same way as Corollary A.2.1, we obtain the optimal gradient complexity of

$$\tilde{O}\left(\left(n + \frac{dn^{\frac{1}{2}}}{\epsilon}\right) \cdot \frac{\gamma^2 L^2}{\alpha^2}\right)$$

with $B = m = \sqrt{n}$ and $\eta = \frac{\alpha}{16\sqrt{2}L^2\sqrt{n}\gamma} \wedge \frac{3\alpha\epsilon}{320dL^2\gamma}$.

Now, when $B \leq m$, from Theorem B.2, we obtain

$$H_\nu(\phi_k) \leq \mathrm{e}^{-\frac{\alpha\eta}{\gamma}k} H_\nu(\phi_0) + \frac{32\eta\gamma dL^2}{3\alpha}\left(2 + \Xi + 2m\Xi\right)$$

$$\leq \mathrm{e}^{-\frac{\alpha\eta}{\gamma}k} H_\nu(\phi_0) + \frac{160\eta\gamma dL^2}{3\alpha}\frac{m}{B}.$$

This leads to a gradient complexity of

$$\tilde{O}\left(\left(n + \frac{d(m + n/B)}{\epsilon}\right) \cdot \frac{\gamma^2 L^2}{\alpha^2}\right)$$

with $\eta = \frac{\alpha}{16\sqrt{2}L^2 m\gamma} \wedge \frac{3\alpha\epsilon}{320dL^2\gamma}\frac{B}{m}$, which is optimal with $B = m = \sqrt{n}$ again.

$$Q.E.D$$

# C  Proof of Theorem 3, Corollaries 3.1 and 3.2

We define $X_k$ like Algorithm 1 in order to simultaneously represent $Y_k$ and $Z_k$.

## C.1  Preparation for the Proof

### C.1.1  Link between Sampling and Optimization

Since

$$\mathbb{E}_{X_k}[F(X_k)] - F(X^*)$$

can be separated into the discretisation error

$$\mathbb{E}_{X_k}[F(X_k)] - \mathbb{E}_{X\sim\nu}[F(X)]$$

and the approximation error due to sampling

$$\mathbb{E}_{X\sim\nu}[F(X)] - F(X^*),$$

in this subsection, we analyse the upper bound of these two terms.

**Property C.1.** *Under Assumption 1, the following holds:*

$$\forall x \in \mathbb{R}^d, \ F(x) - F(x^*) \geq \frac{1}{2L}\|\nabla F(x)\|,$$

*where $x^*$ is the global minimum of $F$.*

*Proof.* Let us define $G(x) := F(x) - F(x^*)$. Since $G$ is also $L$-smooth,

$$G\left(x - \frac{1}{L}\nabla G(x)\right) \leq G(x) - \frac{1}{L}\|\nabla G(x)\|^2 + \frac{1}{2L}\|\nabla G(x)\|^2 = G(x) - \frac{1}{2L}\|\nabla G(x)\|^2,$$

where for the inequality, we used that the following holds for a $L$-smooth function $H$:

$$\forall x, y \in \mathbb{R}^d, \ H(y) \leq H(x) + \langle \nabla H(x), y - x \rangle + \frac{L}{2}\|y - x\|^2.$$

Now, since $G \geq 0$, we obtain

$$F(x) - F(x^*) \geq \frac{1}{2L}\|\nabla F(x)\|,$$

which concludes the proof.

$$Q.E.D$$

**Theorem C.1.** *Under Assumption 1, the following holds for distributions $\rho_k$ and $\nu$:*

$$\mathbb{E}_{X_k \sim \rho_k}[F(X_k)] - \mathbb{E}_{X \sim \nu}[F(X)] \leq LW_2^2(\rho_k, \nu) + \mathbb{E}_{X \sim \nu}[F(X)] - F(X^*),$$

*where $X^*$ is the global minimum of $F$. The same statement holds with $X_k \sim \phi_k$.*

*Proof.* Let $X_k \sim \rho_k$ and $X \sim \nu$ be an optimal coupling so that $\mathbb{E}[\|X_k - X\|^2] = W_2(\rho_k, \nu)^2$. The following holds only from the smoothness of $F$:

$$
\begin{aligned}
F(X_k) - F(X) &= \int_0^1 \langle X_k - X, \nabla F\left((1-t)X + tX_k\right)\rangle \mathrm{d}t \\
&\leq \int_0^1 \|X_k - X\|\|\nabla F\left((1-t)X + tX_k\right)\|\mathrm{d}t \\
&\leq \int_0^1 \|X_k - X\|\|\nabla F\left((1-t)X + tX_k\right) - \nabla F(X)\| \\
&\qquad\qquad\qquad\qquad + \|X_k - X\|\|\nabla F(X)\|\mathrm{d}t \\
&\leq \int_0^1 Lt\|X_k - X\|^2 + \frac{L}{2}\|X_k - X\|^2 + \frac{1}{2L}\|\nabla F(X)\|^2\mathrm{d}t \\
&\leq L\|X_k - X\|^2 + F(X) - F(X^*).
\end{aligned}
$$

For the first inequality, we used the Cauchy-Schwarz inequality, for the third inequality we used the smoothness of $F$ on the first term, and for the fourth inequality, Property C.1.

Hence, taking expectation of both sides, we obtain the desired result.

$$Q.E.D$$

**Corollary C.1.1.** *Under the same assumptions as Theorem C.1, the following holds:*

$$\mathbb{E}_{X_k \sim \rho_k}[F(X_k)] - F(X^*) \leq LW_2^2(\rho_k, \nu) + 2\left(\mathbb{E}_{X \sim \nu}[F(X)] - F(X^*)\right).$$

*The same statement holds with $X_k \sim \phi_k$.*

An important feature of this theorem is that the square of the 2-Wasserstein metric appears. Thanks to this and Talagrand's inequality, we can directly use the results from sampling (e.g., Corollaries 1.1 and 2.1)

The approximation error can be bounded thanks to the following theorem from Raginsky et al. (2017).

**Theorem C.2** ([Raginsky et al. (2017)](), Proposition 11). *Under Assumptions 1 and 3, for all $\gamma \geq \frac{2}{M}$*

$$\mathbb{E}_{X \sim \nu}[F(X)] - F(X^*) \leq \frac{d}{2\gamma} \log \left( \frac{\mathrm{e}L}{M} \left( \frac{b\gamma}{d} + 1 \right) \right).$$

**Corollary C.2.1.** *Under the same assumptions as Theorem C.2, for all $\epsilon > 0$, if we additionally require $\gamma \geq \frac{4d}{\epsilon} \log \left( \frac{\mathrm{e}L}{M} \right) \vee \frac{8db}{\epsilon^2}$, then*

$$\mathbb{E}_{X \sim \nu}[F(X)] - F(X^*) \leq \frac{\epsilon}{4}.$$

*Proof.* Since

$$\frac{d}{2\gamma} \log \left( \frac{\mathrm{e}L}{M} \left( \frac{b\gamma}{d} + 1 \right) \right) = \frac{d}{2\gamma} \log \frac{\mathrm{e}L}{M} + \frac{d}{2\gamma} \log \left( \frac{b\gamma}{d} + 1 \right),$$

it suffices to have $\frac{d}{2\gamma} \log \frac{\mathrm{e}L}{M} \leq \frac{\epsilon}{8}$ and $\frac{d}{2\gamma} \log \left( \frac{b\gamma}{d} + 1 \right) \leq \frac{\epsilon}{8}$. Furthermore, since for all $x \geq 0$

$$\frac{\log (x+1)}{x} \leq \frac{1}{\sqrt{x+1}} \leq \frac{1}{\sqrt{x}}$$

holds, we only need to require $\frac{d}{2\gamma} \log \frac{\mathrm{e}L}{M} \leq \frac{\epsilon}{8}$ and $\frac{b}{2} \frac{1}{\sqrt{b\gamma/d}} \leq \frac{\epsilon}{8}$. Solving these two inequalities according to $\gamma$ leads to the desired result.

$$Q.E.D$$

**Remark C.1.** *The lower bound $\frac{4d}{\epsilon} \log \left( \frac{\mathrm{e}L}{M} \right) \vee \frac{8db}{\epsilon^2}$ is only calculated to acquire a concrete condition on $\gamma$. A more involved analysis could find a better lower bound.*

### C.1.2 Explicit Formulation of the Log-Sobolev Constant

In this subsection, we give an explicit formulation of the Log-Sobolev constant of $\mathrm{d}\nu \propto \mathrm{e}^{-\gamma F} \mathrm{d}x$ in function of $\gamma$ for two cases: under Assumptions 1 and 3, and under Assumptions 1, 3 and 4 to 6. The second case is roughly the first combined with the Morse condition.

When we only assume dissipativity and smoothness, we can obtain a Log-Sobolev constant whose inverse exponentially depends on the inverse temperature $\gamma$. This employs the following result from [Raginsky et al. (2017)]().

**Property C.2** ([Raginsky et al. (2017)](), Proposition 9). *Under Assumptions 1 and 3, for all $\gamma \geq \frac{2}{M}$, $\nu$ satisfies Log-Sobolev inequality with a constant $\alpha$ such that*

$$\frac{1}{\alpha} \leq \frac{2M^2 + 2L^2}{M^2 L\gamma} + \frac{1}{\lambda_*} \left( \frac{6L(d+\gamma)}{M} + 2 \right),$$

*where*

$$\frac{1}{\lambda_*} \leq \frac{1}{M\gamma(d+b\gamma)} + \frac{2C_*(d+b\gamma)}{M\gamma} \exp \left( \frac{2}{M}(L+B_*)(b\gamma + d) + \gamma(A_* + B_*) \right).$$

*Here, $A_* = \max_i \{|f_i(0)|\}$, $B_* = \max_i \{|\nabla f_i(0)|\}$ and $C_*$ is a universal constant that does not depend on $F$.*

From this, we immediately have the following property.

**Property C.3.** *Under Assumptions 1 and 3, for all $\gamma \geq \frac{2}{M}$, we can take a Log-Sobolev constant $\alpha$ of $\nu$ which can be written with constants $C_1$ and $C_2 > 0$ independent of $\gamma$ as follows:*

$$\alpha = \gamma C_1 \mathrm{e}^{-C_2 \gamma},$$

*where*

$$C_1 = \left( \frac{2M^2 + 2L^2}{M^2 L} + \left( \frac{6Ld}{M} + 2 \right) \left( \frac{1}{Md} + \frac{2C_* d}{M} \mathrm{e}^{\frac{2d}{M}(L+B_*)} \right) \right)^{-1},$$

*and*

$$C_2 = \frac{2b}{M}(L+B_*) + (A_* + B_*) + b + 1.$$

*Proof.* From Proposition C.2,

$$\frac{\gamma}{\alpha} \leq \frac{2M^2 + 2L^2}{M^2 L} + \frac{\gamma}{\lambda_*}\left(\frac{6L(d+\gamma)}{M} + 2\right),$$

and

$$\frac{\gamma}{\lambda_*} \leq \frac{1}{M(d+b\gamma)} + \frac{2C_*(d+b\gamma)}{M}\exp\left(\frac{2}{M}(L+B_*)(b\gamma+d) + \gamma(A_*+B_*)\right).$$

Roughly bounding these inequalities, we obtain

$$\frac{\gamma}{\lambda_*} \leq \frac{1}{Md} + \frac{2C_*(d+b\gamma)}{M}\mathrm{e}^{\frac{2d}{M}(L+B_*)}\mathrm{e}^{\left(\frac{2b}{M}(L+B_*)+(A_*+B_*)\right)\gamma}$$

$$\leq \frac{1}{Md} + \frac{2C_*d}{M}\mathrm{e}^{\frac{2d}{M}(L+B_*)}\mathrm{e}^{\left(\frac{2b}{M}(L+B_*)+(A_*+B_*)+b\right)\gamma}$$

$$\leq \left(\frac{1}{Md} + \frac{2C_*d}{M}\mathrm{e}^{\frac{2d}{M}(L+B_*)}\right)\mathrm{e}^{\left(\frac{2b}{M}(L+B_*)+(A_*+B_*)+b\right)\gamma},$$

where for the second inequality we used $d + b\gamma \geq d\mathrm{e}^{b\gamma}$ for all $\gamma > 0$ when $d \geq 1$. Thus,

$$\frac{\gamma}{\alpha} \leq \frac{2M^2+2L^2}{M^2L} + \frac{\gamma}{\lambda_*}\left(\frac{6L(d+\gamma)}{M}+2\right)$$

$$\leq \frac{2M^2+2L^2}{M^2L}\mathrm{e}^{\left(\frac{2b}{M}(L+B_*)+(A_*+B_*)+b+1\right)\gamma} + \frac{\gamma}{\lambda_*}\left(\frac{6Ld}{M}+2\right)\mathrm{e}^{\gamma}$$

$$\leq \left(\frac{2M^2+2L^2}{M^2L} + \left(\frac{6Ld}{M}+2\right)\left(\frac{1}{Md} + \frac{2C_*d}{M}\mathrm{e}^{\frac{2d}{M}(L+B_*)}\right)\right)\mathrm{e}^{\left(\frac{2b}{M}(L+B_*)+(A_*+B_*)+b+1\right)\gamma}.$$

Finally, taking into account that a lower bound of a Log-Sobolev constant automatically satisfies the Log-Sobolev inequality, we obtain the desired result.

$$Q.E.D$$

On the other hand, under the additional condition of Morse, Lipschitzness of $\nabla^2 F$ and other minor assumptions, we can obtain a far better Log-Sobolev constant whose inverse depends only linearly on $\gamma$ as follows. This is a straightforward adaptation of Li and Erdogdu's result (Li and Erdogdu, 2020). We provide a proof in Appendix D.

**Property C.4.** *Under Assumptions 1, 3 and 4 to 6, with $\gamma \geq 1$ such that*

$$\gamma \geq C_\gamma := \max\left(1, \left(\frac{24dL}{C_F^2}\right)^2\frac{4dL'^2}{\lambda^{\dagger 2}}, 4L'^2\left(\frac{24dL}{C_F^2}\right)^6\right),$$

*where $C_F$ is defined in Lemma D.5, $\nu$ satisfies the Log-Sobolev inequality with constant $\alpha$ such that*

$$\frac{1}{\alpha} = \frac{\gamma}{C_3},$$

*where*

$$C_3 := \left(\frac{2M^2+8L^2}{M^2L} + \left(\frac{6L(d+1))}{M}+2\right)\frac{35}{\lambda^\dagger}\right).$$

## C.2 Main Proof

**Theorem C.3** (Theorem 3 restated). *Using SVRG-LD or SARAH-LD, under Assumptions 1 to 3, $0 < \eta < \frac{\alpha}{16\sqrt{6}L^2 m\gamma}$, $\gamma \geq \frac{4d}{\epsilon}\log\left(\frac{\mathrm{e}L}{M}\right) \vee \frac{8db}{\epsilon^2} \vee 1 \vee \frac{2}{M}$ and $B \geq m$, if we take $B = m = \sqrt{n}$ and the largest permissible step size $\eta = \frac{\alpha}{16\sqrt{6}L^2\sqrt{n}\gamma} \wedge \frac{3}{1792}\frac{\alpha^2\epsilon}{L^2 d\gamma}$, the gradient complexity to reach a precision of*

$$\mathbb{E}_{X_k}[F(X_k)] - F(X^*) \leq \epsilon$$

*is*

$$\tilde{O}\left(\left(n + \frac{n^{\frac{1}{2}}}{\epsilon}\cdot\frac{dL}{\alpha}\right)\frac{\gamma^2 L^2}{\alpha^2}\right),$$

*where $\alpha$ is a function of $\gamma$.*

*Proof.* It is sufficient to consider the case of SVRG-LD with $X_k \sim \rho_k$. From Corollary C.1.1, the sufficient condition for

$$\mathbb{E}_{X_k}[F(X_k)] - F(X^*) \leq \epsilon$$

is $LW_2^2(\rho_k, \nu) \leq \epsilon/2$ and $\mathbb{E}_{X \sim \nu}[F(X)] - F(X^*) \leq \epsilon/4$. From Corollary C.2.1, the latter condition is satisfied when $\gamma \geq \frac{4d}{\epsilon} \log\left(\frac{eL}{M}\right) \vee \frac{8bd}{\epsilon^2} \vee 1 \vee \frac{2}{M}$. Moreover, concerning the former, from Talagrand's inequality

$$W_2^2(\rho_k, \nu) \leq \frac{2}{\alpha} H_\nu(\rho_k),$$

it suffices to have

$$H_\nu(\rho_k) \leq \frac{\alpha\epsilon}{4L}.$$

Thus, from Corollaries A.2.1 and B.2.1, under the same conditions, we obtain a gradient complexity of

$$\tilde{O}\left(\left(n + \frac{n^{\frac{1}{2}}}{\epsilon} \cdot \frac{dL}{\alpha}\right)\frac{\gamma^2 L^2}{\alpha^2}\right).$$

*Q.E.D*

This leads to the following corollaries.

**Corollary C.3.1** (Corollary 3.1 restated)**.** *Under the same assumptions as Theorem C.3, taking*

$$\gamma = i(\epsilon) := \frac{4d}{\epsilon} \log\left(\frac{eL}{M}\right) \vee \frac{8db}{\epsilon^2} \vee 1 \vee \frac{2}{M},$$

*we obtain a gradient complexity of*

$$\tilde{O}\left(\left(n + \frac{n^{\frac{1}{2}}}{\epsilon} \cdot \frac{dL}{C_1 i(\epsilon)} e^{C_2 i(\epsilon)}\right) L^2 e^{2C_2 i(\epsilon)}\right)$$

*since $\alpha = \gamma C_1 e^{-C_2 \gamma}$ (Property C.3).*

*Proof.* The proof follows from Property C.3.

*Q.E.D*

**Corollary C.3.2** (Corollary 3.2 restated)**.** *Under the same assumptions as Theorem 11 and Assumptions 4 to 6, taking*

$$\gamma = j(\epsilon) := \frac{4d}{\epsilon} \log\left(\frac{eL}{M}\right) \vee \frac{8db}{\epsilon^2} \vee 1 \vee \frac{2}{M} \vee C_\gamma,$$

*where $C_\gamma$ is a constant independent of $\epsilon$ defined in Property C.4, we obtain a gradient complexity of*

$$\tilde{O}\left(\left(n + \frac{n^{\frac{1}{2}}}{\epsilon} \cdot \frac{dL}{C_3} j(\epsilon)\right) C_3^2 j(\epsilon)^4 L^2\right)$$

*since $\alpha = C_3/\gamma$ (Property C.4).*

*Proof.* The proof follows from Property C.4.

*Q.E.D*

# D  Proof of Property C.4

## D.1  Overview and Main Result

In this appendix, we prove Property C.4 which is only a slight adaptation of Theorem 3.4 from Li and Erdogdu (2020), which builds its foundation from prior work such as Cattiaux et al. (2010) and Menz and Schlichting (2014). We show that with additional Morse, and smoothness assumptions to dissipativity, we can obtain a Log-Sobolev constant of $d\nu \propto e^{-\gamma F} dx$ whose inverse only depends linearly on the inverse temperature parameter. Property C.4 is reminded below in a more precise form.

**Theorem 4.** *Under Assumptions 1, 3 and 4 to 6, with $a > 0$ and $\gamma \geq 1$ such that*

$$a^2 \geq \frac{24dL}{C_F^2},$$

*and*

$$\gamma \geq \max\left(1, a^2 \frac{4dL'^2}{\lambda^{\dagger 2}}, 4L'^2 a^6\right),$$

*$\nu$ satisfies the Log-Sobolev inequality with constant $\alpha$ such that*

$$\frac{1}{\alpha} = \left(\frac{2M^2 + 8L^2}{M^2 L} + \left(\frac{6L(d+1)}{M} + 2\right)\frac{35}{\lambda^\dagger}\right)\gamma.$$

This theorem shows that the strict saddle node assumption is almost sufficient to obtain in the Euclidean space for dissipative distributions a Log-Sobolev constant whose inverse does not exponentially depend on the inverse temperature, which was the case without this assumption.

## D.2 Preliminaries

We first clarify some definitions.

**Definition D.1.** *We say a probability measure $\nu$ satisfies the Poincaré inequality with a constant $\kappa$ if for all smooth $g : \mathbb{R}^d \to \mathbb{R}$,*

$$\mathbb{E}_\nu[g^2] - \mathbb{E}_\nu[g]^2 \leq \frac{1}{\kappa}\mathbb{E}_\nu[\|\nabla g\|^2].$$

**Definition D.2.** *A probability measure $\nu$ on $\mathbb{R}^d$ restricted on a set $\mathcal{Z} \subset \mathbb{R}^d$ is defined as*

$$\nu|_{\mathcal{Z}} := \frac{\nu(x)}{\int_{\mathcal{Z}} \nu(y)dy}\mathbb{1}_{\mathcal{Z}}(x).$$

**Definition D.3.** *We define the following sets:*

$$\mathcal{B} := \left\{x \in \mathbb{R}^d \mid d(x, \mathcal{S})^2 < \frac{a^2}{\gamma}\right\},$$

$$\mathcal{U} := \left\{x \in \mathbb{R}^d \mid d(x, \mathcal{X})^2 < \frac{a^2}{\gamma}\right\},$$

$$\mathcal{A} := \left\{x \in \mathbb{R}^d \mid d(x, \mathcal{S} \cup \mathcal{X})^2 \geq \frac{a^2}{4\gamma}\right\},$$

*where $\mathcal{X}$ is the set of global minima and $\mathcal{S}$ is the set of stationary points except the global minima. Note that $\mathcal{B} \cup \mathcal{U} \cup \mathcal{A} = \mathbb{R}^d$.*

*Here, the distance from a point $x \in \mathbb{R}^d$ and a set $\mathcal{Z} \subset \mathbb{R}^d$ is defined as*

$$d(x, \mathcal{Z}) := \inf_{z \in \mathcal{Z}} \|x - z\|.$$

In this appendix, we only consider the following generator $\mathcal{L}$.

**Definition D.4.** *We define $\mathcal{L}$ such that*

$$\mathcal{L}f := \langle -\nabla f, \nabla F \rangle + \frac{1}{\gamma}\Delta f, \ \forall f \in C^2(\mathbb{R}^d)$$

*which is the generator of the gradient Langevin Dynamics (1).*

We will need some lemmas proved by Li and Erdogdu (2020).

**Lemma D.1** (Li and Erdogdu (2020)). *Under Assumptions 1 and 5, suppose $y \in \mathbb{R}^d$ is a stationary point of $F$. Then, with*

$$H(x) := \nabla^2 F(0) \cdot x$$

*defined in the coordinate centered at $y$, we obtain for all $x \in \mathbb{R}^d$,*

$$\|\nabla F - H(x)\| \leq L'\|x\|^2.$$

*Proof.* From the mean value theorem, there exist a $\hat{x}$ on the line between $0$ and $x$ such that

$$\nabla F(x) = \nabla F(0) + \nabla^2 F(\hat{x}) \cdot x = \nabla^2 F(\hat{x}) \cdot x,$$

where for the last equality, we used that $0$ was a stationary point. Therefore, we obtain

$$\begin{aligned}
\|\nabla F(x) - H(x)\| &= \|\nabla^2 F(\hat{x}) \cdot x - \nabla^2 F(0) \cdot x\| \\
&\leq \|\nabla^2 F(\hat{x}) - \nabla^2 F(0)\| \|x\| \\
&\leq L' \|\hat{x}\| \|x\| \\
&\leq L' \|x\|^2,
\end{aligned}$$

where for the second inequality we used the $L'$-Lipschitzness of $\nabla^2 F$, and for the last inequality we used $\|\hat{x}\| \leq \|x\|$.

$$Q.E.D$$

**Lemma D.2** (Li and Erdogdu (2020), Proposition E.5). *Let $W_t$ and $\tilde{W}_t$ be weak solutions on some filtered probability space of the following one dimensional SDE's:*

$$\begin{aligned}
\mathrm{d}W_t &= \Phi(W_t)\mathrm{d}t + \sigma \mathrm{d}B_t, \\
\mathrm{d}\tilde{W}_t &= \tilde{\Phi}(\tilde{W}_t)\mathrm{d}t + \sigma \mathrm{d}B_t,
\end{aligned}$$

*where $W_0 = \tilde{W}_0$ a.s. and $\sigma > 0$ is a constant. We further assume that for all $T \geq 0$,*

$$\int_0^T |\Phi(W_t)| + |\tilde{\Phi}(\tilde{W}_t)|\mathrm{d}t < \infty, \ a.s.$$

*If $\Phi(W_t) \geq \tilde{\Phi}(\tilde{W}_t)$ for all $x \in \mathbb{R}$, then $W_t \geq \tilde{W}_t$ a.s.*

**Lemma D.3** (Li and Erdogdu (2020), Corollary D.6). *Consider the following Cox-Ingersoll-Ross process defined as*

$$\mathrm{d}W_t = \left(2\lambda^\dagger W_t + \frac{1}{2\gamma}\right)\mathrm{d}t + \frac{2}{\sqrt{\gamma}}\sqrt{W_t}\mathrm{d}B_t, \ W_0 = w_0 \geq 0,$$

*where $\lambda^\dagger > 0$, $\gamma > 0$ and $\{B_t\}_{t \geq 0}$ is a standard one dimensional Brownian motion. Then for its unique strong solution $W_t$, we have the following density function:*

$$f(w; t) = 2^{-\frac{5}{4}}\left(\frac{w}{w_0}\right)^{-\frac{1}{4}} \frac{\lambda^\dagger \gamma}{\mathrm{e}^{\frac{\lambda^\dagger t}{2}}\sinh(\lambda^\dagger t)} \exp\left(\frac{\lambda^\dagger \gamma (w\mathrm{e}^{-2\lambda^\dagger t} - \frac{w_0}{2})}{1 - \mathrm{e}^{-2\lambda^\dagger t}}\right) I_{-\frac{1}{2}}\left(\frac{\lambda^\dagger \gamma}{\sinh(\lambda^\dagger t)}\sqrt{\frac{w w_0}{2}}\right)$$

*for $w > 0$ and $f(w; t) = 0$ for $w = 0$, where $I_{-\frac{1}{2}}$ is the modified Bessel function of the first kind of degree $-\frac{1}{2}$. Thus, $W_t > 0$ a.s.*

**Lemma D.4** (Li and Erdogdu (2020), Lemma C.7). *For the density function $f(w; t)$ defined in Lemma D.3, we have for $w \leq R$ and $t \geq 0$,*

$$f(w; t) \leq C\mathrm{e}^{-2\lambda^\dagger t},$$

*where $C := C(R, \lambda^\dagger, \gamma) > 0$ is a constant independent of $t$ and $w_0$.*

Finally, the next two theorems will be highly useful to establish the Poincaré inequality with an explicit constant.

**Theorem D.1** (Bakry et al. (2008), Theorem 1.4 adapted). *Suppose $\nu|_{\mathcal{Z}}$ ($\mathcal{Z} \subset \mathbb{R}^d$) satisfies the Poincaré inequality with constant $\kappa_{\mathcal{Z}}$ and there exists a Lyapunov function $V \in C^2(\mathcal{Z}')$, where $\mathcal{Z} \subset \mathcal{Z}'$. That is, $V \geq 1$ and there exist constants $\theta > 0$ and $b \geq 0$ such that*

$$\mathcal{L}V = \langle -\nabla F, \nabla V \rangle + \frac{1}{\gamma}\Delta V \leq -\theta V + b\mathbb{1}_{\mathcal{Z}}.$$

*Then $\nu|_{\mathcal{Z}'}$ satisfies the Poincaré inequality with constant*

$$\kappa = \frac{\theta}{1 + b/\kappa_{\mathcal{Z}}}.$$

**Theorem D.2** (Li and Erdogdu (2020), Lemma B.14 adapted). *Set the following neighbourhood of saddle points*

$$\mathcal{B}_r = \{x \in \mathbb{R}^d \mid d(x, \mathcal{S}) < r\}.$$

*Let $r > \tilde{r} > 0$ and suppose $\nu \mid_{\mathcal{B}_{\tilde{r}}}$ satisfies the Poincaré inequality with constant $\tilde{\kappa}$ and there exist a Lyapunov function $1 \leq V \in C^2(\mathcal{B})$ and a constant $\theta > 0$ such that*

$$\mathcal{L}V \leq -\theta V.$$

*Then, $\nu$ satisfies the Poincaré inequality with constant $\kappa$ such that*

$$\frac{1}{\kappa} = \frac{4}{\theta} + \left(\frac{4}{\theta\gamma(r-\tilde{r})^2} + 2\right)\frac{1}{\tilde{\kappa}}.$$

From these theorems, we will be able to find a Poincaré constant for $\nu$ consecutively from $\mathcal{U}$ to $\mathcal{U} \cup \mathcal{A}$ and then to $\mathbb{R}^d$.

### D.3 Lyapunov Function for $\mathcal{B}$

The following theorem gives a sufficient condition to find a Lyapunov function for $\mathcal{B}$ in the sense of Theorem D.2. This is actually a combination of Bovier and Den Hollander's Theorem 7.15 (Bovier and Den Hollander, 2016) and Wainwright's Theorem 2.13 (Wainwright, 2019). See Li and Erdogdu (2020) for details.

**Theorem D.3** (Li and Erdogdu (2020), Proposition 9.5). *If there exist constants $c_1 > 0$ and $c_2 > 0$ such that*

$$P(\tau_{\mathcal{B}^c} \geq t \mid X_0 = x) \leq c_1 e^{-c_2 t}, \quad \forall t \geq 0, \forall x \in \mathcal{B},$$

*where $\tau_{\mathcal{B}^c} := \inf\{t \geq 0 \mid X_t \notin \mathcal{B}\}$, then by defining $V(x) := \mathbb{E}[e^{c_2\tau_{\mathcal{B}^c}/2} \mid X_0 = x]$, the following holds:*

$$\mathcal{L}V \leq -\frac{c_2}{2}V.$$

It is thus enough to establish an exponentially decaying tail bound for $\tau_{\mathcal{B}^c}$ as shown in the next theorem.

**Theorem D.4** (Li and Erdogdu (2020), Proposition 9.6 adapted). *Let $\{X_t\}_{t\geq 0}$ be the Langevin diffusion defined in (1). Under Assumptions 1, 4 and 5, with $a > 0$ and $\gamma > 0$ such that*

$$\gamma \geq \max\left(a^2, 4L'^2 a^6\right),$$

*the following holds:*

$$P(\tau_{\mathcal{B}^c} \geq t \mid X_0 = x) \leq c_1 e^{-\lambda^\dagger t/2}, \quad \forall t \geq 0, \forall x \in \mathcal{B},$$

*where $c_1 := c_1(a, \gamma, \lambda^\dagger)$ is a constant independent of $t$ and $x$. Hence, $V(x) := \mathbb{E}[e^{\theta\tau_{\mathcal{B}^c}} \mid X_0 = x]$ is a Lyapunov function on $\mathcal{B}$ in the sense of Theorem D.2 with parameter $\theta = \frac{\lambda^\dagger}{4}$.*

*Proof.* For each $y \in \mathcal{S}$, we define $v_y$ as the unit eigenvector of $\nabla^2 F(y)$ that corresponds to the minimum eigenvalue of $\nabla^2 F(y)$. From Assumption 4, we have that $\langle v_y, \nabla^2 F(y)v_y\rangle \leq -\lambda^\dagger$. Now, let us fix a $x \in \mathcal{B}$ and take a $y \in \mathcal{S}$ such that $\|x - y\|^2 < \frac{a^2}{\gamma}$. In the remainder of this proof, we will work in coordinates centered at this $y$. Without loss of generality, we can thus set $y = 0$.

Let $r(x) := \|x\|$. Then

$$\nabla r(x) = \frac{x}{\|x\|}.$$

We also define $P : \mathcal{B} \to \tilde{\mathcal{B}}$ where $\tilde{\mathcal{B}} := \left\{tv_0 \mid |t| < \frac{a^2}{\gamma}\right\}$ such that

$$Px := \langle v_0, x\rangle v_0,$$

and $\tilde{r}(x) := |\langle v_0, x \rangle|$.[2] As a result,

$$\nabla \tilde{r}(x) = \frac{Px}{\|Px\|} = \text{sign}(\langle v_0, x \rangle)v_0.$$

Using Itô's formula, we obtain

$$d \left( \frac{1}{2} \tilde{r}(X_t)^2 \right) = \left( \langle -\nabla F(X_t), \tilde{r}(X_t) \nabla \tilde{r}(X_t) \rangle + \frac{1}{\gamma} \left( \|\nabla \tilde{r}(X_t)\|^2 + \tilde{r}(X_t) \Delta \tilde{r}(X_t) \right) \right) dt$$
$$+ \frac{2}{\gamma} \langle \tilde{r}(X_t) \nabla \tilde{r}(X_t), dW_t \rangle.$$

Since $\nabla \tilde{r}(x)$ is a unit vector, we can consider $\langle \nabla \tilde{r}(X_t), dW_t \rangle$ as a standard one-dimensional Brownian motion independent of $X_t$ that we denote as $dB_t$.

Next, considering

$$H(x) := \nabla^2 F(0) \cdot x,$$

it follows from Lemma D.1 that

$$\|\nabla F(x) - H(x)\| \leq L'\|x\|^2.$$

Therefore,

$$\begin{aligned}
\langle -\nabla F(X_t), \nabla \tilde{r}(X_t) \rangle &= \langle -H(X_t), \nabla \tilde{r}(X_t) \rangle - \langle F(X_t) - \nabla H(X_t), \nabla \tilde{r}(X_t) \rangle \\
&\geq \langle -H(X_t), \nabla \tilde{r}(X_t) \rangle - \|H(X_t) - \nabla F(X_t)\| \|\nabla \tilde{r}(X_t)\| \\
&\geq \langle -H(X_t), \nabla \tilde{r}(X_t) \rangle - \|H(X_t) - \nabla F(X_t)\| \\
&\geq \langle -H(X_t), \nabla \tilde{r}(X_t) \rangle - L'\|X_t\|^2,
\end{aligned}$$

where for the second inequality, we used that $\nabla \tilde{r}(x)$ is a unit vector.

Using the definition of $H(x)$, we can further write

$$\langle -H(x), \nabla \tilde{r}(x) \rangle = - \left\langle \nabla^2 F(0) \cdot x, \frac{Px}{\|Px\|} \right\rangle \geq \lambda^\dagger \tilde{r}(x),$$

where for the inequality, we used that $v_0$ is an eigenvector of $\nabla^2 F(0)$ and Assumption 4.

Since $\Delta \tilde{r}(x) \geq 0$ we have that $\|\nabla \tilde{r}(x)\|^2 + \tilde{r}(x) \Delta \tilde{r}(x) \geq 1$. Therefore, from Lemma D.2, $\frac{1}{2} \tilde{r}(X_t)^2$ is lower bounded by the stochastic process $\frac{1}{2} \left( r_t^{(1)} \right)^2$ defined as

$$d \left( \frac{1}{2} \left( r_t^{(1)} \right)^2 \right) = \left( \lambda^\dagger \left( r_t^{(1)} \right)^2 - L'\|X_t\|^2 r_t^{(1)} + \frac{1}{\gamma} \right) dt + \sqrt{\frac{2}{\gamma}} r_t^{(1)} dB_t.$$

Since we are only concerned with $X_t \in \mathcal{B}$, the following holds:

$$\|X_t\|^2 \leq \frac{a^2}{\gamma},$$

and

$$r_t^{(1)} \leq \tilde{r}(X_t) \leq \sqrt{\frac{a^2}{\gamma}}.$$

We can again use Lemma D.2 to obtain a lower bound of $r_t^{(1)}$ defined as

$$\begin{aligned}
d \left( \frac{1}{2} \left( r_t^{(2)} \right)^2 \right) &= \left( \lambda^\dagger \left( r_t^{(2)} \right)^2 - L' \frac{a^3}{\gamma^{3/2}} + \frac{1}{\gamma} \right) dt + \sqrt{\frac{2}{\gamma}} r_t^{(2)} dB_t \\
&= \left( \lambda^\dagger \left( r_t^{(2)} \right)^2 + \frac{1}{\gamma} \left( 1 - L' \frac{a^3}{\gamma^{1/2}} \right) \right) dt + \sqrt{\frac{2}{\gamma}} r_t^{(2)} dB_t.
\end{aligned}$$

---

[2]Note that, $\tilde{r}(x)$ is not differentiable for $x$ such that $\tilde{r}(x) = 0$. For these points, we redefine $\nabla \tilde{r}(x)$ and $\Delta \tilde{r}(x)$ to be some constant $C_r > 0$, but this case can be ignored as shown later.

Since $\gamma \geq 4L'^2 a^6$, $1 - L'\frac{a^3}{\gamma^{1/2}} \geq \frac{1}{2}$. This gives us a further lower bound $r_t^{(3)}$ defined as

$$\mathrm{d}\left(\frac{1}{2}\left(r_t^{(3)}\right)^2\right) = \left(2\lambda^\dagger \frac{1}{2}\left(r_t^{(3)}\right)^2 + \frac{1}{2\gamma}\right)\mathrm{d}t + \sqrt{\frac{2}{\gamma}}r_t^{(3)}\mathrm{d}B_t.$$

This is a Cox-Ingersoll-Ross process with $W_t = \frac{1}{2}\left(r_t^{(3)}\right)^2$. Consequently, from Lemmas D.3 and D.4, when $x \leq \frac{a^2}{2\gamma}$, the density function $f(w;t)$ satisfies

$$f(w;t) \leq C\mathrm{e}^{-\lambda^\dagger t/2},$$

for all $t \geq 0$ and some constant $C := C(a, \gamma, \lambda^\dagger)$ independent of $t$ and $x$. Furthermore, since $\tilde{r}(X_t)$ is lower-bounded by $r_t^{(3)}$ which is almost surely positive by Lemma D.3, the case $\tilde{r}(X_t) = 0$ can be ignored.

As a result, we obtain

$$
\begin{aligned}
P\left(\tau_{\mathcal{B}^c} \geq t \mid X_0 = x\right) &= P\left(\sup_{s \in [0,t]} \frac{1}{2}r(X_s)^2 \leq \frac{a^2}{2\gamma} \mid X_0 = x\right) \\
&\leq P\left(\frac{1}{2}r(X_t)^2 \leq \frac{a^2}{2\gamma} \mid X_0 = x\right) \\
&\leq P\left(\frac{1}{2}\left(r_t^{(3)}\right)^2 \leq \frac{a^2}{2\gamma} \mid r_0^{(3)} = \tilde{r}(x)\right) \\
&= \int_0^{\frac{a^2}{2\gamma}} f(w;t)dw \\
&\leq \frac{a^2}{2\gamma}C\mathrm{e}^{-\lambda^\dagger t/2},
\end{aligned}
$$

where for the second equality, we used Lemma D.3 and for the last inequality we used Lemma D.4. This gives the desired result.

$$Q.E.D$$

## D.4 Lyapunov Function for $\mathcal{A}$

In this section, we prepare statements for the Lyapunov function on $\mathcal{A}$.

**Lemma D.5** (Li and Erdogdu (2020), Lemma 9.7 adapted). *Under Assumptions 1,3, 4 and 5, with $\mathcal{C} := \mathcal{S} \cup \mathcal{X}$, there exists a constant $0 < C_F \leq 1$ such that*

$$\|\nabla F(x)\| \geq C_F d(x, \mathcal{C}),$$

*where*

$$C_F := \min\left(1, \frac{\lambda^\dagger}{2}, \inf_{x:d(x,\mathcal{C})>\frac{\lambda^\dagger}{4L'}} \frac{\|\nabla F(x)\|}{d(x,\mathcal{C})}\right).$$

*Proof.* First, observe that when $F$ is $(M, b)$-dissipative, we have

$$\frac{1}{M}\|\nabla F(x)\|^2 + \frac{M}{2}\|x\|^2 \geq M\|x\|^2 - b,$$

which leads to

$$\frac{\|\nabla F(x)\|}{\|x\|} \geq \sqrt{\frac{M^2}{2} - Mb/\|x\|^2}.$$

We obtain thus

$$\liminf_{\|x\|\to\infty} \frac{\|\nabla F(x)\|}{\|x\|} \geq \sqrt{\frac{M^2}{2}}. \tag{15}$$

Therefore,

$$\inf_{x:d(x,\mathcal{C})>\frac{\lambda^\dagger}{4L'}} \frac{\|\nabla F(x)\|}{d(x,\mathcal{C})} > 0,$$

since by equation (15) $\|\nabla F(x)\|/d(x,\mathcal{C}) > 0$ holds outside a compact set of $x$ around the origin and since in this compact set and away from stationary points there exist an $x$ that minimizes $\|\nabla F(x)\|/d(x,\mathcal{C})$ and that cannot be 0 as we are outside $\mathcal{C}$. As a result, we just have to consider the case when $d(x,\mathcal{C}) \leq \frac{\lambda^\dagger}{4L'}$.

Let $y$ be a stationary point such that $\|x - y\| < 2d(x,\mathcal{C}) \leq \frac{\lambda^\dagger}{2L'}$. Since $\nabla^2 F(x)$ is $L'$-Lipschitz,

$$\|\nabla F(x) - \nabla F(y) - \nabla^2 F(y)^\top (x - y)\| \leq \frac{L'}{2}\|x - y\|^2,$$

and we get

$$\begin{aligned}
\|\nabla F(x)\| &\geq \|\nabla^2 F(y)(x - y)\| - \|\nabla F(x) - \nabla^2 F(y)(x - y)\| \\
&\geq \lambda^\dagger \|x - y\| - \frac{L'}{2}\|x - y\|^2 \\
&\geq \frac{\lambda^\dagger}{2}\|x - y\|.
\end{aligned}$$

We obtain thus the desired result with $C_F > 0$ by taking the minimum of the two constants.

$$Q.E.D$$

**Lemma D.6** (Li and Erdogdu (2020), Lemma 9.8 adapted). *Under Assumptions 1, 3, 4 and 5, for $a > 0$ and $\gamma > 0$ such that*

$$a^2 \geq \frac{24dL}{C_F^2},$$

*the following holds:*

$$\frac{\Delta F(x)}{2} - \frac{\gamma}{4}\|\nabla F(x)\|^2 \leq -dL, \ \forall x \in \mathbb{R}^d : d(x,\mathcal{C}) \geq \frac{a^2}{4\gamma}.$$

*Proof.* From Lemma D.5 and smoothness of $F$, we have

$$\frac{\Delta F(x)}{2} - \frac{\gamma}{4}\|\nabla F(x)\|^2 \leq \frac{dL}{2} - \frac{\gamma}{4}C_F^2 d(x,\mathcal{C}).$$

Since $d(x,\mathcal{C}) \geq \frac{a^2}{\gamma}$ and $a^2 \geq \frac{24dL}{C_F^2}$, we obtain

$$\begin{aligned}
\frac{\Delta F(x)}{2} - \frac{\gamma}{4}\|\nabla F(x)\|^2 &\leq \frac{dL}{2} - \frac{\gamma}{4}C_F^2 d(x,\mathcal{C}) \\
&\leq \frac{dL}{2} - \frac{\gamma}{4}C_F^2 \frac{a^2}{\gamma} \\
&\leq -dL.
\end{aligned}$$

$$Q.E.D$$

## D.5 Poincaré Inequality

In this section, we establish the Poincaré inequality for $\nu$ using Theorem D.1 and D.2. It is easy to find a Poincaré constant for $\nu|_{\mathcal{U}}$ which is our starting point.

**Lemma D.7** (Li and Erdogdu (2020), Lemma 9.9 adapted). *Under Assumptions 1, 3 and 4 to 6, with $a > 0$ and $\gamma > 0$ such that*

$$\gamma \geq a^2 \frac{4dL'^2}{\lambda^{\dagger 2}},$$

*$\nu|_{\mathcal{U}}$ satisfies the Poincaré inequality with constant $\kappa_{\mathcal{U}} = \frac{\lambda^\dagger}{2}$.*

*Proof.* Let $x^*$ be the global minimum. Then $\mathcal{U} = \{x \in \mathbb{R}^d \mid \|x - x^*\|^2 < \frac{a^2}{\gamma}\}$. Using the same idea as Lemma D.5, we have

$$\min_{i \in \{1,\ldots,n\}} \lambda_i \left(\nabla^2 F(x)\right) \geq \lambda^\dagger - dL' \|x - x^*\| \geq \frac{\lambda^\dagger}{2},$$

where we used $\gamma \geq a^2 \frac{4dL'^2}{\lambda^{\dagger 2}}$ and $\|x - x^*\| \leq \sqrt{a^2 \gamma}$.

This implies for all $x \in \mathcal{U}$,

$$\nabla^2 F(x) \geq \frac{\lambda^\dagger}{2} I_{d \times d},$$

where $I_{d \times d}$ is the $d \times d$ unit matrix. Therefore, $\nu|_{\mathcal{U}}$ satisfies the Poincaré inequality with constant $\kappa_{\mathcal{U}} = \frac{\lambda^\dagger}{2}$.

$$Q.E.D$$

Next, we show that $\nu|_{\mathcal{U} \cup \mathcal{A}}$ satisfies the Poincaré inequality.

**Lemma D.8** (Li and Erdogdu (2020), Lemma 9.11 adapted). *Under Assumptions 1, 3 and 4 to 6, with $a > 0$ and $\gamma > 0$ such that*

$$a^2 \geq \frac{24dL}{C_F^2},$$

*and*

$$\gamma \geq a^2 \frac{4dL'^2}{\lambda^{\dagger 2}},$$

$\nu|_{\mathcal{U} \cup \mathcal{A}}$ *satisfies the Poincaré inequality with constant*

$$\kappa_{\mathcal{U} \cup \mathcal{A}} = \frac{1}{1 + 3/(2\kappa_{\mathcal{U}})}.$$

*Proof.* Let us choose the candidate Lyapunov function $V_1(x) = e^{\frac{\gamma}{2} F(x)}$. Then,

$$\frac{\mathcal{L}V_1}{V_1} = \frac{1}{2} \Delta F - \frac{\gamma}{4} \|\nabla F\|^2.$$

From Lemma D.6, for all $x \in \mathcal{A}$ we have

$$\frac{\mathcal{L}V_1}{V_1} \leq -dL.$$

On the other hand, for all $x \in \mathcal{U}$ we obtain

$$\frac{\mathcal{L}V_1}{V_1} = \frac{1}{2} \Delta F \leq \frac{1}{2} dL.$$

This leads to

$$\frac{\mathcal{L}V_1}{V_1} \leq -dL + \frac{3dL}{2} \mathbb{1}_{\mathcal{U}}$$

for all $x \in \mathcal{U} \cup \mathcal{A}$. Since the assumptions of Theorem D.1 are satisfied, we conclude that $\nu|_{\mathcal{U} \cup \mathcal{A}}$ satisfies the Poincaré inequality with a constant

$$\kappa_{\mathcal{U} \cup \mathcal{A}} = \frac{dL}{1 + 3dL/(2\kappa_{\mathcal{U}})}.$$

Since $L \geq 1$ and $d \geq 1$, we can replace this value by its lower bound

$$\kappa_{\mathcal{U} \cup \mathcal{A}} = \frac{1}{1 + 3/(2\kappa_{\mathcal{U}})}.$$

$$Q.E.D$$

Finally, we can establish the Poincaré inequality for $\nu$.

**Lemma D.9** (Li and Erdogdu (2020), Proposition 9.12 adapted). *Under Assumptions 1, 3 and 4 to 6, with $a > 0$ and $\gamma > 0$ such that*

$$a^2 \geq \frac{24dL}{C_F^2},$$

*and*

$$\gamma \geq \max\left( a^2 \frac{4dL'^2}{\lambda^{\dagger 2}}, 4L'^2 a^6 \right),$$

*$\nu$ satisfies the Poincaré inequality with constant*

$$\kappa = \frac{\lambda^\dagger}{35}.$$

*Proof.* Let us select the candidate Lyapunov function

$$V_2(x) = \mathbb{E}[e^{\lambda^\dagger \tau_{\mathcal{B}^c}/4} | X_0 = x],$$

where $\tau_{\mathcal{B}^c} = \inf\{t \geq 0 | X_t \notin \mathcal{B}\}$.

From Theorem D.3 and D.4, we have that $V_2$ satisfies the Lyapunov condition with

$$\frac{\mathcal{L}V_2}{V_2} \leq -\frac{\lambda^\dagger}{4}.$$

Now, using Theorem D.2 with $\theta = \frac{\lambda^\dagger}{4}$, $\tilde{r} = \frac{1}{2}r = \frac{a}{2\sqrt{\gamma}}$, we conclude that $\nu$ satisfies the Poincaré inequality with a constant $\kappa$ such that

$$\frac{1}{\kappa} = \frac{16}{\lambda^\dagger} + \left( \frac{16}{\lambda^\dagger \gamma} \frac{4\gamma}{a^2} + 2 \right) \frac{1}{\kappa_{\mathcal{U} \cup \mathcal{A}}}.$$

Since $C_F \leq 1$, $d \geq 1$ and $L \geq 1$, we have $\frac{1}{a^2} \leq \frac{C_F^2}{24dL} \leq \frac{1}{24}$, which leads to

$$\frac{1}{\kappa} \leq \frac{16}{\lambda^\dagger} + \left( \frac{8}{3\lambda^\dagger} + 2 \right) \frac{1}{\kappa_{\mathcal{U} \cup \mathcal{A}}}.$$

Plugging $1/\kappa_{\mathcal{U} \cup \mathcal{A}} = 1 + 3/(2\kappa_{\mathcal{U}}) = 1 + 3/\lambda^\dagger$ from Lemma D.7 and D.8, we obtain

$$\frac{1}{\kappa} \leq \frac{16}{\lambda^\dagger} + \left( \frac{8}{3\lambda^\dagger} + 2 \right) \left( 1 + \frac{3}{\lambda^\dagger} \right)$$

$$\leq \frac{35}{\lambda^\dagger},$$

where we used $\frac{1}{\lambda^\dagger} \geq 1$ in the last inequality. $\quad\quad\quad\quad\quad\quad\quad\quad\quad\quad\quad\quad\quad\quad$ *Q.E.D*

## D.6 Log-Sobolev Inequality

Finally, we can establish the Log-Sobolev inequality thanks to the following theorem.

**Theorem D.5** (Cattiaux et al. (2010)). *Suppose the following conditions hold for the generator defined in Definition D.4.*

1. *There exist constants $\theta > 0$ and $b > 0$ and a $C^2$ function $V : \mathbb{R}^d \to [1, \infty)$ such that for all $x \in \mathbb{R}^d$*

$$\frac{\gamma \mathcal{L}V(x)}{V(x)} \leq -\theta \|x\|^2 + b.$$

2. *$\nu$ satisfies the Poincaré inequality with a constant $\kappa$.*

3. *There exists some constant $K > 0$, such that $\nabla^2 F \succeq -LI_{d \times d}$.*

*Then, $\nu$ satisfies the Log-Sobolev inequality with a constant $\alpha$ such that*

$$\frac{1}{\alpha} = C_1 + (C_2 + 2)\frac{1}{\kappa},$$

*where*

$$C_1 = \frac{2\gamma L}{\theta} + \frac{2}{\gamma K},$$

*and*

$$C_2 = \frac{2\gamma L}{\theta}\left(b + \theta\int_{\mathbb{R}^d}\|x\|^2\mathrm{d}\nu\right).$$

**Theorem D.6.** *Under Assumptions 1, 3 and 4 to 6, with $a > 0$ and $\gamma \geq 1$ such that*

$$a^2 \geq \frac{24dL}{C_F^2},$$

*and*

$$\gamma \geq \max\left(1, a^2\frac{4dL'^2}{\lambda^{\dagger^2}}, 4L'^2a^6\right),$$

*$\nu$ satisfies the Log-Sobolev inequality with constant $\alpha$ such that*

$$\frac{1}{\alpha} = \left(\frac{2M^2 + 8L^2}{M^2L} + \left(\frac{6L(d+1)}{M} + 2\right)\frac{35}{\lambda^\dagger}\right)\gamma.$$

*Proof.* Let us consider the candidate Lyapunov function $V(x) = \mathrm{e}^{M\gamma\|x\|^2/4}$. Then from $V \geq 1$ and Assumption 3, we obtain

$$\gamma\mathcal{L}V(x) = \left(\frac{M\gamma d}{2} + \frac{M^2\gamma^2}{4}\|x\|^2 - \frac{M\gamma^2}{2}\langle x, \nabla F(x)\rangle\right)V(x)$$

$$\leq \left(\frac{M\gamma(d+b\gamma)}{2} - \frac{M^2\gamma^2}{4}\|x\|^2\right)V(x).$$

Under

$$a^2 \geq \frac{24dL}{C_F^2},$$

and

$$\gamma \geq \max\left(a^2\frac{4dL'^2}{\lambda^{\dagger^2}}, 4L'^2a^6\right),$$

$\nu$ satisfies the Poincaré inequality with a constant $\frac{\lambda_*}{35}$. Moreover, since $F$ is $L$-smooth, $\nabla^2 F \succeq -LI_{d\times d}$. Therefore, all the conditions of Theorem D.5 are satisfied.

We conclude that from Theorem D.5, $\nu$ satisfies the Log-Sobolev inequality with a constant $\alpha$ such that

$$\frac{1}{\alpha} \leq C_1 + (C_2 + 2)\frac{35}{\lambda^\dagger},$$

where constants $C_1$ and $C_2$ can be calculated as

$$C_1 = \frac{2M^2 + 8L^2}{M^2L\gamma},$$

and

$$C_2 \leq \frac{6L(d+\gamma)}{M}$$

from Raginsky et al. (2017). We can replace this value by a simple upper bound which gives us

$$\frac{1}{\alpha} \leq \left(\frac{2M^2 + 8L^2}{M^2L} + \left(\frac{6L(d+1)}{M} + 2\right)\frac{35}{\lambda^\dagger}\right)\gamma$$

since $\gamma \geq 1$.

$$Q.E.D$$

**Remark D.1.** *Once we obtain the Poincaré constant, they are several ways to construct the Log-Sobolev constant. Another approach is possible, maybe simpler, by proceeding as Li and Erdogdu (2020) did in their analysis. Even though their method is interesting, this should not seriously change our main point since we just wanted to show that a polynomial dependence of the Log-Solev constant on the inverse temperature was achievable under certain additional conditions.*

# E  Analysis of an annealing scheme

In this Appendix, we prove the global convergence of SVRG-LD and SARAH-LD combined with an annealed scheme.

## E.1  Algorithm

In the context of optimization, we can use Algorithm 1 by setting a $\gamma$ huge enough so that the stationary distribution concentrates on the global minimizer of $F$. On the other hand, we can also introduce to SVRG-LD and SARAH-LD an increasing inverse temperature and a decreasing step size as follows.

---

**Algorithm 2:** SVRG-LD / SARAH-LD with annealing

---
**1** input: batch size $B$, epoch length $m$
**2** annealing schedule: step size $\eta_s > 0$ and inverse temperature $\gamma_s \geq 1$
**3** initialization: $X_0 = 0$, $X^{(0)} = X_0$
**4** **foreach** $s = 0, 1, \ldots, (K/m)$ **do**
**5**    $\quad v_{sm} = \nabla F(X^{(s)})$
**6**    $\quad$ randomly draw $\epsilon_{sm} \sim N(0, I_{d \times d})$
**7**    $\quad X_{sm+1} = X_{sm} - \eta_s v_{sm} + \sqrt{2\eta_s/\gamma_s}\,\epsilon_{sm}$
**8**    $\quad$ **foreach** $l = 1, \ldots, m-1$ **do**
**9**       $\quad\quad k = sm + l$
**10**      $\quad\quad$ randomly pick a subset $I_k$ from $\{1, \ldots, n\}$ of size $|I_k| = B$
**11**      $\quad\quad$ randomly draw $\epsilon_k \sim N(0, I_{d \times d})$
**12**      $\quad\quad$ **if** *SVRG-LD* **then**
**13**         $\quad\quad\quad v_k = \frac{1}{B}\sum_{i_k \in I_k}(\nabla f_{i_k}(X_k) - \nabla f_{i_k}(X^{(s)})) + v_{sm}$
**14**      $\quad\quad$ **else if** *SARAH-LD* **then**
**15**         $\quad\quad\quad v_k = \frac{1}{B}\sum_{i_k \in I_k}(\nabla f_{i_k}(X_k) - \nabla f_{i_k}(X_{k-1})) + v_{k-1}$
**16**      $\quad\quad$ **end**
**17**      $\quad\quad X_{k+1} = X_k - \eta_s v_k + \sqrt{2\eta_s/\gamma_s}\,\epsilon_k$
**18**   $\quad$ **end**
**19**   $\quad X^{(s+1)} = X_{(s+1)m}$
**20** **end**

---

**Definition 2.** *We define $\psi_k$ as the distribution of $X_k$ generated at the kth step of Algorithm 2.*

## E.2  Preparation for the Proof

Let us first establish some special notations to keep the proof clear and simple.

**Notation E.1.** *We define $\nu_{\gamma_k}$ as the stationary Gibbs distribution of SDE (1) when the inverse temperature parameter is set at $\gamma_k$, namely,*

$$\nu_{\gamma_k} := \mathrm{e}^{-\gamma_k F}/Z_{\gamma_k},$$

*where $Z_{\gamma_k}$ is the normalizing constant, and $\alpha_k$ as the Log-Sobolev constant of $\nu_{\gamma_k}$ under Assumptions 1 and 3. We also abbreviate the KL divergence between the distribution $\psi_{sm+r}$ of the random variable $X_{sm+r}$ generated by Algorithm 3 and the Gibbs distribution $\nu_{\gamma_s}$ as follows, where $s \in \mathbb{N} \cup \{0\}$ and $r = 1, \ldots, m$:*

$$H_{sm+r} := H_{\nu_{\gamma_s}}(\psi_{sm+r}).$$

*Moreover, $H_0 := H_{\nu_{\gamma_0}}(\psi_0)$.*

We will also need the following technical lemma.

**Lemma E.1.** *For all $s \in \mathbb{N} \cup \{0\}$, $\sigma \geq 3$ and $\mu > 2$,*

$$\left(\frac{2}{3}\right)^{\frac{2}{\mu}}(s+1)^{1-\frac{2}{\mu}}\sigma^{-\frac{2}{\mu}} \leq \sum_{i=0}^{s}(i+\sigma)^{-\frac{2}{\mu}},$$

*where $C_\mu$ is a constant independent of $s$ and $\sigma$.*

*Proof.* By a simple argument of area under the curve $y = x^{-\frac{2}{\mu}}$,

$$\sum_{i=0}^{s}(i+\sigma)^{-\frac{2}{\mu}} \geq \int_{\sigma}^{s+\sigma+1} x^{-\frac{2}{\mu}}\mathrm{d}x.$$

According to the mean value theorem for integrals, there exist a constant $c_s \in [\sigma, s+\sigma+1]$ such that

$$\sum_{i=0}^{s}(i+\sigma)^{-\frac{2}{\mu}} \geq \int_{\sigma}^{s+\sigma+1} x^{-\frac{2}{\mu}}\mathrm{d}x = c_s^{-\frac{2}{\mu}}(s+1).$$

We have also

$$\begin{aligned}
c_s^{-\frac{2}{\mu}} &\geq (s+1+\sigma)^{-\frac{2}{\mu}} \\
&= (s+1)^{-\frac{2}{\mu}}(1+\frac{\sigma}{s+1})^{-\frac{2}{\mu}} \\
&\geq (s+1)^{-\frac{2}{\mu}}(1+\sigma)^{-\frac{2}{\mu}} \\
&= (s+1)^{-\frac{2}{\mu}}\sigma^{-\frac{2}{\mu}}\left(\frac{1+\sigma}{\sigma}\right)^{-\frac{2}{\mu}} \\
&\geq (s+1)^{-\frac{2}{\mu}}\sigma^{-\frac{2}{\mu}}\left(\frac{2}{3}\right)^{\frac{2}{\mu}}.
\end{aligned}$$

In the last inequality, we used $\sigma \geq 2$. This implies the inequality of the statement.

$$Q.E.D$$

Now, considering that we only change the step size and the inverse temperature parameter at the beginning of every inner loop, all statements proved in Appendix A (Lemmas A.3 and A.4) and in Appendix B (Lemmas B.1, B.2 and B.3) that consider only the inner loop hold for Algorithm 2 as well.

Moreover, let us consider the annealing schedule

$$\eta_s = \bar{\eta}(s+\sigma)^{-\frac{1}{\mu}}, \tag{16}$$

$$\gamma_s = \bar{\gamma}\log\left\{g(s+\sigma)^{\frac{1}{\mu}}\right\}, \tag{17}$$

where we suppose $\bar{\eta} > 0$, $\bar{\gamma} > 0$, $\sigma \geq 3$, $g \geq e$ and $\mu > 2$. This annealing schedule is chosen on the one hand so that $\sum_{i=0}^{s}\frac{\alpha_i}{\gamma_i}\eta_i$ is explicitly computable, and on the other hand because Chiang et al. (1987) showed that the annealed continuous time GLD

$$\mathrm{d}X_t^{\mathrm{Ann}} = \nabla F(X_t^{\mathrm{Ann}})\mathrm{d}t + \sqrt{T(t)}\mathrm{d}B_t$$

could find the global minimum with the annealing schedule $T(t) \propto \frac{1}{\log t}$, which corresponds to equation (17).

Then, the following theorem holds under this annealing schedule.

**Theorem E.1.** *With the annealing schedule* (16) *and* (17), *under Assumptions 1 and 3,* $0 < \bar{\eta} < \frac{C_1}{16\sqrt{6}gL^2m}$, $\bar{\gamma} = \frac{1}{C_2}$, $\mu > 2$, $g \geq$ e, *and* $B \geq m$, *for all* $k = sm + r$ *where* $s \in \mathbb{N} \cup \{0\}$ *and* $r = 0, \ldots, m-1$, *the following holds in the update of Algorithm 2:*

$$\begin{aligned}
H_{\nu_{\gamma_s}}(\psi_{sm+r+1}) \leq{} & \mathrm{e}^{-\frac{3\alpha_s}{2\gamma_s}\eta_s}\left(1 + \frac{32\gamma_s L^4\eta_s^3}{\alpha_s}\right)H_{\nu_{\gamma_s}}(\psi_{sm+r}) \\
& + \mathrm{e}^{-\frac{3\alpha_s}{2\gamma_s}\eta_s}\sum_{i=0}^{r-1}\frac{128\gamma_s L^4\eta_s^3}{\alpha_s}\mathrm{e}^{-\frac{\alpha_s m}{\gamma_s}\eta_s}H_{\nu_{\gamma_s}}(\psi_{sm+i}) \\
& + 56\eta_s^2 dL^2.
\end{aligned}$$

*Here,* $C = \frac{(n-B)}{B(n-1)}$.

*Proof.* From Property C.3, $\nu_{\gamma_s}$ satisfies Log-Sobolev inequality with a constant $\alpha_s$ such that $\frac{\alpha_s}{\gamma_s} = C_1 e^{-C_2 \gamma_s}$. It thus suffices to notice that under $0 < \bar{\eta} < \frac{C_1}{16\sqrt{6}gL^2m}$ and $\bar{\gamma} = \frac{1}{C_2}$, we have for all $s \in \mathbb{N} \cup \{0\}$,

$$\eta_s = \bar{\eta}(s+\sigma)^{-\frac{1}{\mu}}$$
$$\leq \frac{C_1}{16\sqrt{6}g(s+\sigma)^{\frac{1}{\mu}}L^2m}$$
$$= \frac{\alpha_s}{16\sqrt{6}\gamma_s L^2 m}.$$

In the inequality, we used the fact that with $\bar{\gamma} = \frac{1}{C_2}$,

$$\frac{\alpha_s}{\gamma_s} = C_1 e^{-C_2\gamma_s}$$
$$= C_1 e^{-C_2\bar{\gamma}\log\left\{g(s+\sigma)^{\frac{1}{\mu}}\right\}}$$
$$= \frac{C_1}{g(s+\sigma)^{\frac{1}{\mu}}}.$$

Therefore, all the assumptions of Theorem A.1 and B.1 are satisfied. From the proof of each theorem, we immediately obtain the inequality of the statement from equations (9) and (14).

$$Q.E.D$$

The problem with changing the inverse temperature parameter of each inner loop is that we cannot immediately give an upper bound for each $H_k$ as Theorem A.2 and B.2. The main challenge resides in linking $H_{\nu_{\gamma_s}}(\psi_{sm})$ and $H_{\nu_{\gamma_{s-1}}}(\psi_{sm})$, which corresponds to the shift of optimization trajectory in the space of measures generated by the change of inverse temperature parameter at the beginning of each inner loop. The following lemma suggests that a small enough difference between two consecutive inverse temperatures will solve this issue.

**Lemma E.2.** *Under Assumptions 1, 3 and $F \geq 0$, for all $s \in \mathbb{N}$ and $\gamma_0 \geq \frac{2}{M}$,*

$$H_{\nu_{\gamma_s}}(\psi_{sm}) \leq \left(1 + \Delta\gamma_s \frac{2L}{\alpha_{s-1}}\right) H_{\nu_{\gamma_{s-1}}}(\psi_{sm}) + \Delta\gamma_s\left(\chi + F(X^*)\right).$$

*Here, $\Delta\gamma_s := \gamma_s - \gamma_{s-1}$, $\chi := \max_{\gamma \geq 1}\left\{\frac{d}{\gamma}\log\left(\frac{eL}{M}\left(\frac{b\gamma}{d} + 1\right)\right)\right\}$ and $X^*$ is the global minimum of $F$.*

*Proof.*

$$H_{\nu_{\gamma_s}}(\psi_{sm}) = H_{\nu_{\gamma_{s-1}}}(\psi_{sm}) + H_{\nu_{\gamma_s}}(\psi_{sm}) - H_{\nu_{\gamma_{s-1}}}(\psi_{sm})$$
$$= H_{\nu_{\gamma_{s-1}}}(\psi_{sm}) + \int \psi_{sm}\log\frac{\psi_{sm}}{\nu_{\gamma_s}}dz - \int \psi_{sm}\log\frac{\psi_{sm}}{\nu_{\gamma_{s-1}}}dz$$
$$= H_{\nu_{\gamma_{s-1}}}(\psi_{sm}) + \int \psi_{sm}\log\frac{\nu_{\gamma_{s-1}}}{\nu_{\gamma_s}}dz$$
$$= H_{\nu_{\gamma_{s-1}}}(\psi_{sm}) + \int \psi_{sm}\log\frac{e^{-\gamma_{s-1}F}/Z_{\gamma_{s-1}}}{e^{-\gamma_s F}/Z_{\gamma_s}}dz$$
$$= H_{\nu_{\gamma_{s-1}}}(\psi_{sm}) + \int \psi_{sm}(\gamma_s - \gamma_{s-1})Fdz + \log\frac{Z_{\gamma_s}}{Z_{\gamma_{s-1}}}.$$

Here, as $\gamma_s \geq \gamma_{s-1}$ and $F \geq 0$, we have that

$$-\gamma_s F \leq -\gamma_{s-1}F,$$

which means

$$Z_{\gamma_s} \leq Z_{\gamma_{s-1}}.$$

Thus,

$$H_{\nu_{\gamma_s}}(\psi_{sm}) \leq H_{\nu_{\gamma_{s-1}}}(\psi_{sm}) + \Delta\gamma_s \mathbb{E}_{X \sim \psi_{sm}}[F(X)].$$

Now, from Corollary C.1.1 and Theorem C.2, we know that

$$\begin{aligned}
\mathbb{E}_{X \sim \psi_{sm}}[F(X)] &= \mathbb{E}_{X \sim \psi_{sm}}[F(X)] - F(X^*) + F(X^*) \\
&\leq LW_2^2(\psi_k, \nu_{\gamma_{s-1}}) + 2\left(\mathbb{E}_{X \sim \nu_{\gamma_{s-1}}}[F(X)] - F(X^*)\right) + F(X^*) \\
&\leq LW_2^2(\psi_k, \nu_{\gamma_{s-1}}) + \frac{d}{\gamma_{s-1}}\log\left(\frac{eL}{M}\left(\frac{b\gamma_{s-1}}{d} + 1\right)\right) + F(X^*) \\
&\leq \frac{2L}{\alpha_{s-1}}H_{\nu_{\gamma_{s-1}}}(\psi_{sm}) + \chi + F(X^*).
\end{aligned}$$

We used Corollary C.1.1 at the first inequality, Theorem C.2 at the second inequality and Talagrand's inequality at the last inequality. This gives the desired result.

$$Q.E.D$$

Since the logarithmic function $\log x$ is strictly increasing while its derivative decreases according to $x$, we can find an adequate bound of $\sigma$ to assure that $\Delta\gamma_s$ is small enough. As a reminder, we set $\gamma_s = \bar{\gamma}\log\left\{g(s+\sigma)^{\frac{1}{\mu}}\right\}$ and $\eta_s = \bar{\eta}(s+\sigma)^{\frac{1}{\mu}}$.

**Lemma E.3.** *With the annealing (16) and (17), when $\alpha_s = \gamma_s C_1 e^{-C_2\gamma_s}$,*

$$\sigma \geq 3 \vee \left(\frac{8Lg^2}{C_1^2\bar{\eta}}\right)^{\frac{\mu}{\mu-3}} \vee \left(\frac{2}{\mu C_2 L^2\bar{\eta}^2}\right)^{\frac{\mu}{\mu-2}},$$

*$\bar{\gamma} = \frac{1}{C_2}$, $\mu > 3$ and $g \geq$ e, we have*

$$\Delta\gamma_s\frac{2L}{\alpha_{s-1}} \leq \frac{\alpha_s\eta_s}{2\gamma_s}, \tag{18}$$

*and*

$$\Delta\gamma_s \leq \eta_s^2 L^2 \leq \frac{1}{4} \tag{19}$$

*for all $s \in \mathbb{N} \cup \{0\}$.*

*Proof.* First of all, by the mean value theorem, there exists a $c \in [s-1, s]$ such that,

$$\Delta\gamma_s = \frac{\bar{\gamma}/\mu}{c+\sigma}.$$

Thus,

$$\Delta\gamma_s = \frac{\bar{\gamma}/\mu}{c+\sigma} \leq \frac{\bar{\gamma}/\mu}{s-1+\sigma} = \frac{1}{\mu C_2}\frac{1}{(s-1+\sigma)}.$$

Therefore, in order to satisfy inequality (18), it suffices to have

$$\begin{aligned}
\frac{1}{\mu C_2}\frac{1}{(s-1+\sigma)} &\leq \frac{\alpha_{s-1}}{2L}\frac{\alpha_s\eta_s}{2\gamma_s} \\
&= \frac{\gamma_{s-1}C_1 e^{-C_2\gamma_{s-1}}}{2L}\frac{1}{2}C_1 e^{-C_2\gamma_s}\eta_s.
\end{aligned}$$

A sufficient condition to this is

$$\frac{1}{\mu C_2}\frac{1}{(s-1+\sigma)} \leq \frac{C_1 C_2^{-1}\log\left\{g(s-1+\sigma)^{\frac{1}{\mu}}\right\}}{2Lg(s+\sigma)^{\frac{1}{\mu}}}\frac{\bar{\eta}C_1}{2g(s+\sigma)^{\frac{2}{\mu}}},$$

which gives

$$\frac{4Lg^2}{C_1^2\bar{\eta}} \leq \frac{s+\sigma-1}{(s+\sigma)^{\frac{3}{\mu}}} \log\{g(s-1+\sigma)\}$$

$$= (s+\sigma)^{1-\frac{3}{\mu}}\frac{s+\sigma-1}{s+\sigma} \log\{g(s-1+\sigma)\}$$

$$= (s+\sigma)^{1-\frac{3}{\mu}}\left(1-\frac{1}{s+\sigma}\right) \log\{g(s-1+\sigma)\}.$$

As $\log\{g(s-1+\sigma)\} \geq 1$ and $1 - \frac{1}{s+\sigma} \geq \frac{1}{2}$ when $g \geq$ e, $s \geq 0$ and $\sigma \geq 2$, it suffices to have the following inequality satisfied when $s = 0$:

$$\frac{8L}{C_1^2\bar{\eta}} \leq (s+\sigma)^{1-\frac{3}{\mu}}.$$

From this, we obtain $\sigma \geq \left(\frac{8Lg^2}{C_1^2\bar{\eta}}\right)^{\frac{\mu}{\mu-3}}$.

Likewise, in order to satisfy inequality (19), it suffices to have

$$\frac{1}{\mu C_2}\frac{1}{(s-1+\sigma)} \leq \frac{\bar{\eta}^2 L^2}{(s+\sigma)^{\frac{2}{\mu}}},$$

which gives

$$\frac{1}{\mu C_2 L^2\bar{\eta}^2} \leq \frac{s+\sigma-1}{(s+\sigma)^{\frac{2}{\mu}}}.$$

It thus suffices to have the following inequality satisfied when $s = 0$:

$$\frac{2}{\mu C_2 L^2\bar{\eta}^2} \leq (s+\sigma)^{1-\frac{2}{\mu}}.$$

This gives $\sigma \geq \left(\frac{2}{\mu C_2 L^2\bar{\eta}^2}\right)^{\frac{\mu}{\mu-2}}$.

The last inequality $\eta_s^2 L^2 \leq \frac{1}{2}$ is immediately satisfied with $\eta_s \leq \bar{\eta} \leq \frac{1}{4L}$.

$$Q.E.D$$

### E.3 Main Proof

We are now ready to prove the main results. We first evaluate how $H_k$ decreases compared with the previous step.

**Theorem E.2.** *With the annealing schedule* (16) *and* (17), *under Assumptions 1, 3 and $F \geq 0$, $0 < \bar{\eta} < \frac{C_1}{16\sqrt{6}gL^2m}$, $\bar{\gamma} = \frac{1}{C_2}$, $B \geq m$, $\mu > 3$, $g \geq$ e and*

$$\sigma \geq 3 \vee \left(\frac{8Lg^2}{C_1^2\bar{\eta}}\right)^{\frac{\mu}{\mu-3}} \vee \left(\frac{2}{\mu C_2 L^2\bar{\eta}^2}\right)^{\frac{\mu}{\mu-2}},$$

*for all $k = sm + r$ where $s \in \mathbb{N} \cup \{0\}$ and $r = 0, \ldots, m-1$, the following holds in the update of Algorithm 2:*

$$H_{sm+r+1} \leq e^{-\frac{\alpha_s}{\gamma_s}\eta_s}\left(1+\frac{\alpha_s}{4\gamma_s}\eta_s\right)H_{sm+r}$$

$$+ e^{-\frac{\alpha_s}{\gamma_s}\eta_s}\sum_{i=0}^{r-1}\frac{\alpha_s}{4m\gamma_s}\eta_s e^{-\frac{\alpha_s m}{\gamma_s}\eta_s}H_{sm+i}$$

$$+ \eta_s^2 dL^2 E.$$

*Here, $E = 56 + 2\chi + 2F(X^*)$*

*Proof.* When $r = 0$, from Theorem E.1, we have

$$H_{\nu_{\gamma_s}}(\psi_{sm+1}) \le e^{-\frac{3\alpha_s}{2\gamma_s}\eta_s}\left(1 + \frac{32\gamma_s L^4 \eta_s^3}{\alpha_s}\right) H_{\nu_{\gamma_s}}(\psi_{sm})$$
$$+ 56\eta_s^2 dL^2.$$

Under

$$\sigma \ge 3 \vee \left(\frac{8Lg^2}{C_1^2 \bar{\eta}}\right)^{\frac{\mu}{\mu-3}} \vee \left(\frac{2}{\mu C_2 L^2 \bar{\eta}^2}\right)^{\frac{\mu}{\mu-2}},$$

we can derive the following bound:

$$H_{\nu_{\gamma_s}}(\psi_{sm+1}) \le e^{-\frac{3\alpha_s}{2\gamma_s}\eta_s}\left(1 + \frac{32\gamma_s L^4 \eta_s^3}{\alpha_s}\right)\left(1 + \Delta\gamma_s \frac{2L}{\alpha_{s-1}}\right) H_{\nu_{\gamma_{s-1}}}(\psi_{sm})$$
$$+ e^{-\frac{3\alpha_s}{2\gamma_s}\eta_s}\left(1 + \frac{32\gamma_s L^4 \eta_s^3}{\alpha_s}\right)\Delta\gamma_s\left(\chi + F(X^*)\right)$$
$$+ 56\eta_s^2 dL^2$$
$$\le e^{-\frac{3\alpha_s}{2\gamma_s}\eta_s}\left(1 + \frac{32\gamma_s L^4 \eta_s^3}{\alpha_s}\right)\left(1 + \frac{\alpha_s}{2\gamma_s}\eta_s\right) H_{\nu_{\gamma_{s-1}}}(\psi_{sm})$$
$$+ \left(1 + 2L^2\eta_s^2\right)\eta_s^2 L^2\left(\chi + F(X^*)\right)$$
$$+ 56\eta_s^2 dL^2$$
$$\le e^{-\frac{3\alpha_s}{2\gamma_s}\eta_s}\left(1 + \frac{32\gamma_s L^4 \eta_s^3}{\alpha_s}\right)e^{\frac{\alpha_s}{2\gamma_s}\eta_s} H_{\nu_{\gamma_{s-1}}}(\psi_{sm})$$
$$+ (1+1)\eta_s^2 L^2\left(\chi + F(X^*)\right)$$
$$+ 56\eta_s^2 dL^2$$
$$\le e^{-\frac{\alpha_s}{\gamma_s}\eta_s}\left(1 + \frac{32\gamma_s L^4 \eta_s^3}{\alpha_s}\right) H_{\nu_{\gamma_{s-1}}}(\psi_{sm})$$
$$+ \eta_s^2 dL^2\left(56 + 2\chi + 2F(X^*)\right).$$

We used Lemma E.2 at the first inequality, Lemma E.3 and $\eta_s \le \frac{\alpha_s}{16\gamma_s L^2 m}$ at the second inequality and $\eta_s^2 L^2 \le \frac{1}{2}$ at the last inequality.

When $r \ge 1$, from Theorem E.1, we have

$$H_{\nu_{\gamma_s}}(\psi_{sm+r+1}) \le e^{-\frac{3\alpha_s}{2\gamma_s}\eta_s}\left(1 + \frac{32\gamma_s L^4 \eta_s^3}{\alpha_s}\right) H_{\nu_{\gamma_s}}(\psi_{sm+r})$$
$$+ e^{-\frac{3\alpha_s}{2\gamma_s}\eta_s}\sum_{i=0}^{r-1}\frac{128\gamma_s L^4 \eta_s^3}{\alpha_s}e^{-\frac{\alpha_s m}{\gamma_s}\eta_s} H_{\nu_{\gamma_s}}(\psi_{sm+i})$$
$$+ 56\eta_s^2 dL^2.$$

Under

$$\sigma \ge 3 \vee \left(\frac{8Lg^2}{C_1^2 \bar{\eta}}\right)^{\frac{\mu}{\mu-3}} \vee \left(\frac{2}{\mu C_2 L^2 \bar{\eta}^2}\right)^{\frac{\mu}{\mu-2}},$$

$$H_{\nu_{\gamma_s}}(\psi_{sm+r+1}) \leq e^{-\frac{3\alpha_s}{2\gamma_s}\eta_s} \left(1 + \frac{32\gamma_s L^4 \eta_s^3}{\alpha_s}\right) H_{\nu_{\gamma_s}}(\psi_{sm+r})$$

$$+ e^{-\frac{3\alpha_s}{2\gamma_s}\eta_s} \sum_{i=1}^{r-1} \frac{128\gamma_s L^4 \eta_s^3}{\alpha_s} e^{-\frac{\alpha_s m}{\gamma_s}\eta_s} H_{\nu_{\gamma_s}}(\psi_{sm+i})$$

$$+ e^{-\frac{3\alpha_s}{2\gamma_s}\eta_s} \frac{128\gamma_s L^4 \eta_s^3}{\alpha_s} e^{-\frac{\alpha_s m}{\gamma_s}\eta_s} H_{\nu_{\gamma_s}}(\psi_{sm})$$

$$+ 56\eta_s^2 dL^2$$

$$\leq e^{-\frac{3\alpha_s}{2\gamma_s}\eta_s} \left(1 + \frac{32\gamma_s L^4 \eta_s^3}{\alpha_s}\right) H_{\nu_{\gamma_s}}(\psi_{sm+r})$$

$$+ e^{-\frac{3\alpha_s}{2\gamma_s}\eta_s} \sum_{i=1}^{r-1} \frac{128\gamma_s L^4 \eta_s^3}{\alpha_s} e^{-\frac{\alpha_s m}{\gamma_s}\eta_s} H_{\nu_{\gamma_s}}(\psi_{sm+i})$$

$$+ e^{-\frac{3\alpha_s}{2\gamma_s}\eta_s} \frac{128\gamma_s L^4 \eta_s^3}{\alpha_s} e^{-\frac{\alpha_s m}{\gamma_s}\eta_s} \left(1 + \Delta\gamma_s \frac{2L}{\alpha_{s-1}}\right) H_{\nu_{\gamma_{s-1}}}(\psi_{sm})$$

$$+ e^{-\frac{3\alpha_s}{2\gamma_s}\eta_s} \frac{128\gamma_s L^4 \eta_s^3}{\alpha_s} e^{-\frac{\alpha_s m}{\gamma_s}\eta_s} \Delta\gamma_s \left(\chi + F(X^*)\right)$$

$$+ 56\eta_s^2 dL^2$$

$$\leq e^{-\frac{3\alpha_s}{2\gamma_s}\eta_s} \left(1 + \frac{32\gamma_s L^4 \eta_s^3}{\alpha_s}\right) H_{\nu_{\gamma_s}}(\psi_{sm+r})$$

$$+ e^{-\frac{3\alpha_s}{2\gamma_s}\eta_s} \sum_{i=1}^{r-1} \frac{128\gamma_s L^4 \eta_s^3}{\alpha_s} e^{-\frac{\alpha_s m}{\gamma_s}\eta_s} H_{\nu_{\gamma_s}}(\psi_{sm+i})$$

$$+ e^{-\frac{\alpha_s}{\gamma_s}\eta_s} \frac{128\gamma_s L^4 \eta_s^3}{\alpha_s} e^{-\frac{\alpha_s m}{\gamma_s}\eta_s} H_{\nu_{\gamma_{s-1}}}(\psi_{sm})$$

$$+ 2\eta_s^2 L^2 \Delta\gamma_s \left(\chi + F(X^*)\right) + 56\eta_s^2 dL^2$$

$$\leq e^{-\frac{\alpha_s}{\gamma_s}\eta_s} \left(1 + \frac{32\gamma_s L^4 \eta_s^3}{\alpha_s}\right) H_{\nu_{\gamma_s}}(\psi_{sm+r})$$

$$+ e^{-\frac{\alpha_s}{\gamma_s}\eta_s} \sum_{i=0}^{r-1} \frac{128\gamma_s L^4 \eta_s^3}{\alpha_s} e^{-\frac{\alpha_s m}{\gamma_s}\eta_s} H_{sm+i}$$

$$+ \eta_s^2 dL^2 \left(56 + 2\chi + 2F(X^*)\right).$$

Therefore, for all $r = 0, \ldots, m-1$,

$$H_{sm+r+1} \leq e^{-\frac{\alpha_s}{\gamma_s}\eta_s} \left(1 + \frac{32\gamma_s L^4 \eta_s^3}{\alpha_s}\right) H_{sm+r}$$

$$+ e^{-\frac{\alpha_s}{\gamma_s}\eta_s} \sum_{i=0}^{r-1} \frac{128\gamma_s L^4 \eta_s^3}{\alpha_s} e^{-\frac{\alpha_s m}{\gamma_s}\eta_s} H_{sm+i}$$

$$+ \eta_s^2 dL^2 \left(56 + 2\chi + 2F(X^*)\right)$$

$$\leq e^{-\frac{\alpha_s}{\gamma_s}\eta_s} \left(1 + \frac{\alpha_s}{4\gamma_s}\eta_s\right) H_{sm+r}$$

$$+ e^{-\frac{\alpha_s}{\gamma_s}\eta_s} \sum_{i=0}^{r-1} \frac{\alpha_s}{4m\gamma_s}\eta_s e^{-\frac{\alpha_s m}{\gamma_s}\eta_s} H_{sm+i}$$

$$+ \eta_s^2 dL^2 E.$$

In the last inequality, we used $\eta_s \leq \frac{\alpha_s}{16\sqrt{2}L^2 m\gamma_s}$.

Q.E.D

**Theorem E.3.** *With the annealing schedule* (16) *and* (17), *under Assumptions* 1, 3 *and* $F \geq 0$, $0 < \bar{\eta} < \frac{C_1}{16\sqrt{6}gL^2m}$, $B \geq m$, $\mu > 3$, $\bar{\gamma} = \frac{1}{C_2}$, $g \geq e$ *and*

$$\sigma \geq 3 \vee \left(\frac{8Lg^2}{C_1^2\bar{\eta}}\right)^{\frac{\mu}{\mu-3}} \vee \left(\frac{2}{\mu C_2 L^2 \bar{\eta}^2}\right)^{\frac{\mu}{\mu-2}},$$

*for all* $k = sm$ *where* $s \in \mathbb{N}$, *the following holds in the update of Algorithm 3:*

$$H_k \leq e^{-\frac{C_1\bar{\eta}}{2g}\left(\frac{2}{3}\right)^{\frac{2}{\mu}}k^{1-\frac{2}{\mu}}m^{\frac{2}{\mu}}\sigma^{-\frac{2}{\mu}}}H_0 + \frac{8}{3}\bar{\eta}dL^2EgC_1^{-1}k^{\frac{2}{\mu}}m^{-\frac{2}{\mu}}\sigma^{\frac{2}{\mu}},$$

*where* $E = 56 + 2\chi + 2F(X^*)$.

*Proof.* First of all, we will prove by mathematical induction that in each inner loop the following inequality holds for all $r = 0, \ldots, m-1$:

$$H_{sm+r+1} \leq e^{-\frac{\alpha_s}{2\gamma_s}\eta_s(r+1)}H_{sm} + \eta_s^2 dL^2 E \left(\sum_{i=0}^{r} e^{-\frac{\alpha_s}{2\gamma_s}\eta_s i}\right) \qquad \ldots(**)$$

When $r = 0$, from Theorem E.2, we have

$$H_{sm+1} \leq e^{-\frac{\alpha_s}{\gamma_s}\eta_s}\left(1 + \frac{\alpha_s}{4\gamma_s}\eta_s\right)H_{sm} + \eta_s^2 dL^2 E$$

$$\leq e^{-\frac{\alpha_s}{\gamma_s}\eta_s}\left(1 + \frac{\alpha_s}{2\gamma_s}\eta_s\right)H_{sm} + \eta_s^2 dL^2 E$$

$$\leq e^{-\frac{\alpha_s}{\gamma_s}\eta_s}e^{\frac{\alpha_s}{2\gamma_s}\eta_s}H_{sm} + \eta_s^2 dL^2 E$$

$$\leq e^{-\frac{\alpha_s}{2\gamma_s}\eta_s}H_{sm} + \eta_s^2 dL^2 E.$$

Thus, $(**)$ holds for $r = 0$.

Now, let us suppose that $(**)$ is true for all $r \leq l$. Then, when $r = l+1$ from Theorem E.2, we have

$$H_{sm+l+2} \leq e^{-\frac{\alpha_s}{\gamma_s}\eta_s}\left(1 + \frac{\alpha_s}{4\gamma_s}\eta_s\right)H_{sm+l+1} + e^{-\frac{\alpha_s}{\gamma_s}\eta_s}\sum_{i=0}^{l}\frac{\alpha_s}{4m\gamma_s}\eta_s e^{-\frac{\alpha_s m}{\gamma_s}\eta_s}H_{sm+i}$$

$$+ \eta_s^2 dL^2 E$$

$$\leq e^{-\frac{\alpha_s}{\gamma_s}\eta_s}\left(1 + \frac{\alpha_s}{4\gamma_s}\eta_s\right)\left(e^{-\frac{\alpha_s}{2\gamma_s}\eta_s(l+1)}H_{sm} + \eta_s^2 dL^2 E\left(\sum_{j=0}^{l}e^{-\frac{\alpha_s}{2\gamma_s}\eta_s j}\right)\right)$$

$$+ e^{-\frac{\alpha_s}{\gamma_s}\eta_s}\sum_{i=0}^{l}\frac{\alpha_s}{4m\gamma_s}\eta_s e^{-\frac{\alpha_s m}{\gamma_s}\eta_s}\left(e^{-\frac{\alpha_s}{2\gamma_s}\eta_s i}H_{sm} + \eta_s^2 dL^2 E\left(\sum_{j=0}^{i-1}e^{-\frac{\alpha_s}{2\gamma_s}\eta_s j}\right)\right)$$

$$+ \eta_s^2 dL^2 E$$

$$\leq e^{-\frac{\alpha_s}{\gamma_s}\eta_s}\left(1 + \frac{\alpha_s}{4\gamma_s}\eta_s\right)\left(e^{-\frac{\alpha_s}{2\gamma_s}\eta_s(l+1)}H_{sm} + \eta_s^2 dL^2 E\left(\sum_{j=0}^{l}e^{-\frac{\alpha_s}{2\gamma_s}\eta_s j}\right)\right)$$

$$+ e^{-\frac{\alpha_s}{\gamma_s}\eta_s}\sum_{i=0}^{l}\frac{\alpha_s}{4m\gamma_s}\eta_s\left(e^{-\frac{\alpha_s\eta_s}{2\gamma_s}(l+1)}H_{sm} + \eta_s^2 dL^2 E\left(\sum_{j=0}^{l}e^{-\frac{\alpha_s}{2\gamma_s}\eta_s j}\right)\right)$$

$$+ \eta_s^2 dL^2 E$$

$$\leq e^{-\frac{\alpha_s}{2\gamma_s}\eta_s(l+1)}e^{-\frac{\alpha_s}{\gamma_s}\eta_s}\left(1 + \frac{\alpha_s}{4\gamma_s}\eta_s + \sum_{i=0}^{l}\frac{\alpha_s}{4m\gamma_s}\eta_s\right)H_{sm}$$

$$+ \eta_s^2 dL^2 E e^{-\frac{\alpha_s}{\gamma_s}\eta_s}\left(1 + \frac{\alpha_s}{4\gamma_s}\eta_s + \sum_{i=0}^{l}\frac{\alpha_s}{4m\gamma_s}\eta_s\right)\left(\sum_{j=0}^{l}e^{-\frac{\alpha_s}{2\gamma_s}\eta_s j}\right)$$

$$+ \eta_s^2 dL^2 E.$$

This further implies,

$$H_{sm+l+2} \leq e^{-\frac{\alpha_s}{2\gamma_s}\eta_s(l+1)} e^{-\frac{\alpha_s}{\gamma_s}\eta_s} e^{\frac{\alpha_s}{2\gamma_s}\eta_s} H_{sm}$$

$$+ \eta_s^2 dL^2 E e^{-\frac{\alpha_s}{\gamma_s}\eta_s} e^{\frac{\alpha_s}{2\gamma_s}\eta_s} \left( \sum_{j=0}^{l} e^{-\frac{\alpha_s}{2\gamma_s}\eta_s j} \right) + \eta_s^2 dL^2 E$$

$$\leq e^{-\frac{\alpha_s}{2\gamma_s}\eta_s(l+2)} H_{sm} + \eta_s^2 dL^2 E \left( \sum_{i=0}^{l+1} e^{-\frac{\alpha_s}{2\gamma_s}\eta_s i} \right).$$

Therefore, $(**)$ holds for all inner loop and $r = 0, \ldots, m-1$.

Especially, when $r = m-1$, we obtain

$$H_{(s+1)m} \leq e^{-\frac{\alpha_s}{2\gamma_s}\eta_s m} H_{sm} + \eta_s^2 dL^2 E \left( \sum_{i=0}^{m-1} e^{-\frac{\alpha_s}{2\gamma_s}\eta_s i} \right).$$

Consecutively using this inequality, we obtain

$$H_{(s+1)m} \leq e^{-\frac{\alpha_s}{2\gamma_s}\eta_s m} H_{sm} + \eta_s^2 dL^2 E \left( \sum_{i=0}^{m-1} e^{-\frac{\alpha_s}{2\gamma_s}\eta_s i} \right)$$

$$\leq e^{-\frac{\alpha_s}{2\gamma_s}\eta_s m} \left( e^{-\frac{\alpha_{s-1}}{2\gamma_{s-1}}\eta_{s-1} m} H_{(s-1)m} + \eta_{s-1}^2 dL^2 E \left( \sum_{i=0}^{m-1} e^{-\frac{\alpha_{s-1}}{2\gamma_{s-1}}\eta_{s-1} i} \right) \right)$$

$$+ \eta_s^2 dL^2 E \left( \sum_{i=0}^{m-1} e^{-\frac{\alpha_s}{2\gamma_s}\eta_s i} \right)$$

$$= e^{-\frac{m}{2}\left( \frac{\alpha_s}{\gamma_s}\eta_s + \frac{\alpha_{s-1}}{\gamma_{s-1}}\eta_{s-1} \right)} H_{(s-1)m}$$

$$+ dL^2 E \left( \eta_s^2 \sum_{i=0}^{m-1} e^{-\frac{\alpha_s}{2\gamma_s}\eta_s i} + \eta_{s-1}^2 e^{-\frac{\alpha_s}{2\gamma_s}\eta_s m} \sum_{i=0}^{m-1} e^{-\frac{\alpha_{s-1}}{2\gamma_{s-1}}\eta_{s-1} i} \right)$$

$$\cdots$$

$$\leq e^{-\frac{m}{2}\sum_{i=0}^{s}\frac{\alpha_i}{\gamma_i}\eta_i} H_0$$

$$+ dL^2 E \sum_{i=0}^{s} \left( \eta_i^2 e^{-\frac{m}{2}\sum_{j=i+1}^{s}\frac{\alpha_j}{\gamma_j}\eta_j} \sum_{j=0}^{m-1} e^{-\frac{\alpha_i}{2\gamma_i}\eta_i j} \right),$$

which implies

$$H_{(s+1)m} \leq e^{-\frac{m}{2}\sum_{i=0}^{s}\frac{\alpha_i}{\gamma_i}\eta_i} H_0 + dL^2 E \sum_{i=0}^{s} \left( \eta_i^2 e^{-\frac{m}{2}\sum_{j=i+1}^{s}\frac{\alpha_s}{\gamma_s}\eta_s} \sum_{j=0}^{m-1} e^{-\frac{\alpha_s}{2\gamma_s}\eta_s j} \right)$$

$$\leq e^{-\frac{m}{2}\sum_{i=0}^{s}\frac{\alpha_i}{\gamma_i}\eta_i} H_0 + \bar{\eta}^2 dL^2 E \sum_{i=0}^{\infty} e^{-\frac{\alpha_s}{2\gamma_s}\eta_s i}$$

$$= e^{-\frac{m}{2}\sum_{i=0}^{s}\frac{\alpha_i}{\gamma_i}\eta_i} H_0 + \bar{\eta}^2 dL^2 E \left( 1 - e^{-\frac{\alpha_s}{2\gamma_s}\eta_s} \right)^{-1}$$

$$\leq e^{-\frac{m}{2}\sum_{i=0}^{s}\frac{\alpha_i}{\gamma_i}\eta_i} H_0 + \bar{\eta}^2 dL^2 E \left( \frac{3}{4}\frac{\alpha_s}{2\gamma_s}\eta_s \right)^{-1}$$

$$= e^{-\frac{mC_1\bar{\eta}}{2g}\sum_{i=0}^{s}(i+\sigma)^{-\frac{2}{\mu}}} H_0 + \frac{8}{3}\bar{\eta}dL^2 Eg C_1^{-1}(s+\sigma)^{\frac{2}{\mu}}$$

$$\leq e^{-\frac{C_1\bar{\eta}}{2g}\left( \frac{2}{3} \right)^{\frac{2}{\mu}} m(s+1)^{1-\frac{2}{\mu}}\sigma^{-\frac{2}{\mu}}} H_0 + \frac{8}{3}\bar{\eta}dL^2 Eg C_1^{-1}(s+1)^{\frac{2}{\mu}}\sigma^{\frac{2}{\mu}}.$$

For the first inequality, we used $\frac{\alpha_i}{\gamma_i}\eta_i \geq \frac{\alpha_s}{\gamma_s}\eta_s$ for all $i \leq s$, for the third inequality, we used $1 - e^{-c} \geq \frac{3}{4}c$ holds for all $0 < c = \frac{\alpha_s}{2\gamma_s}\eta_s \leq \frac{1}{4}$, and for the last inequality, we used Lemma E.1.

Setting $k = (s+1)m$, we obtain

$$H_k \leq \mathrm{e}^{-\frac{C_1\bar{\eta}}{2g}\left(\frac{2}{3}\right)^{\frac{2}{\mu}} k^{1-\frac{2}{\mu}} m^{\frac{2}{\mu}} \sigma^{-\frac{2}{\mu}}} H_0$$
$$+ \frac{8}{3}\bar{\eta} dL^2 E g C_1^{-1} k^{\frac{2}{\mu}} m^{-\frac{2}{\mu}} \sigma^{\frac{2}{\mu}}.$$

$$Q.E.D$$

Finally, we obtain the following global convergence guarantee for Algorithm 2.

**Theorem E.4.** *Using Algorithm 2 with the annealing schedule $\eta_s = \bar{\eta}(s + \sigma)^{-\frac{1}{\mu}}$ and $\gamma_s = \bar{\gamma} \log\left\{g(s+\sigma)^{\frac{1}{\mu}}\right\}$, under Assumptions 1, 3 and $F \geq 0$, $0 < \bar{\eta} < \frac{C_1}{16\sqrt{6}gL^2m}$, $B \geq m$, $\epsilon = O\left(\frac{LH_0}{C_1C_2^{-1}}\right)$, $\mu \geq 13$, $g = \mathrm{e}^{\frac{h(\epsilon)\vee 2M \vee \bar{\gamma}}{\bar{\gamma}}}$, $\bar{\gamma} = \frac{1}{C_2}$ and*

$$\sigma = 3 \vee \left(\frac{8Lg^2}{C_1^2\bar{\eta}}\right)^{\frac{\mu}{\mu-3}} \vee \left(\frac{2}{\mu C_2 L^2\bar{\eta}^2}\right)^{\frac{\mu}{\mu-2}},$$

*where*

$$h(\epsilon) := \frac{4d}{\epsilon}\log\left(\frac{\mathrm{e}L}{M}\right) \vee \frac{8bd}{\epsilon^2} \vee 1,$$

*if we take $B = m = \sqrt{n}$, the largest permissible step size according to the value of $\sigma$, the gradient complexity to reach a precision of*

$$\mathbb{E}_{X_k \sim \rho_k}[F(X_k)] - F(X^*) \leq \epsilon$$

*is*

$$\tilde{O}\left(GC_1 + GC_2 + GC_3\right).$$

*where*

$$GC_1 = n g^{\frac{2\mu}{\mu-2}} C_1^{\frac{-2\mu}{\mu-2}} L^{\frac{2\mu}{\mu-2}} + n^{\frac{1}{2} - \frac{5}{2(\mu-5)}} \epsilon^{-\frac{\mu}{\mu-5}} g^{\frac{3\mu}{\mu-5}} C_1^{-\frac{3\mu}{\mu-5}} C_2^{\frac{\mu}{\mu-5}} (dE)^{\frac{\mu}{\mu-5}} L^{\frac{3\mu}{\mu-5}},$$

$$GC_2 = n^{\frac{1}{2} + \frac{\mu^2-3\mu+6}{2(\mu-2)(\mu-3)}} \left(\frac{gL}{C_1}\right)^{\frac{2\mu^2}{(\mu-2)(\mu-3)}}$$
$$+ n^{\frac{1}{2} - \frac{5(\mu-3)}{2(\mu^2-11\mu+15)}} \epsilon^{-\frac{\mu(\mu-1)}{\mu^2-11\mu+15}} \left(\frac{gL}{C_1}\right)^{\frac{(3\mu^2-7\mu+6)\mu}{(\mu-3)(\mu^2-11\mu+15)}} (dE)^{\frac{\mu(\mu-1)}{\mu^2-11\mu+15}},$$

$$GC_3 = n^{\frac{1}{2} + \frac{\mu^2+4}{2(\mu-2)^2}} \left(\frac{gL}{C_1}\right)^{\frac{2\mu^2}{(\mu-2)^2}} C_2^{-\frac{2\mu}{(\mu-2)^2}}$$
$$+ n^{\frac{1}{2} - \frac{5(\mu-2)}{2(\mu^2-13\mu+10)}} \epsilon^{-\frac{\mu(\mu+2)}{\mu^2-13\mu+10}} \left(\frac{gL}{C_1}\right)^{\frac{(3\mu-4)\mu}{\mu^2-13\mu+10}} (dE)^{\frac{\mu(\mu+2)}{\mu^2-13\mu+10}} C_2^{\frac{(\mu^2-12\mu-6)\mu}{(\mu^2-13\mu+10)(\mu-2)}},$$

$$E = 56 + 2\max_{\gamma \geq 1}\left(\frac{d}{\gamma}\log\left(\frac{\mathrm{e}L}{M}\left(\frac{b\gamma}{d} + 1\right)\right)\right) + 2F^*,$$

*and $C_1$ and $C_2$ are defined in Property C.3.*

*Proof.* Let us take $k = (s+1)m$, where $s \in \mathbb{N} \cup \{0\}$. From Corollary C.1.1, the sufficient condition for

$$\mathbb{E}_{X_k \sim \psi_k}[F(X_k)] - F(X^*) \leq \epsilon$$

is $LW_2^2(\psi_k, \nu_{\gamma_s}) \leq \epsilon/2$ and $\mathbb{E}_{X \sim \nu_{\gamma_s}}[F(X)] - F(X^*) \leq \epsilon/4$. From $g \geq \mathrm{e}^{\frac{2M}{\bar{\gamma}}}$, which implies $\gamma_s \geq \frac{2}{M}$, and from Corollary C.2.1, the latter condition is satisfied when $\gamma_s \geq \frac{4d}{\epsilon}\log\left(\frac{\mathrm{e}L}{M}\right) \vee \frac{8bd}{\epsilon^2} \vee 1$. Let us define

$$h(\epsilon) := \frac{4d}{\epsilon}\log\left(\frac{\mathrm{e}L}{M}\right) \vee \frac{8bd}{\epsilon^2} \vee 1.$$

Then, as $\gamma_s = \bar{\gamma}\log\left\{g(s+\sigma)^{\frac{1}{\mu}}\right\}$ and $s + \sigma \geq \mathrm{e}$, a sufficient condition is

$$\bar{\gamma}\log g \geq h(\epsilon).$$

This is satisfied with

$$g = e^{\frac{h(\epsilon) \vee 2M \vee \bar{\gamma}}{\bar{\gamma}}} \geq e^{\frac{h(\epsilon)}{\bar{\gamma}}}.$$

Moreover, concerning the former condition, from Talagrand's inequality

$$W_2^2(\rho_k, \nu_{\gamma_s}) \leq \frac{2}{\alpha_s} H_\nu(\rho_k),$$

it suffices to have

$$H_{\nu_{\gamma_s}}(\rho_k) \leq \frac{\alpha_s \epsilon}{4L} = \frac{\epsilon}{4L} \frac{C_1 C_2^{-1} \log g(s+\sigma)^{\frac{1}{\mu}}}{g(s+\sigma)^{\frac{1}{\mu}}}.$$

As $g \geq e$, $s + \sigma \geq 1$ and $(s + \sigma) \leq (s + 1)\sigma$, we obtain a simpler sufficient condition which is

$$H_{\nu_{\gamma_s}}(\rho_k) \leq \frac{\epsilon C_1 C_2^{-1}}{4Lk^{\frac{1}{\mu}} m^{\frac{-1}{\mu}} \sigma^{\frac{1}{\mu}} g}.$$

Therefore, from Theorem E.3, it is enough to take $\bar{\eta}$ and $k$ such that

$$\frac{8}{3} \bar{\eta} dL^2 Eg C_1^{-1} k^{\frac{2}{\mu}} m^{-\frac{2}{\mu}} \sigma^{\frac{2}{\mu}} \leq \frac{\epsilon C_1 C_2^{-1}}{8Lk^{\frac{1}{\mu}} m^{\frac{-1}{\mu}} \sigma^{\frac{1}{\mu}} g}, \tag{20}$$

and

$$k^{1-\frac{2}{\mu}} \geq 2gC_1^{-1} \left(\frac{3}{2}\right)^{\frac{2}{\mu}} m^{-\frac{2}{\mu}} \sigma^{\frac{2}{\mu}} \bar{\eta}^{-1} \log \left(\frac{8Lk^{\frac{1}{\mu}} m^{\frac{-1}{\mu}} \sigma^{\frac{1}{\mu}} H_0}{\epsilon C_1 C_2^{-1}}\right). \tag{21}$$

Concerning the first inequality (20), we obtain

$$\bar{\eta} \sigma^{\frac{3}{\mu}} \leq \frac{3C_1^2 C_2^{-1}}{64 dL^3 Eg^2} \epsilon k^{-\frac{3}{\mu}} m^{\frac{3}{\mu}}. \tag{22}$$

On the other hand, since $(s+1)^{\frac{1}{\mu}} \geq 1$ and $\sigma^{\frac{1}{\mu}} \geq 1$, as long as $\epsilon = O\left(\frac{LH_0}{C_1 C_2^{-1}}\right)$, we can consider the following condition for the second inequality (21):

$$k^{1-\frac{2}{\mu}} \geq \tilde{\Theta} \left(2gC_1^{-1} \left(\frac{3}{2}\right)^{\frac{2}{\mu}} m^{-\frac{2}{\mu}} \sigma^{\frac{2}{\mu}} \bar{\eta}^{-1}\right) = \tilde{\Theta} \left(gC_1^{-1} m^{-\frac{2}{\mu}} \sigma^{\frac{2}{\mu}} \bar{\eta}^{-1}\right). \tag{23}$$

(I) When

$$\sigma = 3 \vee \left(\frac{8Lg^2}{C_1^2 \bar{\eta}}\right)^{\frac{\mu}{\mu-3}} \vee \left(\frac{2}{\mu C_2 L^2 \bar{\eta}^2}\right)^{\frac{\mu}{\mu-2}} = 3,$$

equation (22) becomes

$$\bar{\eta} \leq \left(\frac{3C_1^2 C_2^{-1} 3^{\frac{-3}{\mu}}}{64 dL^3 Eg^2}\right) \epsilon k^{-\frac{3}{\mu}} m^{\frac{3}{\mu}}.$$

On the other hand, by plugging $\sigma = 3$ to (23), we obtain

$$k^{1-\frac{2}{\mu}} \geq \tilde{\Theta} \left(gC_1^{-1} m^{-\frac{2}{\mu}} \sigma^{\frac{2}{\mu}} \bar{\eta}^{-1}\right)$$
$$\geq \tilde{\Theta} \left(gC_1^{-1} m^{-\frac{2}{\mu}} \bar{\eta}^{-1}\right).$$

If inequality (22) is stronger than $0 < \bar{\eta} < \frac{C_1}{16\sqrt{6}L^2 m}$, then

$$k^{1-\frac{2}{\mu}} \geq \tilde{\Theta} \left(\frac{g^3 dEL^3}{C_1^3 C_2^{-1}} k^{\frac{3}{\mu}} m^{-\frac{5}{\mu}} \epsilon^{-1}\right).$$

This leads to

$$k \geq \tilde{\Theta} \left(\left(\frac{g^3 dEL^3}{C_1^3 C_2^{-1}}\right)^{\frac{\mu}{\mu-5}} m^{-\frac{5}{\mu-5}} \epsilon^{-\frac{\mu}{\mu-5}}\right).$$

From this, if we take the largest permissible step size and the smallest permissible $\sigma$, the gradient complexity can be calculated with an optimal order when $B = m = n^{\frac{1}{2}}$ as

$$\tilde{\Theta}\left(kB + \frac{k}{m}n\right) = \tilde{\Theta}(k\sqrt{n})$$

$$= \tilde{\Theta}\left(n^{\frac{1}{2} - \frac{5}{2(\mu-5)}} \epsilon^{-\frac{\mu}{\mu-5}} g^{\frac{3\mu}{\mu-5}} C_1^{-\frac{3\mu}{\mu-5}} C_2^{\frac{\mu}{\mu-5}} (dE)^{\frac{\mu}{\mu-5}} L^{\frac{3\mu}{\mu-5}}\right).$$

If inequality (22) is weaker than $0 < \bar{\eta} < \frac{C_1}{16\sqrt{6}gL^2m}$, then

$$k^{1-\frac{2}{\mu}} \geq \tilde{\Theta}\left(g^2 C_1^{-2} L^2 m^{1-\frac{2}{\mu}}\right).$$

This leads to

$$k \geq \tilde{\Theta}\left(\left(g^2 C_1^{-2} L^2\right)^{\frac{\mu}{\mu-2}} m\right).$$

From this, if we take the largest permissible step size and the smallest permissible $\sigma$, the gradient complexity can be calculated with an optimal order when $B = m = n^{\frac{1}{2}}$ as

$$\tilde{\Theta}\left(kB + \frac{k}{m}n\right) = \tilde{\Theta}(k\sqrt{n})$$

$$= \tilde{\Theta}\left(ng^{\frac{2\mu}{\mu-2}} C_1^{\frac{-2\mu}{\mu-2}} L^{\frac{2\mu}{\mu-2}}\right).$$

Therefore, we obtain the following gradient complexity for this case:

$$\tilde{\Theta}\left(ng^{\frac{2\mu}{\mu-2}} C_1^{\frac{-2\mu}{\mu-2}} L^{\frac{2\mu}{\mu-2}} + n^{\frac{1}{2} - \frac{5}{2(\mu-5)}} \epsilon^{-\frac{\mu}{\mu-5}} g^{\frac{3\mu}{\mu-5}} C_1^{-\frac{3\mu}{\mu-5}} C_2^{\frac{\mu}{\mu-5}} (dE)^{\frac{\mu}{\mu-5}} L^{\frac{3\mu}{\mu-5}}\right). \tag{24}$$

(II) When

$$\sigma = 3 \vee \left(\frac{8Lg^2}{C_1^2\bar{\eta}}\right)^{\frac{\mu}{\mu-3}} \vee \left(\frac{2}{\mu C_2 L^2 \bar{\eta}^2}\right)^{\frac{\mu}{\mu-2}} = \left(\frac{8Lg^2}{C_1^2\bar{\eta}}\right)^{\frac{\mu}{\mu-3}},$$

equation (22) becomes

$$\bar{\eta} \leq \left(\frac{3C_1^2 C_2^{-1}}{64dL^3 Eg^2}\right)^{\frac{\mu-3}{\mu-6}} \left(\frac{8Lg^2}{C_1^2}\right)^{\frac{-3}{\mu-6}} \epsilon^{\frac{\mu-3}{\mu-6}} k^{-\frac{3(\mu-3)}{\mu(\mu-6)}} m^{\frac{3(\mu-3)}{\mu(\mu-6)}}.$$

On the other hand, by plugging $\sigma = \left(\frac{8Lg^2}{C_1^2\bar{\eta}}\right)^{\frac{\mu}{\mu-3}}$ to (23), we obtain

$$k^{1-\frac{2}{\mu}} \geq \tilde{\Theta}\left(gC_1^{-1} m^{-\frac{2}{\mu}} \sigma^{\frac{2}{\mu}} \bar{\eta}^{-1}\right)$$

$$\geq \tilde{\Theta}\left(g^{\frac{\mu+1}{\mu-3}} C_1^{-\frac{\mu+1}{\mu-3}} L^{\frac{2}{\mu-3}} m^{-\frac{2}{\mu}} \bar{\eta}^{-\frac{\mu-1}{\mu-3}}\right).$$

If inequality (22) is stronger than $0 < \bar{\eta} < \frac{C_1}{16\sqrt{6}gL^2m}$, then

$$k^{1-\frac{2}{\mu}} \geq \tilde{\Theta}\left(\left(\frac{C_1^2 C_2^{-1}}{dEL^3 g^2}\right)^{-\frac{\mu-1}{\mu-6}} \left(\frac{Lg^2}{C_1^2}\right)^{\frac{3(\mu-1)}{(\mu-6)(\mu-3)}} \frac{g^{\frac{\mu+1}{\mu-3}} L^{\frac{2}{\mu-3}}}{C_1^{\frac{\mu+1}{\mu-3}}} \epsilon^{-\frac{\mu-1}{\mu-6}} k^{\frac{3(\mu-1)}{\mu(\mu-6)}} m^{-\frac{5(\mu-3)}{\mu(\mu-6)}}\right).$$

This leads to

$$k \geq \tilde{\Theta}\left(\frac{\left(\left(\frac{C_1^2 C_2^{-1}}{dEL^3 g^2}\right)^{-\frac{\mu-1}{\mu-6}} \left(\frac{Lg^2}{C_1^2}\right)^{\frac{3(\mu-1)}{(\mu-6)(\mu-3)}} g^{\frac{\mu+1}{\mu-3}} C_1^{-\frac{\mu+1}{\mu-3}} L^{\frac{2}{\mu-3}}\right)^{\frac{\mu(\mu-6)}{\mu^2-11\mu+15}}}{m^{\frac{5(\mu-3)}{\mu^2-11\mu+15}} \epsilon^{\frac{\mu(\mu-1)}{\mu^2-11\mu+15}}}\right).$$

From this, if we take the largest permissible step size and the smallest permissible $\sigma$, the gradient complexity can be calculated with an optimal order when $B = m = n^{\frac{1}{2}}$ as

$$\tilde{\Theta}\left(kB + \frac{k}{m}n\right) = \tilde{\Theta}(k\sqrt{n})$$

$$= \tilde{\Theta}\left(n^{\frac{1}{2} - \frac{5(\mu-3)}{2(\mu^2-11\mu+15)}} \epsilon^{-\frac{\mu(\mu-1)}{\mu^2-11\mu+15}}\left(\frac{gL}{C_1}\right)^{\frac{(3\mu^2-7\mu+6)\mu}{(\mu-3)(\mu^2-11\mu+15)}}(dE)^{\frac{\mu(\mu-1)}{\mu^2-11\mu+15}}\right).$$

If inequality (22) is weaker than $0 < \bar{\eta} < \frac{C_1}{16\sqrt{6}gL^2m}$, then

$$k^{1-\frac{2}{\mu}} \geq \tilde{\Theta}\left(\left(\frac{gL}{C_1}\right)^{\frac{2\mu}{\mu-3}} m^{\frac{\mu^2-3\mu+6}{\mu(\mu-3)}}\right).$$

This leads to

$$k \geq \tilde{\Theta}\left(\left(\frac{gL}{C_1}\right)^{\frac{2\mu^2}{(\mu-2)(\mu-3)}} m^{\frac{\mu^2-3\mu+6}{(\mu-2)(\mu-3)}}\right).$$

From this, if we take the largest permissible step size and the smallest permissible $\sigma$, the gradient complexity can be calculated with $B = m = n^{\frac{1}{2}}$ as

$$\tilde{\Theta}\left(kB + \frac{k}{m}n\right) = \tilde{\Theta}(k\sqrt{n})$$

$$= \tilde{\Theta}\left(n^{\frac{1}{2} + \frac{\mu^2-3\mu+6}{2(\mu-2)(\mu-3)}}\left(\frac{gL}{C_1}\right)^{\frac{2\mu^2}{(\mu-2)(\mu-3)}}\right).$$

Therefore, we obtain the following gradient complexity for this case:

$$\tilde{\Theta}\left(n^{\frac{2\mu^2-8\mu+12}{2(\mu-2)(\mu-3)}}\left(\frac{gL}{C_1}\right)^{\frac{2\mu^2}{(\mu-2)(\mu-3)}} + \frac{n^{\frac{1}{2} - \frac{5(\mu-3)}{2(\mu^2-11\mu+15)}}}{\epsilon^{\frac{\mu(\mu-1)}{\mu^2-11\mu+15}}}\left(\frac{gL}{C_1}\right)^{\frac{(3\mu^2-7\mu+6)\mu}{(\mu-3)(\mu^2-11\mu+15)}}(dE)^{\frac{\mu(\mu-1)}{\mu^2-11\mu+15}}\right). \tag{25}$$

(III) When

$$\sigma = 3 \vee \left(\frac{8Lg^2}{C_1^2\bar{\eta}}\right)^{\frac{\mu}{\mu-3}} \vee \left(\frac{2}{\mu C_2L^2\bar{\eta}^2}\right)^{\frac{\mu}{\mu-2}} = \left(\frac{2}{\mu C_2L^2\bar{\eta}^2}\right)^{\frac{\mu}{\mu-2}},$$

equation (22) becomes

$$\bar{\eta} \leq \left(\frac{3C_1^2C_2^{-1}}{64dL^3Eg^2}\right)^{\frac{\mu-2}{\mu-8}}\left(\frac{2}{\mu C_2L^2}\right)^{\frac{-3}{\mu-8}}\epsilon^{\frac{\mu-2}{\mu-8}}k^{-\frac{3(\mu-2)}{\mu(\mu-8)}}m^{\frac{3(\mu-2)}{\mu(\mu-8)}}.$$

On the other hand, by plugging $\sigma = \left(\frac{2}{\mu C_2L^2\bar{\eta}^2}\right)^{\frac{\mu}{\mu-2}}$ to (23), we obtain

$$k^{1-\frac{2}{\mu}} \geq \tilde{\Theta}\left(gC_1^{-1}m^{-\frac{2}{\mu}}\sigma^{\frac{2}{\mu}}\bar{\eta}^{-1}\right)$$

$$\geq \tilde{\Theta}\left(gC_1^{-1}(C_2L^2)^{-\frac{2}{\mu-2}}m^{-\frac{2}{\mu}}\bar{\eta}^{-\frac{\mu+2}{\mu-2}}\right).$$

If inequality (22) is stronger than $0 < \bar{\eta} < \frac{C_1}{16\sqrt{6}gL^2m}$, then

$$k^{1-\frac{2}{\mu}} \geq \tilde{\Theta}\left(gC_1^{-1}(C_2L^2)^{\frac{-2}{\mu-2}}\left(\frac{C_1^2C_2^{-1}}{dEL^3g^2}\right)^{-\frac{\mu+2}{\mu-8}}(C_2L^2)^{\frac{-3(\mu+2)}{(\mu-8)(\mu-2)}}k^{\frac{3(\mu+2)}{\mu(\mu-8)}}m^{-\frac{5(\mu-2)}{\mu(\mu-8)}}\epsilon^{\frac{\mu+2}{\mu-8}}\right).$$

This leads to

$$k \geq \tilde{\Theta}\left(\frac{\left(gC_1^{-1}(C_2L^2)^{\frac{-2}{\mu-2}}\left(\frac{C_1^2C_2^{-1}}{dEL^3g^2}\right)^{-\frac{\mu+2}{\mu-8}}(C_2L^2)^{\frac{-3(\mu+2)}{(\mu-8)(\mu-2)}}\right)^{\frac{\mu(\mu-8)}{\mu^2-13\mu+10}}}{m^{\frac{5(\mu-2)}{\mu^2-13\mu+10}}\epsilon^{\frac{\mu(\mu+2)}{\mu^2-13\mu+10}}}\right).$$

From this, if we take the largest permissible step size and the smallest permissible $\sigma$, the gradient complexity can be calculated with an optimal order when $B = m = n^{\frac{1}{2}}$ as

$$\tilde{\Theta}\left(kB + \frac{k}{m}n\right) = \tilde{\Theta}(k\sqrt{n})$$

$$= \tilde{\Theta}\left(\frac{n^{\frac{1}{2} - \frac{5(\mu-2)}{2(\mu^2-13\mu+10)}}}{\epsilon^{\frac{\mu(\mu+2)}{\mu^2-13\mu+10}}}\left(\frac{gL}{C_1}\right)^{\frac{(3\mu-4)\mu}{\mu^2-13\mu+10}}(dE)^{\frac{\mu(\mu+2)}{\mu^2-13\mu+10}}C_2^{\frac{(\mu^2-12\mu-6)\mu}{(\mu^2-13\mu+10)(\mu-2)}}\right).$$

If inequality (22) is weaker than $0 < \bar{\eta} < \frac{C_1}{16\sqrt{6}gL^2m}$, then

$$k^{1-\frac{2}{\mu}} \geq \tilde{\Theta}\left(gC_1^{-1}(C_2L^2)^{-\frac{2}{\mu-2}}\left(\frac{gL^2}{C_1}\right)^{\frac{\mu+2}{\mu-2}}m^{\frac{\mu^2+4}{\mu(\mu-2)}}\right).$$

This leads to

$$k \geq \tilde{\Theta}\left(\left(gC_1^{-1}(C_2L^2)^{-\frac{2}{\mu-2}}\left(\frac{gL^2}{C_1}\right)^{\frac{\mu+2}{\mu-2}}\right)^{\frac{\mu}{\mu-2}}m^{\frac{\mu^2+4}{(\mu-2)^2}}\right).$$

From this, if we take the largest permissible step size and the smallest permissible $\sigma$, the gradient complexity can be calculated with $B = m = n^{\frac{1}{2}}$ as

$$\tilde{\Theta}\left(kB + \frac{k}{m}n\right) = \tilde{\Theta}(k\sqrt{n})$$

$$= \tilde{\Theta}\left(n^{\frac{1}{2} + \frac{\mu^2+4}{2(\mu-2)^2}}\left(\frac{gL}{C_1}\right)^{\frac{2\mu^2}{(\mu-2)^2}}C_2^{-\frac{2\mu}{(\mu-2)^2}}\right).$$

Therefore, we obtain the following gradient complexity for this case:

$$\tilde{\Theta}\left(n^{\frac{\mu^2-2\mu+4}{(\mu-2)^2}}\left(\frac{gL}{C_1C_2^{\frac{1}{\mu}}}\right)^{\frac{2\mu^2}{(\mu-2)^2}} + \frac{n^{\frac{\mu^2-18\mu+20}{2(\mu^2-13\mu+10)}}}{\epsilon^{\frac{\mu(\mu+2)}{\mu^2-13\mu+10}}}\left(\frac{gL}{C_1}\right)^{\frac{(3\mu-4)\mu}{\mu^2-13\mu+10}}(dE)^{\frac{\mu(\mu+2)}{\mu^2-13\mu+10}}C_2^{\frac{(\mu^2-12\mu-6)\mu}{(\mu^2-13\mu+10)(\mu-2)}}\right). \quad (26)$$

The statement of the theorem is obtained by grouping (24), (25) and (26).

$$Q.E.D$$

Now, looking at the gradient complexity of Theorem E.4, we remark that the dependence on $n$ is slightly improved for finite values of $\mu$ compared with $\mu \to \infty$. However, taking into account that $g = e^{C_2h(\epsilon)}$ most of the time, the dependence on $\epsilon$ becomes worse as the exponent of $g$ of the first term is greater than 2, and that of the second term is greater than 3 for all $GC_i$ ($i = 1, 2, 3$). Since this influence cannot be ignored in this case, we conclude that the best value of $\mu$ is $\mu = \infty$ in our analysis, i.e., the method that keeps $\eta$ and $\gamma$ constants.