# OpenReview forum: "Improved Convergence Rate of Stochastic Gradient Langevin Dynamics with Variance Reduction and its Application to Optimization"
_NeurIPS.cc/2022/Conference — NeurIPS 2022 Accept_

### Official Review · Reviewer_UBuG · 2022-07-10

**Rating:** 6
**Confidence:** 2
**Soundness:** 3 good
**Presentation:** 3 good
**Contribution:** 3 good

**Summary:**

This paper presents improved convergence rates of SGLD with variance reduction. Also, it presents an application to non-convex optimization -- namely an upper bound on the gradient complexity necessary to reach a certain complexity, in SVRG-LD and SARAH-LD methods. The results are novel -- they take the same assumptions of Vempala et al. (2019), who also used the KL divergence as a criterion, but derive a better gradient complexity. Other works have used other assumptions or criteria. Crucially, however, this paper assumes that the batch size and inner loop length is set to \sqrt{n}.

**Questions:**

1. I didn't understand what the "inner loop length" is. This concept needs some clarification in my opinion.
2. Doesn't one have to assume, in Assumption 2, that \rho is asymptotically continuous wrt. \nu.
3. The paragraph on top of page 9, before "Analysis under the weak Morse condition", is very confusing. Why is the second term with n^{1/2} almost all the time dominant? Why does the analysis of Mattingly et al. (2002) support this? Why use "likely" if the work of Raginsky et al. (2017) implies it? (Not a big deal, but maybe this paragraph can be improved.)

**Ethics Review Area:**

["I don’t know"]

**Limitations:**

No limitations and no negative societal impact.

**Strengths And Weaknesses:**

This is theoretically a very strong paper. The main weakness is that its implications are a bit restrictive, as SVRG-LD and SARAH-LD aren't used so much in practice, as far as I know. (In fact, variance reduction is not used much in practice; instead, it has been shown recently that the stochasticity in SGD is vital for generalization and escaping minima/saddle-points.)

Strengths:
The analysis is very detailed and technically involved. I did not read the Appendix, but could follow the exposition in the main paper. The literature review is very good and contains most relevant prior work. Table 1 is very helpful to the reader. (A few more words about the comparison with Vempala et al. (2019) could be added to the caption though.) The texts are clear and easy to read, and the authors are transparent about their assumptions.

Weaknesses:
1. The discussion of variance reduction methods is lacking in nuance: Lately it has been demonstrated the _stochasticity_, i.e. the variance, actually makes SGD so good by improving generalization. See e.g. https://arxiv.org/abs/2006.15081?context=stat or https://arxiv.org/pdf/2202.02831.pdf . Also, the variance is useful to escape saddle points and local minima: https://opt-ml.org/papers/2021/paper24.pdf and https://arxiv.org/abs/1902.04811. As far as I know, people don't really use variance reduction in practice, for precisely this reason. This is not a fundamental criticism of the paper: It is still important to study variance reduction, but there should be a discussion of the role of stochasticity and the usefulness of the studied methods in the Introduction. Also, the restriction of the results to variance-reduced method of course reduces the significance of the result.
2. There should be a better discussion of the assumption that "the batch size and the inner loop length are set to \sqrt{n}". While the other assumptions are fairly standard, this assumption is highly unusual. Also, it doesn't appear in Table 1 (why?). There should be a better discussion of this assumption, if this appears in other papers, too, and why it is necessary.
3. Some plots or experiments demonstrating these experiments would be nice. (The proofs are very complicated and it would make the results more plausible if there were a few experiments, demonstrating them.)

Bottom line: If the authors provide a few experiments, I am fine with accepting this paper.

---

> ### Author Response · Authors · 2022-08-01
> **Thank you very much for spending your time on carefully reviewing our paper.**
>
> We really appreciate all the advice you provide to improve our paper. Please find below answers to your questions and some clarifications to other important points of your review.
> > a. A few more words [...] to the caption
> b. [...] Assumption 2, that \rho is asymptotically continuous wrt. \nu.
>  c. The paragraph on top of page 9 [...] is very confusing. [...]
>
> Thank you very much for pointing out these issues. We took into account all of them in the revised version.
> > The discussion of variance reduction methods is lacking in nuance […]
>
> First of all, we would like to emphasize that SGD and SGLD should not be considered as the same stochastic algorithm. In the case of GLD, noise is already intentionally added which results in a strongly noisy algorithm with the scheme of SGLD. It is thus important to reduce superfluous noises engendered by mini-batching to accelerate the convergence while keeping the benefits of SGD. In that sense, variance reduction techniques are theoretically and practically of high relevance. Furthermore, even though we processed variance reduction, stochasticity is conserved since the stationary distribution is the same. One of the biggest difficulties overcome here is precisely this balance between the noise that enables escape saddle points and variance reduction that attenuates noises. We not only realized this but also provided the scenario with the best trade-off.
> The benefits of variance of SGD (generalization and escape from saddle points) were also mentioned in this part of review. Even though generalization is important, we focused more on the precise behavior of our variance reduction methods as numerical algorithms, as it is the case for other usual theoretical research on optimization algorithms and MCMC methods. There are many cases where rigorous calculation is needed instead of generalization such as sampling from Bayesian posterior distribution. As for saddle points escape, we showed this was possible as well. We even proved in Section 4 global convergence under the weak Morse assumption (similar to strict saddle), better than a simple escape from saddle points of SGLD. Since sampling can also be seen as an optimization problem, in both Sections 3 and 4, escape from saddle points is realized.
> As mentioned earlier, all this is achieved as we succeeded in finding the right balance between the stochasticity and variance reduction. Therefore, we believe our result to the variance-reduced methods does not reduce the significance of the paper and shows the possibility that variance reduction methods can be successful in Langevin Dynamics.
> > There should be a better discussion of the assumption that "the batch size and the inner loop length are set to \sqrt{n}". [...]
>
> Maybe we were not clear, but this is **neither an assumption nor a condition** at all. We only set the batch size and the inner loop length to $\Theta( \sqrt{n})$ in order to obtain the best gradient complexity. The general gradient complexity is provided in the proof of Corollary A.2.1. as $O(n+(n/m+B)/\epsilon)$ with $mB=O(n)$. Since $B\ge m$ was required, the optimal value is obtained with $m=\Theta(\sqrt{n})$ and $B=\Theta(\sqrt{n})$. This is of course not a necessary condition for the convergence proof as it can be seen in Theorems 1 and 2 with no assumptions of this kind on B and m.
> This optimal value of B and m is totally standard in the literature on variance reduction algorithms. See for example, Zou et al. (2019a) for SVRG-LD, Li et al. (2019) for SSRGD. There are even papers that actually prove theoretically and experimentally that the optimal batch size is of order √n for stochastic gradient in some cases [1].
> On the other hand, we agree there is still a gap between theory and practice. This batch size may become impractical in the case n is really huge. One of the main goals of theoretical papers like ours is precisely to fill this gap. In this regard, our work should be viewed as an improvement since previous work on GLD and its stochastic scheme have suggested bigger batch sizes, which even depend on the accuracy \epsilon, than ours (see for example Xu et al., 2018).
>
> > [...] if there were a few experiments, demonstrating them
>
> Thank you very much for your advice. We would like to include experiments in the camera ready version.
> > I didn't understand what the "inner loop length" is. This concept needs some clarification in my opinion.
>
> We apologize for not being clear. In Algorithm 1, there are two “for loops”, one inside the other. The outer loop is that with length K/m and the inner loop is that with length m. This m is called "inner loop length".
>
> We hope we could address all your questions and concerns. We will be glad to provide further explanation and clarification if necessary.
>
> [1] Nitanda, A., Murata, T., & Suzuki, T. (2021). Sharp characterization of optimal minibatch size for stochastic finite sum convex optimization. Knowledge and Information Systems, 63(9), 2513-2539.

---

> > ### Comment · Reviewer_UBuG · 2022-08-03
> > **Thank you for these clarifications**
> >
> > Thank you for these clarifications! After reading these replies and the other reviews I am in favor of accepting this paper and will raise my score accordingly!

---

### Official Review · Reviewer_4MBs · 2022-07-11

**Rating:** 6
**Confidence:** 3
**Soundness:** 3 good
**Presentation:** 3 good
**Contribution:** 2 fair

**Summary:**

In this paper, the authors proved the convergence rate of the Stochastic Variance Reduced Gradient Langevin Dynamics and the Stochastic Recursive Gradient Langevin Dynamics. The proof is based on Log-Sobolev inequality and a relaxed assumption of smoothness and the result shows an improvement from the previous analysis.


**Questions:**

1. Wondering if the authors could comment on the scale of the learning rate. I feel in order to have Theorem 1 and Corollary 1.1. to hold, the required learning rate could be quite small.

2. While setting the batch size to be $\sqrt(n)$ leads to an improved gradient complexity, such batch size is very large and rarely used in practice. What would the gradient complexity look like if using a much smaller batch size?


**Limitations:**

In order to have the theorems hold, the choice of a few hyper-parameters, such as learning rate and batch size seem, seem to be far away from what has been used in practice.


**Strengths And Weaknesses:**

This paper provides a non-asymptotic analysis of the convergence of SVRG-LD and SARAH-LD to the Gibbs distribution in terms of KL-divergence. The analysis is based on the relaxed assumptions of smoothness and LSI which are weaker conditions than those used in prior works for these algorithms. Overall, the proven convergence rate of SVRG is better than in previous work.


However, I feel the result seems to be an incremental improvement of the work by Li and Erdogdu (2020) and Vempala and Wibisono (2019).

---

> ### Author Response · Authors · 2022-08-01
> **Thank you very much for spending your time on carefully reviewing our paper.**
>
> Please find below answers to your questions and some clarifications to other important points of your review.
>
> > I feel the result seems to be an incremental improvement of the work by Li and Erdogdu (2020) and Vempala and Wibisono (2019).
>
> Thank you for sharing this point of view. However, we would like to emphasize that our work is not a simple incremental improvement of previous work.
> Neither Vempala and Wibisono (2019) nor Li and Erdogdu (2020) discussed stochastic gradient methods with variance reduction. In fact, variance reduction methods are on their own an important and huge theme of research in the field of stochastic optimization and the study of the effect of variance reduction on existing algorithms is an important topic of research which attracts a lot of interests of the community (Johnson et al., 2013; Dubey et al., 2016). Moreover, in general, the extension of known algorithms to variance reduction schemes is also far from being straightforward. In our paper, the stochastic gradient and GLD both add important noises, and in order to find a good balance between them to assure the best convergence of SVRG-LD while keeping the advantages of the stochastic gradient such as decreased gradient computation and escape from saddle points, an adequate evaluation of the variance of the stochastic gradient and a precise estimation of the step size were crucial. One of the biggest difficulties overcome in our paper is precisely this balance between the noise that enables escape saddle points and the variance reduction method that attenuates noises. We not only realized this balance but also provided the scenario with the best trade-off. It was also not trivial whether acceleration of convergence in terms of KL-divergence, a strong criterion, was possible and needed some work on it. Therefore, we believe our contribution does not limit to an incremental improvement with regard to the whole line of research on variance reduction and Langevin Dynamics, and our paper shows the possibility that our researched variance reduction methods can be successful in Langevin Dynamics.
>
> > Wondering if the authors could comment on the scale of the learning rate. [...] the required learning rate could be quite small.
>
> Indeed, the learning rate (or step size) could become quite small in some situations. Nevertheless, this is unavoidable to compensate for the use of mini batches and is actually indispensable to obtain the benefit of variance reduction as we can observe in our proof. As overall this leads to a better gradient complexity, we believe this result on the learning rate is acceptable in our problem setting with SVRG-LD and SARAH-LD.
>
> > While setting the batch size to be √n leads to an improved gradient complexity, such batch size is very large and rarely used in practice.
>
> Thank you for raising this point. We would like to first emphasize that this is totally standard in the literature of variance reduction algorithms. Many papers similarly showed the optimal upper bound is realized with this kind of batch size. See for example, Zou et al. (2019a) for SVRG-LD, Li (2019) for SSRGD. There are even papers that actually prove theoretically and experimentally that the optimal batch size is of order √n for stochastic gradient algorithms in some cases [1].
> On the other hand, we agree there is still a gap between theory and practice. This batch size may become impractical in the case n is really huge. One of the main goals of theoretical papers like ours is precisely to fill this gap. In this regard, our work should be viewed as an improvement since some previous work on GLD and its stochastic schemes have suggested bigger batch sizes, which even depend on the accuracy $\epsilon$, than ours (see for example Xu et al., 2018).
>
> > What would the gradient complexity look like if using a much smaller batch size.
>
> As long as $mB=O(n)$ and $B\ge m$, Corollary A.2.1 tells us the gradient complexity is $O(n+(n/m+B)/\epsilon)$, with the lowest upper bound realized when substituting $B=m=\Theta(√n)$. A similar argument can be still constructed based on smaller batch sizes, inevitably providing a bigger gradient complexity than our optimal result. For example, when $B=m=n^{1/3}$, the gradient complexity becomes $O(n+n^{2/3}/\epsilon)$. When $B=m=n^{1/10}$, $O(n+n^{9/10}/\epsilon) $etc. In conclusion, for any B, we will always obtain an improvement compared to the deterministic gradient algorithm which has a gradient complexity of $O(n/\epsilon)$. Therefore, our paper can account for the empirical success of stochastic gradient algorithms for diverse values of B as well.
>
> We hope we could address all your questions and concerns. We will be glad to provide further explanation and clarification if necessary.
>
> [1] Nitanda, A., Murata, T., & Suzuki, T. (2021). Sharp characterization of optimal minibatch size for stochastic finite sum convex optimization. Knowledge and Information Systems, 63(9), 2513-2539.

---

> > ### Comment · Reviewer_4MBs · 2022-08-08
> > **Thank you very much for your clarification**
> >
> > Thank you very much for your clarifications. I think they have addressed most of my concerns. I will rise my score to 6.

---

### Official Review · Reviewer_rRnx · 2022-07-11

**Rating:** 7
**Confidence:** 3
**Soundness:** 3 good
**Presentation:** 4 excellent
**Contribution:** 2 fair

**Summary:**

The connection between Langevin sampling and optimization has been extensively studied. A line of research in literature tries to adapt the known optimization techniques to get new sampling algorithms. This paper follows the same direction. The authors studies two algorithms SVRG-LD and SARAH-LD which are respectively based on the variance-reduced stochastic optimization algorithms SVRG and SARAH. Under the smoothness and LSI conditions the convergence of both algorithms is established in the KL divergence through a descent-like inequality. Under additional conditions, they apply these algorithms for the problem of non-convex optimization.

**Questions:**

* How does your work compare to the above mentioned paper?
* In the statement of the theorems 1 and 2, the introduction of the constant $\Lambda$  seems to be redundant as $\Lambda = 1 + 2\Xi$.



**Limitations:**

* The assumption 6 is restrictive as one might have multiple global minima.
* The discussion on the limitations of the work seem to be absent in the paper.

**Strengths And Weaknesses:**

Strengths
* The paper is very well written. It explains the problem and the setting very clearly. One of the best written papers I have reviewed this year.
* The proposed methods work in the relaxed LSI regime instead of strong convexity.


Weaknesses
* In the literature review the paper _"On the Theory of Variance Reduction for SG-MC " _ by Chetterji et al. is missing. This paper studies a similar problem. In particular, they propose an adaptation of the SAGA algorithm to the LMC problem. After a brief overview of the results, I saw that the their iteration complexity of Chatterji et al. is smaller than the one proposed in this paper. A discussion on this paper is required.

---

> ### Author Response · Authors · 2022-08-01
> **Thank you very much for spending your time on carefully reviewing our paper.**
>
> We really appreciate your comment on the presentation quality. Please find below answers to your questions and some clarifications to other important points of your review.
>
> > In the literature review the paper _"On the Theory of Variance Reduction for SG-MC " _ by Chetterji et al. is missing. […] the iteration complexity of Chatterji et al. is smaller than the one proposed in this paper. A discussion on this paper is required.
> How does your work compare to the above mentioned paper?
>
> The work of Chatterji et al. [1], which studies two sampling algorithms SAGA-LD and SVRG-LD, is indeed an important paper that we did not include in ours. Thank you for pointing out this. However, it is important to note that we clearly worked on a broader problem setting than Chatterji et al. [1] and that their gradient complexity is not smaller than ours. First, Chatterji et al. [1] assumed (smoothness (A2),) strong convexity (A3) and Hessian Lipschitzness (A4) to prove their result, and these are stricter conditions than those we imposed in our paper (only LSI and smoothness for sampling). Especially, strong convexity of Chatterji et al. [1] is a really strong assumption. On the contrary, all of our results were proved for **non-convex** functions, and we even derived global convergence. Moreover, our convergence guarantee is proved in terms of KL divergence which is a stronger criterion (i.e., can upper bound) than that used in Chatterji et al. [1], i.e., 2-Wasserstein distance. While the gradient complexity alone may seem to be better than ours, the objective function in Chatterji et al. [1] is actually different. That is, it does not have the factor 1/n before the finite sum. Zou et al. [2] re-wrote the result of Chatterji et al. [1] for our problem setting in Table 1 of their paper. It appears that gradient complexities for SVRG-LD and for SAGA-LD proved by Chatterji et al. [1] become $\tilde O(n+n^{2/3}/\epsilon)$ and $\tilde O(n+n^{1/2}/\epsilon)$ respectively, which are equivalent or worse than ours even with better assumptions in our case. Zou et al. [2] already provided an improved result of Chatterji et al. [1] for SVRG-LD. This can be found in Table 1 (of our paper), which we adjusted in the revised version. Therefore, there is a clear improvement made in our paper compared to Chatterji et al. [1].
> We added a reference in the revised version (see line 49). As we do not have enough space, further explanation on SAGA-LD will be added in the additional one page, if the paper is accepted.
>
> > In the statement of the theorems 1 and 2, the introduction of the constant Λ seems to be redundant as Λ=1+2Ξ.
>
> Thank you for pointing out this issue. It is indeed redundant, and we have corrected this in the revised version of our paper.
>
> > The assumption 6 is restrictive as one might have multiple global minima.
>
> We would like to first emphasize that this assumption is only used to prove Corollary 3.2  (indexes of statement have changed a bit in the revised version), and we believe it does not constitute a major limitation of our work (please notice that our main results, Theorems 1,2 and 3, do not require Assumption 6). The main concern of our paper is sampling from a probability distribution, and we only used Assumption 6 to show that an acceleration of global convergence in non-convex optimization from exponential to polynomial was possible by adding Assumptions 4 to 6. Concerning this Corollary 3.2, Assumption 6 is indeed quite restrictive and further research could be conducted to relax this condition.
>
> > The discussion on the limitations of the work seem to be absent in the paper.
>
> We apologize for this. Please find a discussion on the limitations in the Discussion and Conclusion section of the revised version of our paper.
>
> We hope we could address all your questions and concerns. We will be glad to provide further explanation and clarification if necessary.
>
> [1] Chatterji, N., Flammarion, N., Ma, Y., Bartlett, P., & Jordan, M. (2018, July). On the theory of variance reduction for stochastic gradient Monte Carlo. In International Conference on Machine Learning (pp. 764-773). PMLR.
> [2] Zou, D., Xu, P., & Gu, Q. (2018, January). Subsampled stochastic variance-reduced gradient Langevin dynamics. In International Conference on Uncertainty in Artificial Intelligence.

---

> > ### Comment · Reviewer_rRnx · 2022-08-03
> > **My questions were answered.**
> >
> > The paper deserves to be published. I raise my score to 7.
> >
> >
> > minor remark: The hyperlink (Section 4) in the discussion section does not work.

---

### Meta-Review · Area_Chair_88uS · 2022-08-26

**Recommendation:** Accept
**Confidence:** Certain

**Metareview:**

This paper proposes an improved convergence rate for stochastic gradient Langevin dynamics with variance reduction under smoothness and Log-Sobolev inequality assumptions, which improves a long line of prior works. After author response and reviewer discussion, the paper receives unanimous support from the reviewers. Thus, I recommend acceptance.

**Award:**

No

---

### Decision · Program_Chairs · 2022-09-14

Accept